# CLIP-DR: Robust Test-Time Adaptation via Dual Regularization Beyond Optimal Transport

## Abstract

Despite the remarkable zero-shot performance of vision-language models, such as Contrastive Language-Image Pretraining (CLIP), on many downstream tasks, their potential may be degraded under distributional shifts. Test-time adaptation (TTA) offers a solution by adapting the model to these shifts during inference, without requiring labeled data. Prior methods like CLIP-OT leverage optimal transport for pseudo-labeling. However, the quality of these labels can be unreliable, leading to suboptimal adaptation and error accumulation. To address this, we propose CLIP-DR, which introduces two extra key components: (1) a cosine similarity loss to align image features with textual prototypes, stabilizing the adaptation direction; and (2) an information maximization regularizer to promote confident and diverse predictions, preventing model collapse. Extensive evaluation on seven benchmarks (covering 15 corruption types and domain shifts, totaling ∼6000 trials) demonstrates that CLIP-DR consistently outperforms state-of-the-art methods while adding ∼0.01 seconds of computing time per batch (e.g., 4% and 12% higher than CLIP-OT and WATT-S on the TinyImageNet-C dataset with 1.98 second per batch). The codes are available at https://anonymous.4open.science/r/TTA-Codes-6DF3.

## 1 Introduction

Vision language models (VLMs), such as contrastive language-image pretraining (CLIP) model Radford et al. (2021); Wu et al. (2025); Yang et al. (2025), have demonstrated promising zero-shot transferability due to their strong semantic feature understanding capabilities. However, if there exists a distribution or texture shifts between the training set and the test set, the performance of the model can be degraded Silva-Rodriguez et al. (2024). Intuitively, a simple solution for this challenge is to fine-tune the trained model using domain-specific labeled data Goyal et al. (2023); Chaddad et al. (2025). These approaches have several limitations in real-world applications. For example, they require a large amount of labeled data, which can be difficult to obtain Carvalho & Abad (2025). Furthermore, fine-tuning the model can degrade its transferability (e.g., its zero-shot capabilities) Kim et al. (2024).

Test-time adaptation (TTA) introduces a practical solution to address these limitations Tong et al. (2025); Rifat et al. (2025). The key in TTA is to optimize a pre-trained model in a real-time adaptation situation without accessing any supervisory signals (e.g., label) Hu et al. (2025). However, despite the progress in convolutional neural network (CNN) and vision transformer (ViT) based TTA, the study related to Vision-Language Models (VLMs) remains less explored. Initial methods, such as entropy minimization (TENT) Wang et al., pseudo-labels guidance Osowiechi et al. (2024) have been used for CLIP adaptations. However, these approaches can lead to suboptimal solutions due to insufficient supervision information Hakim et al. (2025). Recent studies, such as CLIP-OT, uses optimal transport (OT) to align the outputs of CLIP and the pseudo logits generated by text prototypes Mishra et al. (2025). This achieves promising performance with reasonable computational overhead on several TTA benchmarks. However, using OT alone can lead to poor pseudo logit quality, thereby reducing reliability in prediction. Figure 1 shows an example of CLIP-OT in prediction using a reliability diagram. CLIP-OT provides a higher expected calibration error (ECE Zhang et al. (2025)), indicating its overconfidence and poor pseudo-label quality during inference.

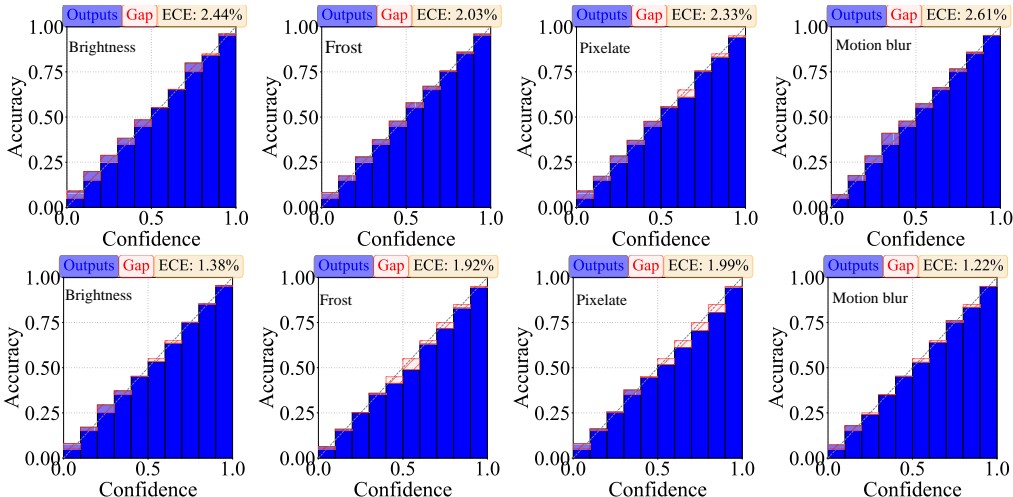

Figure 1: CLIP-OT (**first row**) vs. CLIP-DR (**second row**). Relying on OT loss leads to overconfidence for model prediction, resulting in a higher calibration error.

In light of the above limitation, this study introduces two components to refine the quality of the pseudo-labels in CLIP-OT: (i) a cosine similarity loss to align the predicted logits of the CLIP and the virtual logits generated by text prototypes, and (ii) an information maximization (IM) strategy to regularize the virtual logits in an unsupervised manner. Experimental results show that the proposed approach provides a feasible improvement in test accuracy (e.g., 5.42% higher on Tiny-ImageNet-C dataset) with nearly identical computation load compared to CLIP-OT.

The contributions of this paper can be summarized as follows:

1. *Algorithm.* To improve the quality of the pseudo-labels in CLIP-OT, we extended CLIP-OT by: (i) We design a self-supervised cosine similarity loss that encourages the model to learn high quality pseudo logits, thereby improving the robustness to image corruptions. (ii) We propose using an information maximization loss function to regularize the predicted logits. This reduces prediction uncertainties and prevents mode collapse on test data.

2. *Empirical analysis.* We validate the CLIP-DR on seven publicly available datasets in: (i) robustness to different corruptions, (ii) generalization ability to domain shifts, (iii) computational overhead, (iv) robustness and (v) stability under small batch sizes.

## 2 RELATED WORK

**Test-time adaptation**. TTA adapts a pre-trained model (e.g., on ImageNet) to an incoming stream of unlabeled data processed in batches during testing Liang et al. (2025). In Sun et al. (2020), they propose the fundamentals of TTA and introduce self-supervised loss to adapt the model to the target data. However, this approach requires source data, which limits its potential. In Wang et al., they first formulate the full TTA task and design an entropy-based regularization technique to fine-tune the normalization layer parameters using the target test data. Furthermore, in Chen et al. (2022), they applied self-supervised contrastive learning between different augmented images with a pseudo-labeling technique to adapt the model at test time.

**Test-time adaptation with VLMs**. Test-Time Prompt Tuning (TPT) Shu et al. (2022) explored the usefulness of prompt tuning in TTA tasks using VLMs, such as CLIP. Specifically, it optimizes the prompt by minimizing the entropy with confidence selection to encourage the model to produce consistent predictions across different augmented views of each test sample. However, it requires multiple augmented samples, increasing memory usage. In Osowiechi et al. (2024), a diverse set of templates for text prompts was used with a text ensemble strategy to enhance text features by aggregating various textual cues (Weight average test-time adaption, WATT). Finally, it optimizes the normalization layer parameters with pseudo-labels. Similarly to WATT, in Hakim et al. (2025), they

design a novel process for tuning text prompts (CLIPArTT). Basically, multiple predicted classes are aggregated into a new text prompt (e.g., a picture of a "cat" or a "dog") that is used as a pseudo-label to reclassify the images transductively. By updating the normalization layer parameters, CLIPArTT provides feasible performance with lower overhead. Furthermore, in Lafon et al. (2025), they argue that previous gradient-based approaches for VLMs can degrade learned knowledge during TTA. Thus, they proposed a soft contrastive loss that aligns with CLIP's pretraining objective. In Maharana et al. (2025), they propose a bimodal online TTA method designed to improve the robustness of CLIP to common image corruptions. Specifically, they adapted the visual encoders and aligned the image and text features, promoting a stronger association between the image class prototype and the corresponding text feature generated by pseudo-labels.

Unlike previous works, we extend CLIP-OT by introducing cosine similarity based feature alignment and information maximization to regularize the predictions and reduce uncertainties. Specifically, it solves the over-confidence in OT framework, which causes high ECE. Our approach introduces negligible computational costs during adaptation, enhancing its practicality in a wide range of applications.

## 3 METHOD

### 3.1 PROBLEM SETTING

We address the problem of adapting a pretrained VLM at test time. In particular, given a model trained on the source domain $\mathcal{D}_S$ (e.g., ImageNet), the goal is to adapt the model online to the new target domain $\mathcal{D}_T$, where only unlabeled data are available (note that data is received as a stream of batches, and predictions must be provided).

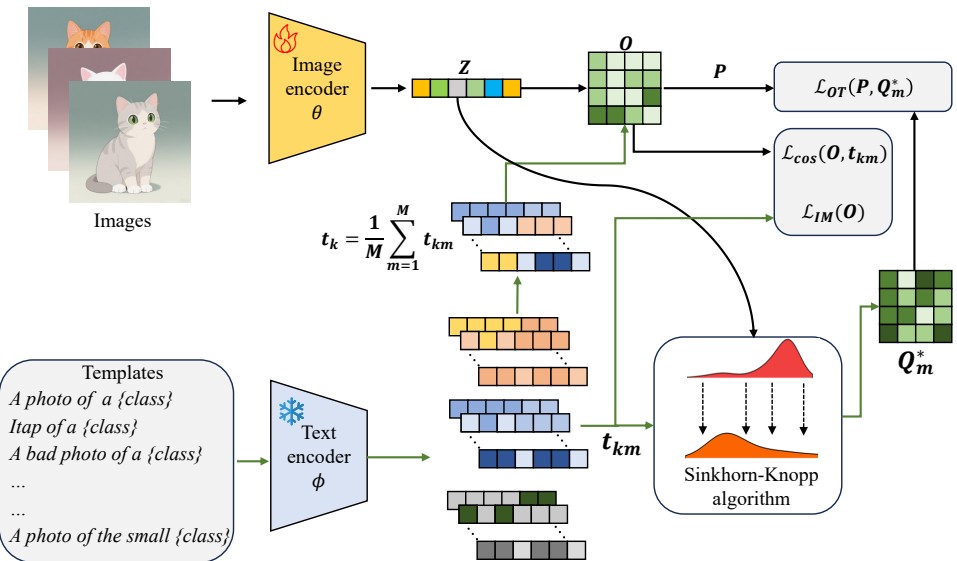

Figure 2: Pipeline of CLIP-DR. It uses optimal transport with the Sinkhorn algorithm to generate $\mathbf{Q}_m^*$, while use the original image and averaged text features to yield the probability matrix $\mathbf{P}$. Then, it minimizes the pseudo cross-entropy loss between $\mathbf{P}$ and $\mathbf{Q}_m^*$, the similarity loss between $\mathbf{O}$ and text prototypes $\mathbf{t}_{km}$, and the regularization loss $\mathcal{L}_{IM}$ using $\mathbf{Z}$ and $\mathbf{t}_{km}$ as unsupervised loss during TTA. Note that at each test batch, following Mishra et al. (2025), our model runs for $m$ iterations, each leveraging a different template $m$ to obtain the $\mathbf{Q}_m^*$.

Figure 2 shows the pipeline of our approach. It has two parts, 1) feature extraction, and 2) test-time adaptation. We will elaborate on these parts as follows.

## 3.2 FEATURE EXTRACTION WITH CLIP

CLIP is an image-text based VLM, which requires two encoders, namely an image encoder $\theta$ and a text encoder $\phi$. Basically, given the inputs $\mathbf{x}$ with corresponding prompts $\mathbf{t}$ (e.g., "A PHOTO OF A CLASS"), CLIP generates the vision and text features $\mathbf{z} \in \mathbb{R}^d$ and $\mathbf{t}_k \in \mathbb{R}^d$ normalized by $l_2$ distance, and provides the predicted probability as follows:

$$p(y = k|\mathbf{x}_i) = \frac{\exp\left(\mathbf{z}_i^\top \mathbf{t}_k / \tau\right)}{\sum_{j=1}^K \exp\left(\mathbf{z}_i^\top \mathbf{t}_j / \tau\right)}, \tag{1}$$

where $\tau$ is a pre-defined scaling parameter.

However, the original CLIP relies on a single prompt, that is, lacks diversity. Then, we introduce a popular strategy Radford et al. (2021) that designs multiple prompts to obtain the text features. Specifically, given a prompts set $\mathcal{T} = \{\{T_{km}\}_{k=1}^M\}_{m=1}^K$, whose embedding for template $m$ and class $k$ is calculated as $\mathbf{t}_{km} = \phi(\text{"prompts"})$. Finally, the $\mathbf{t}_k$ is obtained as $\mathbf{t}_k = \frac{1}{M}\sum_{m=1}^M \phi(T_{km})$ Details of the templates are reported in Appendix A.5.

## 3.3 TEST TIME ADAPTATION

**Optimal transport.**. Following Mishra et al. (2025), CLIP-DR optimizes the following loss functions:

$$\mathcal{L}(p, q) = -\frac{1}{N}\sum_{i=1}^N \sum_{c=1}^C q(y = k|\mathbf{x}_i) \log p(y = k|\mathbf{x}_i) \tag{2}$$

where $q(y = k|\mathbf{x}_i)$ is posterior distributions encoded using the model predictions.

Furthermore, since the probability is obtained using Eq. (1), the objective in Eq. (2) can be expressed as:

$$\mathcal{L}_{OT} = \mathcal{L}(p, q) = -\frac{1}{N}\sum_{i=1}^N \left[\frac{1}{\tau}\mathbf{z}_i^\top \mathbf{T}\mathbf{q}_i - \log\sum_{k=1}^K \exp\left(\frac{\mathbf{z}_i^\top \mathbf{t}_k}{\tau}\right)\right], \tag{3}$$

where $\mathbf{T} = [\mathbf{t}_1, \cdots, \mathbf{t}_K]$ represents the matrix consisting of class text prototypes. In practical, it can be reformulated as the following objective:

$$\max_{\mathbf{Q} \in \mathcal{Q}} tr(\mathbf{Q}^\top \mathbf{T}^\top \mathbf{Z}), \tag{4}$$

where the $\mathbf{Q}$ should be an element of the transportation polytope:

$$\mathcal{Q} := \left\{\mathbf{Q} \in \mathbb{R}^{K \times N} \mid \mathbf{Q}\mathbf{1}_N = \frac{1}{K}\mathbf{1}_K, \mathbf{Q}^\top\mathbf{1}_K = \frac{1}{N}\mathbf{1}_N\right\}, \tag{5}$$

where $\mathbf{1}_K$ and $\mathbf{1}_N$ represent the vectors of ones in dimension $K$ and $N$, respectively.

To efficiently solve the previous objective function, the Sinkhorn algorithm is used to reformulate the Eq. (4) as follows:

$$\max_{\mathbf{Q} \in \mathcal{Q}} tr(\mathbf{Q}^\top \mathbf{T}^\top \mathbf{Z}) + \epsilon\mathcal{H}(\mathbf{Q}) \tag{6}$$

where $\mathcal{H}$ is a Shannon entropy function. Finally, we can obtain $\mathbf{Q}^*$ with a few iterations as:

$$\mathbf{Q}^\star = \text{Diag}(\mathbf{u}^{(t)})\exp\left(\frac{\mathbf{T}^\top\mathbf{Z}}{\tau}\right)\text{Diag}(\mathbf{v}^{(t)}), \tag{7}$$

where $\mathbf{u}$ and $\mathbf{v}$ are renormalization vectors in $\mathbb{R}^K$ and $\mathbb{R}^B$ respectively, with $t$ indicating the iteration. Specifically, $\mathbf{u}$ and $\mathbf{v}$ can be obtained using the iterative Sinkhorn-Knopp algorithm Knight (2008). Since we use multiple templates, in each batches of data, it will iterate $m$ times for each template $m$ as follows:

$$\mathbf{Q}_m^\star = \text{Diag}(\mathbf{u}_m^{(t)})\exp\left(\frac{\mathbf{T}_m^\top\mathbf{Z}}{\tau}\right)\text{Diag}(\mathbf{v}_m^{(t)}) \tag{8}$$

where for each $\mathbf{Q}_m^*$, the loss is calculated as $\ell(\mathbf{P}, \mathbf{Q}_m^*) = -\mathbf{Q}_m^*\log\mathbf{P}$.

**Logits alignment**. To maintain CLIP's cross-modal alignment during adaptation, we regularize the output logits $\mathbf{O}$ obtained from CLIP to align with the text prototypes. Specifically, we construct virtual logits $\bar{\mathbf{f}}_{\hat{y}}$ from the text prototypes to serve as alignment targets. Following Choi et al. (2025), we enforce this alignment using cosine similarity:

$$s_{\mathbf{O},\bar{\mathbf{f}}_{\hat{y}}} = \frac{\langle \mathbf{O}, \bar{\mathbf{f}}_{\hat{y}} \rangle}{\|\mathbf{O}\| \cdot \|\bar{\mathbf{f}}_{\hat{y}}\|} \text{ , where } \bar{\mathbf{f}}_{\hat{y}} = \mathbf{t}_{km}^{\top} \mathbf{t}_{km}^{\hat{y}} \text{ , } \mathbf{O} = \mathbf{z}_i^{\top} \mathbf{t}_k / \tau \tag{9}$$

where the label $\hat{y}$ corresponds to the most confident class for target sample $\mathbf{x}_j$. The final loss can be formulated as follows:

$$\mathcal{L}_{cos} = \sum_{j=1}^{B} \exp(C(\mathbf{x}_j) - C_0) \cdot (1 - s_{\mathbf{O},\bar{\mathbf{f}}_{\hat{y}}}) \cdot \mathbb{I}\{C(\mathbf{x}_j) \geq C_0\} \tag{10}$$

where $\mathbb{I}$ is an indicator function. $C(\mathbf{x})$ is the predictive confidence of sample $\mathbf{x}$, and the $C_0$ is the confidence threshold to filter prediction uncertainties. A lower value of $\mathcal{L}_{cos}$ indicates that the output $\mathbf{O}$ is similar to the virtual logits $\bar{\mathbf{f}}_{\hat{y}}$.

**Information maximization**. To prevent posterior collapse and ensure meaningful feature learning, we introduce a regularization term using an Information Maximization (IM) function Tschannen et al. (2020). Specifically, we minimize the entropy of individual predictions to ensure the model produces confident, sharp outputs on the unlabeled test data, while we maximize the entropy of the average prediction distribution across the entire test samples to force the model to produce diverse predictions as follows:

$$\mathcal{L}_{\text{IM}} = \frac{1}{|\mathcal{B}_t|} \sum_{\mathbf{x} \in \mathcal{B}_t} H(p(\mathbf{x})) - H(\bar{p}) \tag{11}$$

where $H(p(\mathbf{x})) = -\sum_{k=1}^{C} p_k(\mathbf{x}) \log p_k(\mathbf{x})$ is the entropy of the prediction distribution for a single sample $\mathbf{x}$, $\bar{p} = \frac{1}{|\mathcal{B}_t|} \sum_{\mathbf{x} \in \mathcal{B}_t} p(\mathbf{x})$ is the average prediction distribution over the test samples in each batch, and $H(\bar{p}) = -\sum_{k=1}^{C} \bar{p}_k \log \bar{p}_k$ is the entropy of the average prediction distribution.

Finally, we optimize the following loss function:

$$\mathcal{L} = \mathcal{L}_{OT} + \lambda_1 \mathcal{L}_{cos} + \lambda_2 \mathcal{L}_{IM} \tag{12}$$

where $\lambda_1$ and $\lambda_2$ are two hyper-parameters. Following Mishra et al. (2025), we optimize the Layernorm layer while freezing the other layers to improve computational efficiency.

Typically, over-confidence (i.e., high ECE) arises from noisy pseudo-labels and collapsed predictions. During adaptation, $\mathcal{L}_{cos}$ reduces pseudo-label noise by aligning features with correct prototypes, minimizing confidently wrong predictions. Furthermore, $\mathcal{L}_{IM}$ directly penalizes overconfidence by enforcing diverse outputs via entropy maximization while maintaining per-sample confidence, which is a regularization for calibration Patel et al. (2020). Together, they jointly optimize the model to avoid feature misalignment and distribution collapse, providing promising calibration ability.

# 4 EXPERIMENTS

## 4.1 DATASETS

CLIP-DR is evaluated on three families of adaptation benchmarks, namely no corruptions (CIFAR-10 Krizhevsky et al. (2009) and TinyImageNet Le & Yang (2015)), corruptions (CIFAR-10/100-C Hendrycks & Dietterich (2019), ImageNet-C Hendrycks & Dietterich (2019), TinyImageNet-C Hendrycks & Dietterich (2019) with 15 perturbations), and domain shifts (PACS Li et al. (2017) and OfficeHome Venkateswara et al. (2017)). In this study, we consider the same severity level as used in CLIP-OT.

## 4.2 IMPLEMENTATION DETAILS

Pretrained CLIP (ViT-B/32) Radford et al. (2021) is used as the feature extractor backbone. The batchsize is set to 128 with a learning rate of $10^{-4}$ for an Adam optimizer. The $\tau$ is set to 0.01. The

Table 1: Accuracy (%) of the different approaches on CIFAR Corruptions benchmarks. **Bold** means the best result. Δ represents the difference between Ours and CLIP-OT.

| | Dataset | CLIP _ICLR'21_ | TENT _ICLR'21_ | TPT _NeurIPS'22_ | CLIPArTT _WACV'25_ | WATT-P _NeurIPS'24_ | WATT-S _NeurIPS'24_ | CLIP-OT _Arxiv'25_ | CLIP-DR _Ours_ | Δ |
|---|---|---|---|---|---|---|---|---|---|---|
| | **CIFAR-10** | 88.74 | 91.69 ±0.10 | 88.06 ±0.06 | 90.04 ±0.13 | 91.41 ±0.17 | 91.05 ±0.06 | 93.23 ±0.08 | **93.43±0.06** | +0.2 |
| CIFAR-10C | Gaussian Noise | 35.27 | 41.27 ±0.27 | 33.90 ±0.08 | 59.90 ±0.36 | 61.89 ±0.24 | 63.84 ±0.24 | 65.03 ±0.16 | **65.25±0.13** | +0.22 |
| | Shot Noise | 39.67 | 47.20 ±0.23 | 38.20 ±0.02 | 62.77 ±0.07 | 63.52 ±0.08 | 65.28 ±0.21 | **67.43 ±0.19** | 67.38±0.15 | -0.05 |
| | Impulse Noise | 42.61 | 48.58 ±0.31 | 37.66 ±0.08 | 56.02 ±0.16 | 57.13 ±0.02 | 58.64 ±0.11 | 62.15 ±0.36 | **62.24±0.23** | +0.09 |
| | Defocus Blur | 69.76 | 77.12 ±0.16 | 67.83 ±0.28 | 76.74 ±0.05 | 78.86 ±0.09 | 78.94 ±0.12 | 82.26 ±0.09 | **82.4±0.13** | +0.14 |
| | Glass Blur | 42.40 | 52.65 ±0.30 | 38.81 ±0.12 | 61.77 ±0.16 | 62.88 ±0.06 | 65.12 ±0.07 | **67.86 ±0.02** | 67.69±0.04 | -0.17 |
| | Motion Blur | 63.97 | 71.25 ±0.09 | 63.39 ±0.13 | 76.01 ±0.14 | 76.85 ±0.26 | 77.81 ±0.14 | 81.4 ±0.15 | **81.64±0.19** | +0.24 |
| | Zoom Blur | 69.83 | 76.20 ±0.19 | 68.95 ±0.16 | 77.40 ±0.20 | 79.35 ±0.04 | 79.32 ±0.07 | 83.14 ±0.19 | **83.6±23** | +0.46 |
| | Snow | 71.78 | 78.29 ±0.20 | 70.16 ±0.10 | 77.29 ±0.16 | 79.44 ±0.09 | 79.79 ±0.06 | 83.61 ±0.14 | **84.1±14** | +0.49 |
| | Frost | 72.86 | 79.84 ±0.09 | 72.39 ±0.22 | 79.20 ±0.08 | 80.13 ±0.10 | 80.54 ±0.12 | 83.24 ±0.22 | **83.45±0.18** | +0.21 |
| | Fog | 67.04 | 77.39 ±0.01 | 64.31 ±0.28 | 75.74 ±0.14 | 77.68 ±0.07 | 78.53 ±0.22 | 82.29 ±0.12 | **82.38±0.08** | +0.09 |
| | Brightness | 81.87 | 87.78 ±0.03 | 81.30 ±0.18 | 86.59 ±0.16 | 87.10 ±0.10 | 87.11 ±0.11 | 89.64 ±0.10 | **89.75±0.11** | +0.11 |
| | Contrast | 64.37 | 79.47 ±0.11 | 62.26 ±0.31 | 77.82 ±0.14 | 80.04 ±0.24 | 81.20 ±0.22 | 84.75 ±0.13 | **84.98±0.12** | +0.23 |
| | Elastic Transform | 60.83 | 70.00 ±0.25 | 56.43 ±0.27 | 70.20 ±0.01 | 71.76 ±0.10 | 72.66 ±0.15 | 75.98 ±0.22 | **76.16±0.17** | +0.18 |
| | Pixelate | 50.53 | 63.74 ±0.18 | 42.80 ±0.40 | 66.52 ±0.13 | 69.28 ±0.09 | 71.11 ±0.13 | 76.64 ±0.06 | **77.11±0.04** | +0.47 |
| | JPEG Compression | 55.48 | 62.64 ±0.14 | 53.67 ±0.25 | 63.51 ±0.14 | 66.49 ±0.14 | 67.36 ±0.28 | 70.18 ±0.31 | **70.5±0.27** | +0.32 |
| | **Average** | 59.22 | 67.56 | 56.80 | 71.17 | 72.83 | 73.82 | 77.04 | **77.24** | +0.2 |
| CIFAR-100C | Gaussian Noise | 14.80 | 14.38±0.14 | 14.03±0.10 | 25.32±0.14 | 31.28±0.03 | 32.07±0.23 | 33.19±0.11 | **33.55±0.13** | +0.36 |
| | Shot Noise | 16.03 | 17.34±0.27 | 15.25±0.17 | 27.90±0.05 | 33.44±0.11 | 34.36±0.11 | 34.75±0.13 | **35.23±0.15** | +0.48 |
| | Impulse Noise | 13.85 | 10.03±0.13 | 13.01±0.13 | 25.62±0.09 | 29.40±0.11 | 30.33±0.03 | 30.49±0.33 | **30.61±0.21** | +0.12 |
| | Defocus Blur | 36.74 | 49.05±0.07 | 37.60±0.17 | 49.88±0.23 | 52.32±0.28 | 52.99±0.16 | 53.50±0.25 | **53.77±0.19** | +0.27 |
| | Glass Blur | 14.19 | 3.71±0.07 | 16.41±0.02 | 27.89±0.03 | 31.20±0.12 | 32.15±0.30 | 34.81±0.24 | **35.3±0.26** | +0.49 |
| | Motion Blur | 36.14 | 46.62±0.27 | 37.52±0.23 | 47.93±0.14 | 49.72±0.15 | 50.53±0.12 | 52.70±0.05 | **52.81±0.07** | +0.11 |
| | Zoom Blur | 40.24 | 51.84±0.15 | 42.99±0.11 | 52.70±0.06 | 54.72±0.04 | 55.30±0.22 | 56.73±0.15 | **56.83±0.16** | +0.1 |
| | Snow | 38.95 | 46.71±0.21 | 42.35±0.13 | 49.72±0.01 | 51.79±0.04 | 52.77±0.15 | 53.83±0.12 | **54.24±0.11** | +0.41 |
| | Frost | 40.56 | 44.90±0.27 | 43.31±0.14 | 49.63±0.12 | 53.04±0.08 | 53.79±0.31 | 54.63±0.05 | **54.91±0.08** | +0.28 |
| | Fog | 38.00 | 47.31±0.04 | 38.81±0.17 | 48.77±0.04 | 50.78±0.24 | 51.49±0.21 | 53.53±0.03 | **53.87±0.08** | +0.34 |
| | Brightness | 48.18 | 60.58±0.18 | 50.23±0.11 | 61.27±0.08 | 62.65±0.25 | 63.57±0.21 | 64.07±0.29 | **64.47±0.21** | +0.4 |
| | Contrast | 29.53 | 45.90±0.11 | 28.09±0.09 | 48.55±0.24 | 51.34±0.10 | 52.76±0.27 | 54.75±0.32 | **54.87±0.28** | +0.12 |
| | Elastic Transform | 26.33 | 33.09±0.08 | 28.12±0.15 | 37.45±0.08 | 39.97±0.06 | 40.90±0.43 | 43.06±0.21 | **43.47±0.17** | +0.41 |
| | Pixelate | 21.98 | 26.47±0.09 | 20.43±0.14 | 33.88±0.14 | 39.59±0.09 | 40.97±0.16 | 44.68±0.28 | **45.31±0.15** | +0.68 |
| | JPEG Compression | 25.91 | 29.89±0.07 | 28.82±0.09 | 36.07±0.32 | 38.99±0.16 | 39.59±0.08 | 40.66±0.17 | **41.07±0.11** | +0.41 |
| | **Average** | 29.43 | 35.19 | 30.46 | 41.51 | 44.68 | 45.57 | 47.02 | **47.35** | +0.33 |

Table 2: Test accuracy (%) on Tiny-ImageNet-C corruption benchmarks. **Bold** means the best result. Δ represents the difference between CLIP-DR and CLIP-OT.

| Dataset | CLIP _ICLR'21_ | TENT _ICLR'21_ | TPT _NeurIPS'22_ | CLIPTT _WACV'25_ | WATT-S _NeurIPS'24_ | TDA _CVPR'24_ | CLIP-OT _Arxiv'25_ | CLIP-DR _Ours_ | Δ |
|---|---|---|---|---|---|---|---|---|---|
| Tiny-ImageNet | 58.29 | 57.72 | 58.90 | 59.85 | 61.35 | 62.36 | 63.69 | **64.38** | +0.69 |
| Gaussian Noise | 7.08 | 8.01 | 9.29 | 14.44 | 13.02 | 24.01 | 21.36 | **24.76** | +3.4 |
| Shot Noise | 9.41 | 10.04 | 11.70 | 17.44 | 15.94 | 28.21 | 24.51 | **32.53** | +7.63 |
| Impulse Noise | 3.44 | 4.18 | 4.85 | 10.37 | 6.90 | 18.65 | 18.21 | **24.68** | +7.34 |
| Defocus Blur | 21.71 | 24.53 | 27.56 | 31.46 | 29.91 | 36.77 | 36.25 | **44.42** | +8.17 |
| Glass Blur | 9.12 | 10.09 | 11.03 | 15.84 | 14.01 | 22.58 | 23.08 | **29.24** | +6.16 |
| Motion Blur | 34.52 | 36.94 | 38.97 | 41.34 | 41.26 | 37.38 | 47.78 | **49.45** | +1.67 |
| Zoom Blur | 27.44 | 29.48 | 34.29 | 35.06 | 33.96 | 36.59 | 40.96 | **44.78** | +3.82 |
| Snow | 32.51 | 32.20 | 34.45 | 36.86 | 37.76 | 33.23 | 41.28 | **43.85** | +2.57 |
| Frost | 36.33 | 35.72 | 37.13 | 38.20 | 39.65 | 35.94 | 46.03 | **47.66** | +1.63 |
| Fog | 25.94 | 27.46 | 28.89 | 33.44 | 32.13 | 38.17 | 39.93 | **49.31** | +9.38 |
| Brightness | 40.15 | 39.79 | 43.31 | 46.43 | 46.93 | 42.98 | 53.81 | **57.9** | +4.09 |
| Contrast | 1.81 | 2.24 | 3.15 | 6.24 | 3.53 | 14.70 | 12.94 | **18.21** | +5.27 |
| Elastic Transf. | 30.40 | 31.92 | 33.88 | 33.89 | 35.01 | 42.38 | 41.53 | **49.5** | +7.97 |
| Pixelate | 22.78 | 24.79 | 27.70 | 34.85 | 31.55 | 34.97 | 42.90 | **47.54** | +4.64 |
| JPEG Compr. | 29.59 | 30.93 | 33.60 | 37.32 | 36.46 | 40.64 | 42.88 | **50.93** | +8.05 |
| **Mean** | 22.14 | 23.22 | 25.32 | 28.88 | 27.87 | 32.48 | 35.56 | **40.98** | +5.16 |

Table 3: Detailed performance values for PACS and OfficeHome datasets. Δ represents the difference between Ours and the CLIP-OT. **Bold** indicates the best result.

| Dataset | Domain | CLIP _ICLR'21_ | TENT _ICLR'21_ | TPT _NeurIPS'22_ | CLIPArTT _WACV'25_ | WATT-P _NeurIPS'24_ | WATT-S _NeurIPS'24_ | CLIP-OT _Arxiv'25_ | CLIP-DR _Ours_ | Δ |
|---|---|---|---|---|---|---|---|---|---|---|
| **PACS** | Art | 96.34 | 96.65±0.05 | 95.52±0.20 | 96.57±0.09 | 96.31±0.01 | 96.39±0.01 | 96.54±0.02 | **97.07±0.01** | +0.53 |
| | Cartoon | 96.08 | 96.22±0.05 | 94.77±0.20 | 96.00±0.02 | 96.52±0.02 | 96.62±0.02 | 97.66±0.00 | **98.00±0.00** | +0.34 |
| | Photo | 99.34 | 99.40±0.00 | 99.42±0.06 | 99.28±0.06 | 99.48±0.03 | 99.52±0.00 | **99.76±0.03** | **99.76±0.03** | - |
| | Sketch | 82.85 | 82.96±0.12 | 83.22±0.14 | 83.93±0.14 | 86.92±0.04 | 86.65±0.12 | 86.54±0.01 | **87.14±0.01** | +0.6 |
| | **Mean** | 93.65 | 93.81 | 93.23 | 93.95 | 94.81 | 94.80 | 95.13 | **95.5** | +0.37 |
| **OfficeHome** | Art | 73.75 | 74.03±0.27 | 75.76±0.27 | 73.84±0.20 | 75.65±0.27 | 75.76±0.39 | 78.78 ±0.15 | **79.38 ±0.22** | +0.6 |
| | Clipart | 63.33 | 63.42±0.04 | 63.08±0.31 | 63.54±0.06 | 66.23±0.13 | 65.77±0.11 | 66.21±0.19 | **67.74±0.12** | +1.53 |
| | Product | 85.32 | 85.51±0.08 | 84.07±0.28 | 85.23±0.16 | 85.41±0.09 | 85.41±0.01 | 86.13±0.03 | **86.25±0.03** | +0.12 |
| | Real World | 87.71 | 87.74±0.05 | 85.89±0.33 | 87.61±0.05 | 88.22±0.15 | **88.37±0.07** | 87.79±0.19 | 87.96±0.17 | +0.17 |
| | **Mean** | 77.53 | 77.68 | 77.20 | 77.56 | 78.88 | 78.83 | 78.98 | **80.33** | +1.35 |

Table 4: Test accuracy (%) on Tiny-ImageNet-C dataset with different components. **Bold** indicates the best result.

| Components | | | Noises | | | | | | | | | | | | | | | |
|---|---|---|---|---|---|---|---|---|---|---|---|---|---|---|---|---|---|---|
| $\mathcal{L}_{OT}$ | $\mathcal{L}_{IM}$ | $\mathcal{L}_{cos}$ | Gaussian | Shot | Impulse | Defocus | Glass | Motion | Zoom | Snow | Frost | Fog | Brightness | Contrast | Elastic | Pixelate | JPEG | Avg. |
| ✓ | ✗ | ✗ | 21.36 | 24.51 | 18.21 | 36.25 | 23.08 | 47.78 | 40.96 | 41.28 | 46.03 | 39.93 | 53.81 | 12.94 | 41.53 | 42.90 | 42.88 | 35.56 |
| ✓ | ✓ | ✗ | 23.75 | 29.7 | 22.12 | 40.01 | 27.55 | 49.42 | 42.87 | 43.47 | 47.21 | 47.79 | 57.59 | 15.71 | 46.26 | 47.02 | 49.92 | 39.36 |
| ✓ | ✗ | ✓ | 24.57 | 30.36 | 22.36 | 40.71 | 27.48 | 49.09 | 41.72 | 43.33 | 47.33 | 48.13 | 57.16 | 16.06 | 45.26 | 46.41 | 49.89 | 39.32 |
| ✓ | ✓ | ✓ | **24.76** | **32.53** | **24.68** | **44.42** | **29.24** | **49.45** | **44.78** | **43.85** | **47.66** | **49.31** | **57.9** | **18.21** | **49.5** | **47.54** | **50.93** | **40.98** |

$\epsilon$ used in Eq. 6 is set to 0.7 and 0.5 for CIFAR-10/100-C and TinyImageNet-C datasets, respectively. Furthermore, the values of $\lambda_1$ and $\lambda_2$ are set to 0.1 and 0.3 for CIFAR-10-C, 0.1 and 0.1 for CIFAR-100-C, and 1.0 and 0.1 for TinyImageNet-C, respectively. All experiments are based on the Windows 11 operating system, and feature an Intel 13900KF CPU with 128 GB of RAM and an RTX 4090 GPU. We use PyTorch 1.13.1 with Python 3.8. Furthermore, eight predefined text templates from CLIP were used to evaluate the proposed model's adaptability and performance. Note that the templates are the same as the ones employed in WATT Osowiechi et al. (2024). For comparison, we consider CLIP-based TTA approaches, including TENT Wang et al., TPT Shu et al. (2022), CLIPArTT Hakim et al. (2025), TDA Karmanov et al. (2024), WATT Osowiechi et al. (2024) and CLIP-OT Mishra et al. (2025). Following the CLIP-OT Mishra et al. (2025), we report the results on these datasets as the average ± standard deviation (three times under different random seeds). Baseline results (excluding CLIP-OT and TDA) are from Mishra et al. (2025); we reran CLIP-OT and TDA using their official code with optimal settings they provided.

### 4.3 RESULTS

**No corruptions**. As reported in Table 1 and Table 2, CLIP-DR achieves higher test accuracy than CLIP-OT, e.g., a 0.69% improvement on TinyImageNet. These results suggest that CLIP-DR effectively adapts the features without compromising the performance of the model on its original clean data.

**Performance under common corruptions**. Table 1 and Table 2 report the test accuracy for CLIP-DR and baselines on CIFAR-10, CIFAR-10/100-C and TinyImageNet-C datasets. Notably, CLIP-DR achieves a substantial improvement of 4.82% in Avg. accuracy compared to CLIP-OT on the TinyImageNet-C dataset. Across its 15 corruptions, CLIP-DR shows consistent gains, with improvements ranging from 1.63% to 9.38%. On the CIFAR benchmarks, CLIP-DR provides modest performance gains (e.g., +0.33% Avg. accuracy on CIFAR-100-C). These results suggest that CLIP-DR enhances the performance on both simple and complex data while maintains a stable adaptation process. Furthermore, Table 5 summarizes the performance on ImageNet-C. Results show that CLIP-OT suffers from negative adaptation, degrading the performance from 37.89% (CLIP zero-shot) to 35.45%. This demonstrates that the pseudo-labels generated by OT can be highly unreliable. CLIP-DR reduces performance degradation and provides a higher Avg. accuracy of 39.03%, outperforming CLIP-OT by 3.58%. This proves that our dual regularization framework is essential for stabilizing the adaptation process and preventing error accumulation. The results of the TinyImageNet-C dataset with standard deviation are summarized in Appendix Table 30.

**Texture and style shifts**. Table 3 presents the test accuracy for the PACS and OfficeHome datasets. For example, CLIP-DR achieves an Avg. accuracy of 80.33%, outperforming CLIP-OT by 1.35% on OfficeHome. Overall, CLIP-DR provides the highest Avg. accuracy across all datasets, highlighting its adaptability to texture and style shifts.

Table 5: Accuracy (%) on ImageNet-C for CLIP-DR and baselines. **Bold** indicates the best result.

| Methods | Noises | | | | | | | | | | | | | | | |
|---|---|---|---|---|---|---|---|---|---|---|---|---|---|---|---|---|
| ImageNet-C | Gaussian | Shot | Impulse | Defocus | Glass | Motion | Zoom | Snow | Frost | Fog | Brightness | Contrast | Elastic | Pixelate | JPEG | Avg. |
| CLIP | 35.4 | 35.75 | 35.96 | 35.59 | 19.45 | 36.63 | 26.88 | 31.35 | 30.04 | **43.99** | 53.38 | **46.95** | **46.59** | 44.42 | **46.01** | 37.89 |
| CLIP-OT | 31.56 | 32.08 | 33 | 32.5 | 19.65 | 31.73 | 27.65 | 30.53 | 28.94 | 40.29 | 52.51 | 43.22 | 44.15 | 40.73 | 43.29 | 35.45 |
| CLIP-DR | **37.51** | **37.75** | **37.71** | **36.73** | **25.1** | **38.22** | **30.98** | **33.23** | **30.42** | 43.5 | 52.97 | 46.36 | 45.69 | 44.02 | 45.35 | **39.03** |

**Computation overhead**. Figure 3 compares the per-batch inference time of the baselines and CLIP-DR using a batch size of 128. As shown, CLIP-DR requires an extra 0.01 and 0.02 seconds on Tiny-ImageNet-C and OfficeHome datasets compared to CLIP-OT. This minimal overhead demonstrates that the proposed components can be integrated into the TTA with negligible impact on throughput, making CLIP-DR suitable for practical situations.

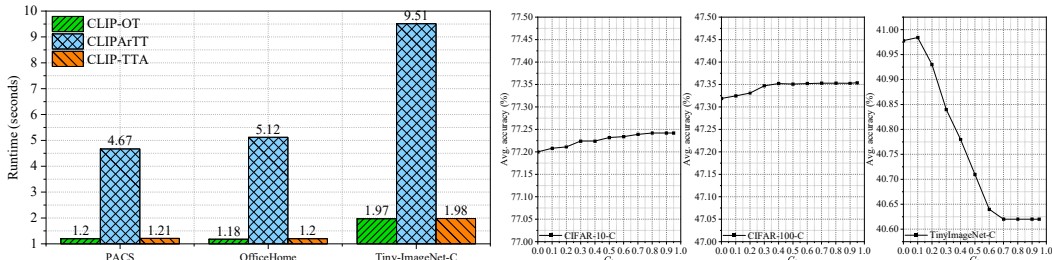

Figure 3: Runtime (**Left**) of recent TTA approaches and CLIP-DR for each batch of data and Avg. accuracy with different $C_0$ (**Right**) values on PACS, OfficeHome and Tiny-ImageNet-C datasets.

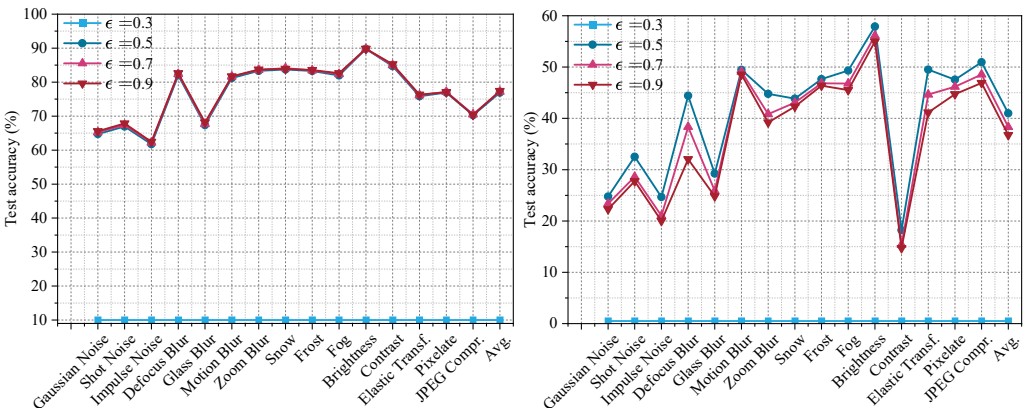

Figure 4: Test accuracy (%) on CIFAR-10-C (**Left**) and Tiny-ImageNet-C (**Right**) corruption benchmarks with different $\epsilon$ values.

### 4.4 MODEL ANALYSIS

**Ablation study**. Table 4 reports the test accuracy on Tiny-ImageNet-C with different components. With $\mathcal{L}_{cos}$, CLIP-DR learns more robust features, provides a higher accuracy compared to CLIPOT (e.g., 24.57% vs. 21.4% under Gaussian noise), while the use of both $\mathcal{L}_{cos}$ and $\mathcal{L}_{IM}$ leads to the highest accuracy (e.g., 24.76%). These results highlight the usefulness of the proposed components.

**Impact of $\epsilon$**. We validate the usefulness of the hyper-parameter $\epsilon$ using CIFAR-10-C and Tiny-ImageNet-C datasets. As shown in Figure 4, a lower $\epsilon$ value such as 0.3 can not guarantee the model learn useful feature representations, thereby provides 0.5% and 10% Avg. accuracy on TinyImageNet-C and CIFAR-10-C datasets, respectively. This is because in certain cases, the **Q** matrix became NaN, causing numerical instabilities during iterative updates. Instead, a higher $\epsilon$ value, such as 0.5 improves stability, leading to the highest Avg. accuracy of 40.98% on TinyImageNet-C. Overall, the chosen values (0.5, 0.7) strike a balance between stability and performance, yielding robust results across these noises.

**Impact of $C_0$**. We use different threshold ($C_0$) to validate its importance in $\mathcal{L}_{cos}$. Specifically, we consider $C_0 \in \{0.1, 0.2, \cdots, 0.95\}$ for experiments using CIFAR-10-C, CIFAR-100-C and TinyImageNet-C datasets. As shown in Figure 3, setting a higher $C_0$ leads to a higher Avg. accuracy compared to low $C_0$ on CIFAR-10-C (e.g., 77.24%) and CIFAR-100-C (e.g., 47.35%) datasets, while using a small $C_0$ provides higher Avg. accuracy on TinyImageNet-C dataset (e.g., 40.98%). We hypothesize that on simpler datasets such as CIFAR-10-C, a high threshold effectively filters out true uncertainties. On complex datasets, the model can be overconfident in its incorrect predictions, requiring a lower threshold.

**Hyperparameter sensitivity**. We explore the impact of each hyperparameter on the CIFAR-10-C, CIFAR-100-C, and Tiny-ImageNet-C datasets. Specifically, we consider $\lambda_1$ using values ranging from [0.1, 0.2, ..., 1.0] for Tiny-ImageNet-C, CIFAR-10-C and CIFAR-100-C. We set the values of $\lambda_2$ ranging from [0.1, 0.2, 0.3, ..., 1.0] (Tiny-ImageNet-C, CIFAR-10-C and CIFAR-100-C). Figure

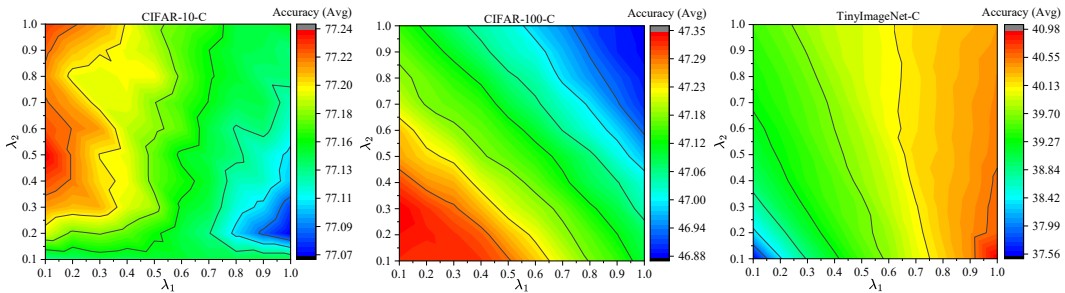

Figure 5: Heatmaps of the average accuracy with respect to $\lambda_1$ and $\lambda_2$.

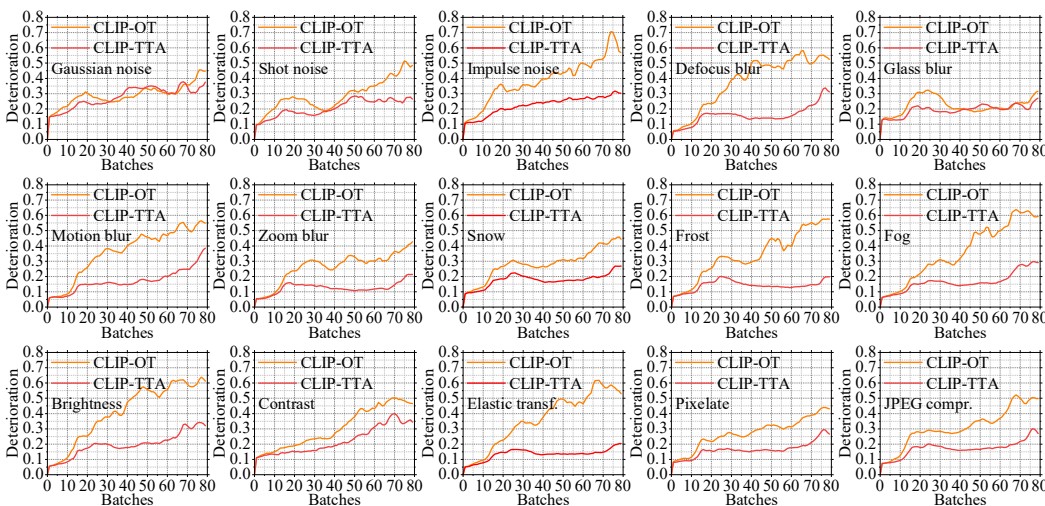

Figure 6: The deterioration ratio for CLIP-OT and CLIP-DR on TinyImageNet-C dataset.

5 shows the sensitivity of the Avg. accuracy to the hyperparameters $\lambda_1$ and $\lambda_2$. For example, the heatmap for Tiny-ImageNet-C shows an optimal performance region (i.e., red) where $\lambda_1$ is approximately in [0.8, 1.0] and $\lambda_2$ in [0.1, 0.5]. The highest Avg. accuracy of 40.98% is achieved at a value of 1.0 for $\lambda_1$ and 0.1 for $\lambda_2$. Furthermore, the performance of the model depends strongly on $\lambda_1$, with Avg. accuracy consistently higher than 40% when $\lambda_1 \geq 0.8$, while the impact of $\lambda_2$ is minimal (i.e., the accuracy changes are $< 0.5\%$). However, for CIFAR-10-C, achieving optimal performance requires a higher value of $\lambda_2$ and a lower value of $\lambda_1$. For CIFAR-100-C, setting smaller values for both hyperparameters provides higher Avg. accuracy (e.g., $\lambda_1 = 0.1$, $\lambda_2 = 0.3$, with an Avg. of 47.35%). Our results indicate that the optimal hyperparameter region is dataset-dependent. A simple tuning strategy can be drawn from these findings: prioritize tuning $\lambda_1$ on large-scale datasets, while for smaller datasets, careful tuning of $\lambda_2$ is essential. The details of the test accuracy for each corruption are reported in Appendix A.6.1.

**Robustness**. Figure 6 shows the deterioration ratio (i.e., the fraction of initially correct predictions that become incorrect during adaptation) on TinyImageNet-C dataset based on a single run (seed 42). For example, CLIP-OT exhibits a high deterioration ratio on many corruptions, such as Shot noise ($\sim$0.5) and Defocus blur ($\sim$0.6), indicating severe catastrophic forgetting. However, CLIP-DR reduces these ratios to $\sim$0.3 on the same corruptions. Overall, CLIP-DR yields a lower deterioration ratio under all 15 corruptions, highlighting its stability effect during adaptation.

**Small Batch Sizes**. Recent studies Wang et al. (2025); Karmanov et al. (2024) suggest one challenging TTA setting, which uses a small batchsize (e.g., four). Table 6 reports the test accuracy on CIFAR-10-C and TinyImageNet-C datasets with a batchsize of four. For CIFAR-10-C dataset, the small batch size limits the potential of CLIP-OT, reducing its Avg. accuracy to 57.01%. We notice that CLIP-OT yields a lower Avg. accuracy compared to CLIP (59.22%). Similar conclusion can

be drawn from TinyImageNet-C dataset. This indicates that using OT alone can lead to negative adaptation under highly constrained conditions. In contrast, CLIP-DR, with its dual regularization components, achieves an Avg. accuracy of 64.21% on CIFAR-10-C dataset, representing a 7.2% improvement over CLIP-OT and a 4.99% improvement over the CLIP. This highlights the robustness of our approach in practical low-batch-size settings.

Table 6: Performance under different batchsize for CLIP-DR and CLIP-OT. **Bold** indicates the best result.

| Methods | Noises | | | | | | | | | | | | | | | |
|---|---|---|---|---|---|---|---|---|---|---|---|---|---|---|---|---|
| CIFAR-10-C | Gaussian | Shot | Impulse | Defocus | Glass | Motion | Zoom | Snow | Frost | Fog | Brightness | Contrast | Elastic | Pixelate | JPEG | Avg. |
| CLIP-OT | 49.93 | 51.16 | 48.2 | 59.54 | 51.53 | 60.27 | 61.7 | 61.68 | 65.9 | 59.35 | 64.11 | 58.32 | 58.32 | 50.06 | 55.13 | 57.01 |
| CLIP-DR | **55.65** | **56.71** | **52.72** | **69.9** | **57.83** | **68.78** | **71.27** | **69.7** | **71.61** | **66.58** | **73.29** | **66.95** | **64.21** | **58.62** | **59.35** | **64.21** |
| TinyImageNet-C | | | | | | | | | | | | | | | | |
| CLIP-OT | 12.1 | 15.39 | 11.92 | 17.17 | 14.69 | 18.7 | 17.83 | 19 | 20.44 | 20.62 | 23.98 | 7.11 | 21.94 | 21.71 | 24.16 | 17.78 |
| CLIP-DR | **18.81** | **25.17** | **18.56** | **30.67** | **20.73** | **33.47** | **32.13** | **27.64** | **31.21** | **36.74** | **37.42** | **12.07** | **38.18** | **29.42** | **37.76** | **28.66** |

**Cross-dataset**. We use several cross-datasets to explore the generalization on different data types. Specifically, ImageNet-R Hendrycks et al. (2021), ImageNetV2 Recht et al. (2019), Aircraft Maji et al. (2013), Caltech-101 Bansal et al. (2023), EuroSAT Helber et al. (2019), DTD Cimpoi et al. (2014) and Pets Parkhi et al. (2012) are used for experiments. Table 7 reports the test accuracy. While the gains of CLIP-DR on some datasets are modest, the key finding is that CLIP-DR never decreases the original performance, unlike CLIP-OT which can cause degradation (e.g., on Imagenet-R, Pets). This reliability is crucial for real-world TTA applications.

Table 7: Performance on other datasets for CLIP-DR and CLIP-OT. **Bold** indicates the best result.

| Methods | Datasets | | | | | | | |
|---|---|---|---|---|---|---|---|---|
| | Imagenet-R | EuroSAT | Aircraft | Caltech-101 | ImageNetV2 | DTD | ImageNet-S | Pets |
| CLIP | 68.36 | 33.30 | 25.87 | 80.74 | 55.02 | 43.38 | 74.33 | 77.51 |
| CLIP-OT | 68.26 | 34.87 | 26.50 | **80.80** | 55.19 | 43.59 | 73.91 | 77.30 |
| CLIP-DR | **68.40** | **35.94** | **26.63** | 80.77 | **55.25** | **43.70** | **74.70** | **77.85** |

**Different backbones**. We replace the CLIP backbone with SigLIP Zhai et al. (2023) and EVA02-CLIP Fang et al. (2024) to evaluate the generalizability of the proposed dual-regularization approach. Table 8 reports the test accuracy on CIFAR-10/100-C datasets. CLIP-TTA consistently provides a higher Avg. accuracy (e.g., 89.38% vs. 86.92%) compared to CLIP-OT with SigLIP and EVA02-CLIP backbones. These results confirm that the proposed dual-regularization principle is a general and effective strategy for robust test-time adaptation, not just being a mere tweak for CLIP model.

Table 8: Accuracy (%) on CIFAR10-C (3-5 and 7-9 rows) and CIFAR100-C (11-13 and 15-17 rows) for CLIP-DR and baselines using different backbones. **Bold** indicates the best result.

| Methods | Noises | | | | | | | | | | | | | | | |
|---|---|---|---|---|---|---|---|---|---|---|---|---|---|---|---|---|
| SigLIP | Gaussian | Shot | Impulse | Defocus | Glass | Motion | Zoom | Snow | Frost | Fog | Brightness | Contrast | Elastic | Pixelate | JPEG | Avg. |
| SigLIP | 13.27 | 14.8 | 20.52 | 54.34 | 23.36 | 46 | 52.9 | 25.49 | 32.06 | 68.74 | 28.35 | 78.53 | 38.36 | 36.84 | 26.69 | 37.35 |
| CLIP-OT | 39.98 | 50.2 | 60.65 | 76.92 | 56.69 | 74.88 | 78.65 | 53.04 | 64.15 | 85.06 | 64.27 | **89.42** | 66.83 | 68.1 | 53.73 | 65.50 |
| CLIP-DR | **40.34** | **51.35** | **61.34** | **77.69** | **57.39** | **75.2** | **79.71** | **54.1** | **66.33** | **85.74** | **66.12** | 89.04 | **67.91** | **70.42** | **54.92** | **66.51** |
| EVA02-CLIP | | | | | | | | | | | | | | | | |
| EVA02-CLIP | 71.87 | 74.2 | 73.11 | 90.14 | 64.85 | 86.87 | 91.44 | 92.4 | **92.55** | **88.82** | 96.5 | 89.42 | 75.94 | 61.92 | 79.05 | 81.93 |
| CLIP-OT | 81.18 | 76.36 | 80.7 | 90.73 | 83.54 | 89.53 | 91.07 | 92.53 | 87.54 | 82.46 | **96.87** | 94.28 | 83.73 | **89.96** | 83.41 | 86.92 |
| CLIP-DR | **83.05** | **81.61** | **86.55** | **93.14** | **83.66** | **90.53** | **94.03** | **93.69** | 92.21 | | 94.89 | **94.52** | **87.85** | 87.72 | **85.59** | **89.38** |
| CIFAR-100-C | | | | | | | | | | | | | | | | |
| SigLIP | 2.78 | 2.97 | 4.51 | 6.85 | 17.49 | 24.04 | 5.26 | 8.57 | 37.98 | 6.82 | 46.08 | 10.01 | 5.88 | | | 14.18 |
| CLIP-OT | 11.96 | 14.63 | 20.39 | 40.41 | 19.66 | 37.45 | 42.03 | 20.01 | 26.08 | 54.21 | 21.65 | 58.16 | 25.66 | 26.87 | 18.4 | 29.17 |
| CLIP-DR | **12.89** | **15.71** | **21.12** | **41.32** | **20.05** | **38.19** | **43.04** | **21.03** | **26.72** | **55.1** | **22.82** | **59.23** | **26.27** | **27.58** | **18.73** | **29.99** |
| EVA02-CLIP | | | | | | | | | | | | | | | | |
| EVA02-CLIP | 45.66 | 47.74 | 45.37 | **68.84** | 36.99 | 64.78 | **72.28** | **71.86** | **71.55** | 65.29 | **80.21** | 64.23 | 45.28 | 40.14 | 53.75 | 58.26 |
| CLIP-OT | 55.44 | 52.94 | 56.89 | 66.54 | 54.35 | 63.15 | 67.7 | 66 | 68.22 | **65.64** | 72.79 | 64.68 | 57.15 | 62.43 | 59.37 | 62.22 |
| CLIP-DR | **56.74** | **56.44** | **58.86** | 67.78 | **57.1** | **65.86** | 69.94 | 68.75 | 71 | **65.64** | 76.54 | **67.51** | **61.26** | **65.65** | **61.35** | **64.69** |

# 5 CONCLUSION

In this study, we extend the study in CLIP-OT by introducing a cosine similarity loss to align the image features and text prototypes, while adding an IM regularization technique to reduce the uncertainties in prediction. Experimental results on seven datasets show that CLIP-DR outperforms recent state-of-the-art results in several settings with minimal computation overhead, highlighting its potential for efficient TTA.

ETHICS STATEMENT

This is a numerical simulation study for which no ethical approval was required.

REPRODUCIBILITY STATEMENT

To ensure reproducibility, we provide our implementation details and setups in Section 4. This includes information on implementation benchmark, hyperparameters, and computing resources. Our source code is publicly available at `https://anonymous.4open.science/r/TTA-Codes-6DF3`.

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

# A APPENDIX

## A.1 THE USE OF LARGE LANGUAGE MODELS

We affirm that all text, code, and data presented in this work were prepared by the authors. No large language models (e.g., ChatGPT) or other generative AI tools were used in the creation of this manuscript.

## A.2 ANALYSIS OF $\mathcal{L}_{cos}$ AND $\mathcal{L}_{IM}$ FOR REDUCING ECE

We provide the analysis of $\mathcal{L}_{cos}$ and $\mathcal{L}_{IM}$ for reducing ECE. Specifically, the ECE can be formulated as follows:

$$\text{ECE} = \sum_{m=1}^{M} \frac{|B_m|}{n} |\text{acc}(B_m) - \text{conf}(B_m)| \tag{13}$$

where $B_m$ is the m-th bin. $n$ is the number of samples. A higher ECE value is dominated by confidently wrong samples residing in high-confidence, low-accuracy bins.

Concerning $\mathcal{L}_{IM}$, it is defined as:

$$\mathcal{L}_{\text{IM}} = \frac{1}{|\mathcal{B}_t|} \sum_{\mathbf{x} \in \mathcal{B}_t} \underbrace{H(p(\mathbf{x}))}_{\text{confidence}} - \underbrace{H(\bar{p})}_{\text{diversity}} \tag{14}$$

The first term, namely confidence regularization, it encourages the model to make more confident predictions for each sample. This helps to reduce under-confidence, ensuring that predictions in medium-confidence bins are meaningful and not due to model uncertainty.

The second term, namely diversity regularization, it controls over-confidence of all predictions through maximizing $H(\bar{p})$. This directly penalizes the model if it becomes over-confident on average (e.g., always predicting a few classes with high probability), reducing the number of samples that are incorrectly assigned to high-confidence bins.

By jointly optimizing these two opposing objectives, $\mathcal{L}_{IM}$ strikes a balance that prevents both under-confident and over-confident predictions, leading to a better-calibrated model and a lower ECE.

Considering $\mathcal{L}_{cos}$, it is represented as:

$$\mathcal{L}_{cos} = \sum_{j=1}^{B} \exp(C(\mathbf{x}_j) - C_0) \cdot (1 - s_{\mathbf{O}, \bar{\mathbf{f}}_{\hat{y}}}) \cdot \mathbb{I}\{C(\mathbf{x}_j) \geq C_0\} \tag{15}$$

where $s$ is defined as:

$$s_{\mathbf{O}, \bar{\mathbf{f}}_{\hat{y}}} = \frac{\langle \mathbf{O}, \bar{\mathbf{f}}_{\hat{y}} \rangle}{\|\mathbf{O}\| \cdot \|\bar{\mathbf{f}}_{\hat{y}}\|} \, , \text{ where } \bar{\mathbf{f}}_{\hat{y}} = \mathbf{t}_{km}^{\top} \mathbf{t}_{km}^{\hat{y}} \, , \mathbf{O} = \mathbf{z}_i^{\top} \mathbf{t}_k / \tau \tag{16}$$

Specifically, it will push the image features toward the pseudo-text prototypes. Thus, it will change the logits $s$ with the following two points: 1) increase the number of correct predicted samples, and 2) decrease the incorrect predictions. This reduces the number of confidently wrong samples. These samples are the primary contributors to high ECE, as they cause large $|\text{acc}(B_m) - \text{conf}(B_m)|$ in high-confidence bins. By minimizing $\mathcal{L}_{cos}$, we systematically reduce this miscalibration.

In summary, $\mathcal{L}_{cos}$ and $\mathcal{L}_{IM}$ form a synergistic framework: $\mathcal{L}_{cos}$ works at the feature level to reduce the source of confident errors (noisy pseudo-labels), while $\mathcal{L}_{IM}$ operates at the output level to ensure a well-calibrated probability distribution. This theoretical framework is validated by our empirical results.

## A.3 DISCUSSION BETWEEN CLIP-DR AND C-TPT

In Yoon et al. (2024), they propose C-TPT (Calibrated Test-Time Prompt Tuning), sharing a similar goal of improving model calibration during TTA for CLIP without requiring labeled data. Specifically, C-TPT identifies that the choice of prompts considerably impacts calibration in CLIP, and

proposes that prompts leading to higher text feature dispersion result in better-calibrated predictions. Based on this insight, they propose the Average Text Feature Dispersion (ATFD) to optimize prompts for enhanced calibration. Their results show that ATFD has a negative correlation with ECE value (i.e., a higher ATFD value leads to a lower ECE value and vice versa). This analytical framework of linking text feature dispersion to calibration error provides a valuable perspective. While C-TPT offers a novel prompt-based strategy, our work demonstrates that calibration can be achieved through a feature and output space regularizations within a pseudo-labeling TTA framework. We will explore the integration of both prompt-tuning and image feature adaptation for CLIP-based TTA.

## A.4 ALGORITHM

Algorithm 1 shows the pseudo-code of CLIP-DR.

---

**Algorithm 1** Adaptation procedure of CLIP-DR for one corruption.

---

1: **input:** test dataset $\mathcal{X} = \{\mathbf{x}_n\}$ (images), text templates $\mathcal{T}$ (class descriptions), visual and text encoders $(\boldsymbol{\theta}, \phi)$.
  // Split $\mathcal{X}$ into $B$ batches of size $B_s$.
  // Compute $K \times M$ text prototypes ($\phi(\mathcal{T}), \forall m, k$).
2: $\mathbf{T} \in \mathbb{R}^{d \times K \times M} = [\mathbf{t}_{km}]_{k=1,\ldots,K, m=1,\ldots,M}$
3: **for** sampled minibatch $\{\mathbf{x}_i\}_{i=1}^{B_s}$ **do**
    // 1 - Test-time Adaptation
4:   **for** each template $m$ in $\{1, 2, \ldots, M\}$ **do**
5:     $\mathbf{T}_m \in \mathbb{R}^{d \times K}$ // class text embeddings for $m$.
6:     $\mathbf{Z} = [\mathbf{z}_1, \ldots, \mathbf{z}_{B_s}]$ // visual features ($\boldsymbol{\theta}_{\text{LN}}^{(m-1)}$).
7:     Compute codes $\mathbf{Q}_m^*$ // (Eq. 8).
8:     Compute cosine similarity regularization loss $\mathcal{L}_{cos}$ // (Eq. 10).
9:     Compute information maximization loss $\mathcal{L}_{IM}$ // (Eq. 11).
10:    $\mathbf{t}_k = \frac{1}{M} \sum_{m=1}^{M} \mathbf{t}_{km} \quad (\mathbf{t}_k \in \mathbb{R}^d)$ // average class text embeddings over all M templates.
11:    $\mathbf{P} = [\mathbf{p}_1, \ldots, \mathbf{p}_{B_s}]$ // predict: $\mathbf{t}_k, \forall k$, Eq. (1).
12:    Min. cross-entropy with $\mathbf{P}$ and $\mathbf{Q}_m^*$.
13:    Min. $\mathcal{L}_{cos}$ and $\mathcal{L}_{IM}$.
14:    $\theta_{\text{LN}}^{(m-1)} \to \theta_{\text{LN}}^{(m)}$                    // Update layer norm (LN) of $\boldsymbol{\theta}$.
15:  **end for**
    // 2 - Inference (for all images in the batch).
16:  $\mathbf{Z} = [\mathbf{z}_1, \ldots, \mathbf{z}_{B_s}]$                    // visual features (with $\boldsymbol{\theta}_{\text{LN}}^{(m)}$).
17:  $\mathbf{P} = [\mathbf{p}_1, \ldots, \mathbf{p}_{B_s}]$                    // predict with $\mathbf{t}_k, \forall k$, Eq. (1).
18: **end for**

---

## A.5 TEMPLATES

Table 9 reports the templates we used in our study.

Table 9: The different templates used during the experiments.

| | Template |
|---|---|
| 1: | "a photo of a {class $k$}" |
| 2: | "itap of a {class $k$}" |
| 3: | "a bad photo of the {class $k$}" |
| 4: | "a origami {class $k$}" |
| 5: | "a photo of the large {class $k$}" |
| 6: | "a {class $k$} in a video game" |
| 7: | "art of the {class $k$}" |
| 8: | "a photo of the small {class $k$}" |

## A.6 ADDITIONAL RESULTS

### A.6.1 RESULTS FOR DIFFERENT HYPER-PARAMETERS

Table 10 to Table 31 report the test accuracy on CIFAR-10/100-C and TinyImageNet-C datasets with different $\lambda_1$ and $\lambda_2$ values.

Table 10: Test accuracy (%) under different $\lambda_1$ values ($\lambda_2 = 0.1$) on CIFAR-10-C dataset.

| Corruption | $\lambda_1$ | | | | | | | | | |
| | 0.10 | 0.20 | 0.30 | 0.40 | 0.50 | 0.60 | 0.70 | 0.80 | 0.90 | 1.00 |
| --- | --- | --- | --- | --- | --- | --- | --- | --- | --- | --- |
| Gaussian Noise | 65.13 | 65.15 | 65.20 | 65.18 | 65.16 | 65.17 | 65.15 | 65.18 | 65.21 | 65.18 |
| Shot Noise | 67.35 | 67.34 | 67.34 | 67.37 | 67.36 | 67.35 | 67.34 | 67.33 | 67.34 | 67.34 |
| Impulse Noise | 62.27 | 62.26 | 62.25 | 62.20 | 62.23 | 62.25 | 62.25 | 62.23 | 62.22 | 62.25 |
| Defocus Blur | 82.33 | 82.33 | 82.32 | 82.32 | 82.36 | 82.37 | 82.33 | 82.33 | 82.34 | 82.34 |
| Glass Blur | 67.86 | 67.89 | 67.90 | 67.90 | 67.92 | 67.93 | 67.93 | 67.93 | 67.93 | 67.94 |
| Motion Blur | 81.41 | 81.41 | 81.41 | 81.41 | 81.41 | 81.41 | 81.41 | 81.42 | 81.44 | 81.43 |
| Zoom Blur | 83.34 | 83.36 | 83.39 | 83.42 | 83.42 | 83.42 | 83.45 | 83.43 | 83.42 | 83.40 |
| Snow | 83.81 | 83.82 | 83.81 | 83.80 | 83.80 | 83.80 | 83.77 | 83.78 | 83.75 | 83.75 |
| Frost | 83.33 | 83.34 | 83.34 | 83.35 | 83.35 | 83.35 | 83.35 | 83.35 | 83.36 | 83.36 |
| Fog | 82.26 | 82.29 | 82.30 | 82.30 | 82.31 | 82.31 | 82.34 | 82.32 | 82.35 | 82.35 |
| Brightness | 89.71 | 89.71 | 89.72 | 89.72 | 89.72 | 89.71 | 89.72 | 89.72 | 89.72 | 89.72 |
| Contrast | 84.99 | 84.99 | 85.00 | 84.98 | 84.97 | 84.97 | 84.96 | 84.96 | 84.97 | 84.97 |
| Elastic Transform | 76.05 | 76.04 | 76.04 | 76.04 | 76.05 | 76.06 | 76.06 | 76.07 | 76.06 | 76.08 |
| Pixelate | 76.95 | 76.94 | 76.96 | 76.96 | 76.96 | 76.95 | 76.95 | 76.95 | 76.93 | 76.92 |
| JPEG Compression | 70.21 | 70.21 | 70.22 | 70.21 | 70.21 | 70.19 | 70.17 | 70.18 | 70.19 | 70.20 |

Table 11: Test accuracy (%) under different $\lambda_1$ values ($\lambda_2 = 0.2$) on CIFAR-10-C dataset.

| Corruption | $\lambda_1$ | | | | | | | | | |
| | 0.10 | 0.20 | 0.30 | 0.40 | 0.50 | 0.60 | 0.70 | 0.80 | 0.90 | 1.00 |
| --- | --- | --- | --- | --- | --- | --- | --- | --- | --- | --- |
| Gaussian Noise | 65.29 | 65.17 | 65.23 | 65.17 | 64.99 | 64.99 | 64.93 | 64.93 | 64.82 | 64.75 |
| Shot Noise | 67.43 | 67.38 | 67.38 | 67.38 | 67.37 | 67.32 | 67.34 | 67.28 | 67.31 | 67.25 |
| Impulse Noise | 62.16 | 62.19 | 62.22 | 62.19 | 62.13 | 62.04 | 61.89 | 61.85 | 61.88 | 61.82 |
| Defocus Blur | 82.36 | 82.31 | 82.36 | 82.30 | 82.30 | 82.26 | 82.31 | 82.26 | 82.30 | 82.28 |
| Glass Blur | 67.87 | 67.76 | 67.68 | 67.68 | 67.62 | 67.63 | 67.58 | 67.54 | 67.54 | 67.61 |
| Motion Blur | 81.46 | 81.49 | 81.54 | 81.57 | 81.66 | 81.70 | 81.73 | 81.75 | 81.72 | 81.70 |
| Zoom Blur | 83.44 | 83.42 | 83.47 | 83.55 | 83.55 | 83.56 | 83.56 | 83.49 | 83.48 | 83.54 |
| Snow | 83.82 | 83.85 | 83.86 | 83.90 | 83.91 | 83.87 | 83.88 | 83.87 | 83.77 | 83.65 |
| Frost | 83.42 | 83.43 | 83.43 | 83.42 | 83.41 | 83.40 | 83.40 | 83.40 | 83.40 | 83.34 |
| Fog | 82.41 | 82.39 | 82.33 | 82.33 | 82.36 | 82.34 | 82.34 | 82.31 | 82.33 | 82.35 |
| Brightness | 89.78 | 89.79 | 89.75 | 89.77 | 89.75 | 89.75 | 89.76 | 89.74 | 89.73 | 89.74 |
| Contrast | 85.02 | 85.00 | 84.97 | 84.94 | 84.90 | 84.90 | 84.89 | 84.89 | 84.87 | 84.85 |
| Elastic Transform | 76.10 | 76.12 | 76.10 | 76.06 | 76.08 | 76.08 | 76.10 | 76.09 | 76.04 | 76.07 |
| Pixelate | 77.00 | 76.97 | 76.97 | 76.99 | 77.06 | 77.08 | 77.11 | 77.08 | 77.00 | 76.97 |
| JPEG Compression | 70.32 | 70.34 | 70.38 | 70.31 | 70.28 | 70.27 | 70.20 | 70.15 | 70.15 | 70.11 |

Table 12: Test accuracy (%) under different $\lambda_1$ values ($\lambda_2 = 0.3$) on CIFAR-10-C dataset.

| Corruption | $\lambda_1$ | | | | | | | | | |
| | 0.10 | 0.20 | 0.30 | 0.40 | 0.50 | 0.60 | 0.70 | 0.80 | 0.90 | 1.00 |
| --- | --- | --- | --- | --- | --- | --- | --- | --- | --- | --- |
| Gaussian Noise | 65.23 | 65.25 | 65.20 | 65.16 | 65.05 | 64.93 | 64.93 | 64.84 | 64.81 | 64.68 |
| Shot Noise | 67.46 | 67.44 | 67.38 | 67.34 | 67.33 | 67.27 | 67.31 | 67.24 | 67.27 | 67.16 |
| Impulse Noise | 62.18 | 62.13 | 62.20 | 62.10 | 62.07 | 61.99 | 61.92 | 61.99 | 61.98 | 61.90 |
| Defocus Blur | 82.46 | 82.33 | 82.39 | 82.40 | 82.40 | 82.37 | 82.34 | 82.28 | 82.29 | 82.34 |
| Glass Blur | 67.73 | 67.68 | 67.69 | 67.70 | 67.63 | 67.55 | 67.53 | 67.63 | 67.65 | 67.68 |
| Motion Blur | 81.61 | 81.61 | 81.68 | 81.70 | 81.74 | 81.74 | 81.73 | 81.75 | 81.73 | 81.66 |
| Zoom Blur | 83.49 | 83.51 | 83.52 | 83.52 | 83.53 | 83.54 | 83.58 | 83.54 | 83.56 | 83.56 |
| Snow | 83.91 | 83.95 | 83.94 | 83.96 | 83.98 | 83.94 | 83.93 | 83.85 | 83.79 | 83.70 |
| Frost | 83.43 | 83.43 | 83.44 | 83.41 | 83.43 | 83.40 | 83.36 | 83.35 | 83.36 | 83.38 |
| Fog | 82.35 | 82.31 | 82.29 | 82.30 | 82.35 | 82.37 | 82.40 | 82.36 | 82.33 | 82.34 |
| Brightness | 89.76 | 89.77 | 89.74 | 89.77 | 89.78 | 89.75 | 89.72 | 89.69 | 89.71 | 89.73 |
| Contrast | 85.05 | 84.98 | 84.98 | 84.96 | 84.92 | 84.92 | 84.89 | 84.86 | 84.85 | 84.84 |
| Elastic Transform | 76.11 | 76.15 | 76.11 | 76.02 | 76.01 | 76.10 | 76.14 | 76.13 | 76.11 | 76.09 |
| Pixelate | 77.06 | 77.13 | 77.11 | 77.16 | 77.19 | 77.14 | 77.16 | 77.08 | 77.05 | 77.01 |
| JPEG Compression | 70.38 | 70.45 | 70.48 | 70.38 | 70.33 | 70.28 | 70.24 | 70.20 | 70.13 | 70.13 |

Table 13: Test accuracy (%) under different $\lambda_1$ values ($\lambda_2 = 0.4$) on CIFAR-10-C dataset.

| Corruption | $\lambda_1$ | | | | | | | | | |
|---|---|---|---|---|---|---|---|---|---|---|
| | 0.10 | 0.20 | 0.30 | 0.40 | 0.50 | 0.60 | 0.70 | 0.80 | 0.90 | 1.00 |
| Gaussian Noise | 65.25 | 65.20 | 65.16 | 65.12 | 64.98 | 64.90 | 64.88 | 64.83 | 64.78 | 64.74 |
| Shot Noise | 67.37 | 67.37 | 67.38 | 67.37 | 67.28 | 67.21 | 67.25 | 67.26 | 67.22 | 67.28 |
| Impulse Noise | 62.16 | 62.18 | 62.05 | 62.06 | 62.01 | 61.98 | 61.95 | 61.98 | 62.04 | 61.98 |
| Defocus Blur | 82.42 | 82.35 | 82.36 | 82.36 | 82.36 | 82.37 | 82.34 | 82.34 | 82.33 | 82.32 |
| Glass Blur | 67.74 | 67.71 | 67.65 | 67.64 | 67.69 | 67.64 | 67.66 | 67.66 | 67.65 | 67.60 |
| Motion Blur | 81.68 | 81.71 | 81.69 | 81.70 | 81.68 | 81.73 | 81.75 | 81.70 | 81.64 | 81.65 |
| Zoom Blur | 83.55 | 83.56 | 83.57 | 83.61 | 83.63 | 83.59 | 83.58 | 83.58 | 83.60 | 83.61 |
| Snow | 84.00 | 84.03 | 84.05 | 84.02 | 84.00 | 83.98 | 83.90 | 83.84 | 83.81 | 83.72 |
| Frost | 83.49 | 83.45 | 83.42 | 83.40 | 83.40 | 83.39 | 83.40 | 83.36 | 83.38 | 83.38 |
| Fog | 82.35 | 82.34 | 82.34 | 82.36 | 82.35 | 82.39 | 82.36 | 82.35 | 82.37 | 82.30 |
| Brightness | 89.77 | 89.77 | 89.78 | 89.76 | 89.73 | 89.74 | 89.76 | 89.74 | 89.74 | 89.76 |
| Contrast | 84.99 | 84.97 | 84.98 | 84.93 | 84.92 | 84.86 | 84.86 | 84.84 | 84.86 | 84.83 |
| Elastic Transform | 76.14 | 76.09 | 76.09 | 76.08 | 76.12 | 76.18 | 76.19 | 76.16 | 76.19 | 76.15 |
| Pixelate | 77.06 | 77.10 | 77.09 | 77.14 | 77.11 | 77.16 | 77.09 | 77.09 | 77.08 | 77.05 |
| JPEG Compression | 70.45 | 70.45 | 70.42 | 70.39 | 70.32 | 70.28 | 70.28 | 70.15 | 70.17 | 70.20 |

Table 14: Test accuracy (%) under different $\lambda_1$ values ($\lambda_2 = 0.5$) on CIFAR-10-C dataset.

| Corruption | $\lambda_1$ | | | | | | | | | |
|---|---|---|---|---|---|---|---|---|---|---|
| | 0.10 | 0.20 | 0.30 | 0.40 | 0.50 | 0.60 | 0.70 | 0.80 | 0.90 | 1.00 |
| Gaussian Noise | 65.25 | 65.18 | 65.12 | 65.08 | 65.02 | 64.90 | 64.87 | 64.84 | 64.80 | 64.68 |
| Shot Noise | 67.38 | 67.36 | 67.35 | 67.32 | 67.24 | 67.24 | 67.28 | 67.25 | 67.15 | 67.21 |
| Impulse Noise | 62.24 | 62.18 | 62.08 | 62.02 | 62.03 | 62.03 | 62.05 | 62.02 | 62.02 | 61.97 |
| Defocus Blur | 82.40 | 82.37 | 82.31 | 82.36 | 82.35 | 82.32 | 82.35 | 82.32 | 82.31 | 82.34 |
| Glass Blur | 67.69 | 67.65 | 67.68 | 67.73 | 67.69 | 67.64 | 67.64 | 67.62 | 67.57 | 67.58 |
| Motion Blur | 81.64 | 81.66 | 81.72 | 81.71 | 81.70 | 81.67 | 81.68 | 81.65 | 81.67 | 81.62 |
| Zoom Blur | 83.60 | 83.61 | 83.59 | 83.62 | 83.58 | 83.59 | 83.58 | 83.56 | 83.58 | 83.58 |
| Snow | 84.10 | 84.09 | 84.02 | 84.03 | 83.99 | 83.96 | 83.88 | 83.86 | 83.79 | 83.74 |
| Frost | 83.45 | 83.41 | 83.37 | 83.37 | 83.36 | 83.38 | 83.39 | 83.41 | 83.42 | 83.44 |
| Fog | 82.38 | 82.36 | 82.39 | 82.38 | 82.40 | 82.38 | 82.41 | 82.37 | 82.36 | 82.34 |
| Brightness | 89.75 | 89.71 | 89.75 | 89.75 | 89.77 | 89.77 | 89.76 | 89.76 | 89.74 | 89.74 |
| Contrast | 84.98 | 84.96 | 84.93 | 84.89 | 84.87 | 84.86 | 84.81 | 84.85 | 84.87 | 84.86 |
| Elastic Transform | 76.16 | 76.15 | 76.16 | 76.16 | 76.16 | 76.15 | 76.15 | 76.22 | 76.22 | 76.22 |
| Pixelate | 77.11 | 77.13 | 77.13 | 77.14 | 77.11 | 77.08 | 77.09 | 77.10 | 77.08 | 77.03 |
| JPEG Compression | 70.50 | 70.46 | 70.37 | 70.35 | 70.31 | 70.31 | 70.25 | 70.23 | 70.25 | 70.27 |

Table 15: Test accuracy (%) under different $\lambda_1$ values ($\lambda_2 = 0.6$) on CIFAR-10-C dataset.

| Corruption | $\lambda_1$ | | | | | | | | | |
|---|---|---|---|---|---|---|---|---|---|---|
| | 0.10 | 0.20 | 0.30 | 0.40 | 0.50 | 0.60 | 0.70 | 0.80 | 0.90 | 1.00 |
| Gaussian Noise | 65.23 | 65.11 | 65.07 | 65.01 | 65.00 | 64.92 | 64.89 | 64.89 | 64.86 | 64.78 |
| Shot Noise | 67.31 | 67.38 | 67.36 | 67.29 | 67.34 | 67.28 | 67.27 | 67.14 | 67.20 | 67.14 |
| Impulse Noise | 62.18 | 62.07 | 62.08 | 62.05 | 62.08 | 62.12 | 62.05 | 62.02 | 62.00 | 61.97 |
| Defocus Blur | 82.34 | 82.39 | 82.37 | 82.41 | 82.35 | 82.35 | 82.33 | 82.35 | 82.34 | 82.33 |
| Glass Blur | 67.72 | 67.73 | 67.68 | 67.65 | 67.68 | 67.67 | 67.61 | 67.62 | 67.63 | 67.61 |
| Motion Blur | 81.68 | 81.66 | 81.67 | 81.61 | 81.63 | 81.63 | 81.62 | 81.62 | 81.61 | 81.58 |
| Zoom Blur | 83.59 | 83.58 | 83.57 | 83.56 | 83.57 | 83.53 | 83.52 | 83.53 | 83.56 | 83.54 |
| Snow | 84.09 | 84.09 | 84.07 | 84.04 | 83.99 | 83.99 | 83.91 | 83.83 | 83.81 | 83.77 |
| Frost | 83.42 | 83.43 | 83.44 | 83.42 | 83.40 | 83.39 | 83.39 | 83.43 | 83.46 | 83.49 |
| Fog | 82.39 | 82.39 | 82.41 | 82.43 | 82.37 | 82.40 | 82.40 | 82.39 | 82.38 | 82.39 |
| Brightness | 89.79 | 89.80 | 89.78 | 89.78 | 89.78 | 89.76 | 89.75 | 89.73 | 89.75 | 89.75 |
| Contrast | 84.94 | 84.92 | 84.90 | 84.90 | 84.89 | 84.85 | 84.87 | 84.85 | 84.87 | 84.88 |
| Elastic Transform | 76.21 | 76.25 | 76.24 | 76.23 | 76.24 | 76.27 | 76.24 | 76.23 | 76.24 | 76.22 |
| Pixelate | 77.06 | 77.10 | 77.14 | 77.09 | 77.07 | 77.10 | 77.09 | 77.07 | 77.07 | 77.07 |
| JPEG Compression | 70.45 | 70.40 | 70.42 | 70.34 | 70.31 | 70.31 | 70.26 | 70.28 | 70.30 | 70.27 |

### A.6.2 RESULTS FOR LAION-C

Table 41 reports the test accuracy on LAION-C Li et al. (2025). The results conclusively prove that the proposed dual regularization can enhance the performance within the OT optimization framework (e.g., a reliable performance gain (+1.46% in Avg.)).

Table 16: Test accuracy (%) under different $\lambda_1$ values ($\lambda_2 = 0.7$) on CIFAR-10-C dataset.

| Corruption | $\lambda_1$ | | | | | | | | | |
|---|---|---|---|---|---|---|---|---|---|---|
| | 0.10 | 0.20 | 0.30 | 0.40 | 0.50 | 0.60 | 0.70 | 0.80 | 0.90 | 1.00 |
| Gaussian Noise | 65.14 | 65.11 | 64.98 | 64.99 | 64.96 | 64.90 | 64.95 | 64.89 | 64.85 | 64.83 |
| Shot Noise | 67.36 | 67.44 | 67.35 | 67.36 | 67.25 | 67.23 | 67.17 | 67.14 | 67.16 | 67.13 |
| Impulse Noise | 62.08 | 62.01 | 61.99 | 62.08 | 62.11 | 62.11 | 62.04 | 62.05 | 62.00 | 61.97 |
| Defocus Blur | 82.39 | 82.43 | 82.43 | 82.41 | 82.42 | 82.37 | 82.33 | 82.38 | 82.41 | 82.35 |
| Glass Blur | 67.75 | 67.71 | 67.66 | 67.67 | 67.68 | 67.63 | 67.68 | 67.68 | 67.66 | 67.60 |
| Motion Blur | 81.60 | 81.60 | 81.57 | 81.59 | 81.60 | 81.60 | 81.61 | 81.59 | 81.61 | 81.60 |
| Zoom Blur | 83.53 | 83.54 | 83.55 | 83.55 | 83.57 | 83.50 | 83.51 | 83.56 | 83.54 | 83.53 |
| Snow | 84.08 | 84.05 | 84.05 | 84.06 | 84.02 | 83.99 | 83.91 | 83.85 | 83.81 | 83.79 |
| Frost | 83.45 | 83.44 | 83.45 | 83.46 | 83.44 | 83.44 | 83.44 | 83.46 | 83.46 | 83.44 |
| Fog | 82.42 | 82.43 | 82.44 | 82.41 | 82.39 | 82.43 | 82.44 | 82.43 | 82.40 | 82.38 |
| Brightness | 89.81 | 89.78 | 89.79 | 89.80 | 89.77 | 89.77 | 89.78 | 89.76 | 89.77 | 89.78 |
| Contrast | 84.93 | 84.91 | 84.93 | 84.87 | 84.85 | 84.86 | 84.87 | 84.87 | 84.90 | 84.87 |
| Elastic Transform | 76.30 | 76.27 | 76.33 | 76.28 | 76.24 | 76.26 | 76.25 | 76.24 | 76.28 | 76.25 |
| Pixelate | 77.04 | 77.05 | 77.02 | 77.06 | 77.06 | 77.09 | 77.08 | 77.02 | 77.07 | 77.11 |
| JPEG Compression | 70.44 | 70.41 | 70.35 | 70.32 | 70.30 | 70.32 | 70.26 | 70.23 | 70.26 | 70.26 |

Table 17: Test accuracy (%) under different $\lambda_1$ values ($\lambda_2 = 0.8$) on CIFAR-10-C dataset.

| Corruption | $\lambda_1$ | | | | | | | | | |
|---|---|---|---|---|---|---|---|---|---|---|
| | 0.10 | 0.20 | 0.30 | 0.40 | 0.50 | 0.60 | 0.70 | 0.80 | 0.90 | 1.00 |
| Gaussian Noise | 65.18 | 65.03 | 64.99 | 65.00 | 64.93 | 64.95 | 64.93 | 64.87 | 64.85 | 64.84 |
| Shot Noise | 67.42 | 67.36 | 67.34 | 67.30 | 67.26 | 67.22 | 67.16 | 67.13 | 67.15 | 67.18 |
| Impulse Noise | 62.00 | 62.04 | 62.06 | 62.07 | 62.09 | 62.05 | 62.05 | 62.03 | 62.03 | 61.99 |
| Defocus Blur | 82.45 | 82.44 | 82.42 | 82.44 | 82.46 | 82.38 | 82.41 | 82.42 | 82.42 | 82.39 |
| Glass Blur | 67.73 | 67.72 | 67.73 | 67.72 | 67.71 | 67.70 | 67.69 | 67.66 | 67.64 | 67.68 |
| Motion Blur | 81.61 | 81.56 | 81.57 | 81.59 | 81.59 | 81.59 | 81.59 | 81.59 | 81.58 | 81.60 |
| Zoom Blur | 83.55 | 83.53 | 83.55 | 83.56 | 83.59 | 83.53 | 83.51 | 83.50 | 83.49 | 83.50 |
| Snow | 84.02 | 84.04 | 84.03 | 84.02 | 84.07 | 84.02 | 83.96 | 83.89 | 83.81 | 83.84 |
| Frost | 83.49 | 83.46 | 83.44 | 83.46 | 83.47 | 83.47 | 83.46 | 83.44 | 83.44 | 83.47 |
| Fog | 82.44 | 82.43 | 82.46 | 82.47 | 82.44 | 82.41 | 82.38 | 82.43 | 82.41 | 82.40 |
| Brightness | 89.79 | 89.80 | 89.80 | 89.80 | 89.80 | 89.80 | 89.80 | 89.80 | 89.78 | 89.79 |
| Contrast | 84.90 | 84.91 | 84.88 | 84.85 | 84.85 | 84.87 | 84.88 | 84.90 | 84.88 | 84.90 |
| Elastic Transform | 76.25 | 76.25 | 76.27 | 76.28 | 76.33 | 76.30 | 76.26 | 76.27 | 76.26 | 76.18 |
| Pixelate | 77.00 | 77.01 | 77.03 | 77.04 | 77.04 | 77.01 | 77.01 | 77.02 | 77.04 | 77.07 |
| JPEG Compression | 70.42 | 70.34 | 70.31 | 70.35 | 70.32 | 70.31 | 70.26 | 70.27 | 70.28 | 70.25 |

Table 18: Test accuracy (%) under different $\lambda_1$ values ($\lambda_2 = 0.9$) on CIFAR-10-C dataset.

| Corruption | $\lambda_1$ | | | | | | | | | |
|---|---|---|---|---|---|---|---|---|---|---|
| | 0.10 | 0.20 | 0.30 | 0.40 | 0.50 | 0.60 | 0.70 | 0.80 | 0.90 | 1.00 |
| Gaussian Noise | 65.08 | 65.04 | 64.98 | 65.01 | 64.91 | 64.95 | 64.92 | 64.89 | 64.92 | 64.85 |
| Shot Noise | 67.36 | 67.32 | 67.35 | 67.26 | 67.22 | 67.15 | 67.14 | 67.17 | 67.17 | 67.17 |
| Impulse Noise | 62.10 | 62.10 | 62.08 | 62.11 | 62.11 | 62.05 | 62.03 | 62.03 | 62.01 | 61.99 |
| Defocus Blur | 82.47 | 82.41 | 82.41 | 82.43 | 82.44 | 82.42 | 82.44 | 82.43 | 82.45 | 82.43 |
| Glass Blur | 67.82 | 67.78 | 67.76 | 67.74 | 67.71 | 67.68 | 67.67 | 67.68 | 67.72 | 67.67 |
| Motion Blur | 81.55 | 81.55 | 81.55 | 81.56 | 81.62 | 81.57 | 81.58 | 81.56 | 81.55 | 81.58 |
| Zoom Blur | 83.52 | 83.57 | 83.55 | 83.56 | 83.56 | 83.51 | 83.48 | 83.46 | 83.52 | 83.51 |
| Snow | 84.07 | 84.04 | 84.04 | 84.03 | 84.00 | 83.98 | 83.93 | 83.88 | 83.84 | 83.78 |
| Frost | 83.50 | 83.46 | 83.48 | 83.47 | 83.48 | 83.50 | 83.50 | 83.49 | 83.49 | 83.51 |
| Fog | 82.46 | 82.46 | 82.43 | 82.43 | 82.44 | 82.45 | 82.41 | 82.42 | 82.39 | 82.38 |
| Brightness | 89.79 | 89.80 | 89.80 | 89.80 | 89.81 | 89.82 | 89.81 | 89.78 | 89.77 | 89.77 |
| Contrast | 84.91 | 84.88 | 84.86 | 84.84 | 84.86 | 84.87 | 84.88 | 84.88 | 84.86 | 84.90 |
| Elastic Transform | 76.25 | 76.30 | 76.31 | 76.30 | 76.30 | 76.28 | 76.25 | 76.21 | 76.19 | 76.20 |
| Pixelate | 77.07 | 77.01 | 77.01 | 77.03 | 77.00 | 77.00 | 76.99 | 76.99 | 76.99 | 77.03 |
| JPEG Compression | 70.40 | 70.40 | 70.39 | 70.34 | 70.33 | 70.31 | 70.30 | 70.32 | 70.29 | 70.27 |

### A.6.3 RESULTS FOR UPDATING TEXT ENCODER

Table 42 reports the test accuracy on CIAFR-10-C for updating both the image and text LayerNorm parameters. It can be seen that optimizing the text encoder $\phi$ leads to negative effects.

Table 19: Test accuracy (%) under different $\lambda_1$ values ($\lambda_2 = 1.0$) on CIFAR-10-C dataset.

| Corruption | $\lambda_1$ | | | | | | | | | |
|---|---|---|---|---|---|---|---|---|---|---|
| | 0.10 | 0.20 | 0.30 | 0.40 | 0.50 | 0.60 | 0.70 | 0.80 | 0.90 | 1.00 |
| Gaussian Noise | 65.17 | 65.11 | 65.02 | 64.98 | 64.93 | 64.95 | 64.92 | 64.92 | 64.91 | 64.94 |
| Shot Noise | 67.37 | 67.36 | 67.29 | 67.23 | 67.15 | 67.19 | 67.17 | 67.16 | 67.14 | 67.15 |
| Impulse Noise | 62.12 | 62.13 | 62.12 | 62.12 | 62.06 | 62.03 | 62.06 | 62.03 | 62.02 | 61.98 |
| Defocus Blur | 82.47 | 82.45 | 82.43 | 82.43 | 82.44 | 82.46 | 82.48 | 82.45 | 82.41 | 82.41 |
| Glass Blur | 67.83 | 67.80 | 67.77 | 67.75 | 67.72 | 67.71 | 67.70 | 67.72 | 67.71 | 67.69 |
| Motion Blur | 81.57 | 81.56 | 81.58 | 81.57 | 81.59 | 81.57 | 81.60 | 81.56 | 81.58 | 81.59 |
| Zoom Blur | 83.53 | 83.57 | 83.56 | 83.50 | 83.50 | 83.48 | 83.44 | 83.44 | 83.46 | 83.48 |
| Snow | 84.03 | 84.02 | 84.02 | 84.00 | 83.98 | 83.96 | 83.93 | 83.88 | 83.88 | 83.84 |
| Frost | 83.51 | 83.50 | 83.53 | 83.55 | 83.50 | 83.50 | 83.53 | 83.54 | 83.55 | 83.54 |
| Fog | 82.44 | 82.43 | 82.47 | 82.48 | 82.46 | 82.44 | 82.43 | 82.42 | 82.42 | 82.41 |
| Brightness | 89.77 | 89.79 | 89.82 | 89.82 | 89.81 | 89.82 | 89.77 | 89.76 | 89.77 | 89.76 |
| Contrast | 84.89 | 84.86 | 84.84 | 84.81 | 84.86 | 84.88 | 84.88 | 84.86 | 84.86 | 84.90 |
| Elastic Transform | 76.33 | 76.31 | 76.34 | 76.35 | 76.30 | 76.26 | 76.25 | 76.19 | 76.21 | 76.25 |
| Pixelate | 77.06 | 77.05 | 77.03 | 77.03 | 77.04 | 77.00 | 76.97 | 76.99 | 76.98 | 77.02 |
| JPEG Compression | 70.38 | 70.39 | 70.37 | 70.34 | 70.35 | 70.34 | 70.34 | 70.29 | 70.31 | 70.33 |

Table 20: Test accuracy (%) under different $\lambda_1$ values ($\lambda_2 = 0.1$) on CIFAR-100-C dataset.

| Corruption | $\lambda_1$ | | | | | | | | | |
|---|---|---|---|---|---|---|---|---|---|---|
| | 0.10 | 0.20 | 0.30 | 0.40 | 0.50 | 0.60 | 0.70 | 0.80 | 0.90 | 1.00 |
| Gaussian Noise | 33.55 | 33.51 | 33.52 | 33.53 | 33.48 | 33.38 | 33.37 | 33.40 | 33.35 | 33.22 |
| Shot Noise | 35.23 | 35.16 | 35.15 | 35.12 | 35.20 | 35.13 | 35.12 | 35.04 | 34.87 | 34.81 |
| Impulse Noise | 30.61 | 30.63 | 30.58 | 30.53 | 30.52 | 30.50 | 30.43 | 30.33 | 30.36 | 30.31 |
| Defocus Blur | 53.77 | 53.86 | 53.87 | 53.82 | 53.78 | 53.73 | 53.62 | 53.59 | 53.57 | 53.52 |
| Glass Blur | 35.3 | 35.35 | 35.36 | 35.36 | 35.20 | 35.07 | 34.96 | 34.80 | 34.79 | 34.93 |
| Motion Blur | 52.81 | 52.76 | 52.75 | 52.75 | 52.66 | 52.67 | 52.61 | 52.64 | 52.58 | 52.55 |
| Zoom Blur | 56.83 | 56.77 | 56.73 | 56.77 | 56.72 | 56.62 | 56.67 | 56.58 | 56.52 | 56.42 |
| Snow | 54.24 | 54.19 | 54.26 | 54.21 | 54.24 | 54.26 | 54.20 | 54.19 | 54.27 | 54.23 |
| Frost | 54.91 | 54.89 | 54.88 | 54.85 | 54.87 | 54.82 | 54.79 | 54.87 | 54.89 | 54.85 |
| Fog | 53.87 | 53.79 | 53.74 | 53.80 | 53.93 | 53.90 | 53.99 | 53.95 | 53.95 | 53.89 |
| Brightness | 64.47 | 64.34 | 64.41 | 64.42 | 64.47 | 64.46 | 64.49 | 64.46 | 64.50 | 64.48 |
| Contrast | 54.87 | 55.03 | 54.94 | 54.88 | 54.77 | 54.74 | 54.66 | 54.64 | 54.70 | 54.64 |
| Elastic Transform | 43.47 | 43.47 | 43.51 | 43.53 | 43.49 | 43.39 | 43.30 | 43.24 | 43.17 | 43.11 |
| Pixelate | 45.31 | 45.20 | 45.21 | 45.25 | 45.21 | 45.26 | 45.23 | 45.19 | 45.14 | 45.01 |
| JPEG Compression | 41.07 | 40.86 | 40.97 | 41.02 | 40.90 | 40.89 | 40.76 | 40.67 | 40.50 | 40.48 |

Table 21: Test accuracy (%) under different $\lambda_1$ values ($\lambda_2 = 0.2$) on CIFAR-100-C dataset.

| Corruption | $\lambda_1$ | | | | | | | | | |
|---|---|---|---|---|---|---|---|---|---|---|
| | 0.10 | 0.20 | 0.30 | 0.40 | 0.50 | 0.60 | 0.70 | 0.80 | 0.90 | 1.00 |
| Gaussian Noise | 33.51 | 33.57 | 33.48 | 33.47 | 33.44 | 33.41 | 33.43 | 33.37 | 33.29 | 33.25 |
| Shot Noise | 35.23 | 35.19 | 35.18 | 35.22 | 35.14 | 35.14 | 35.03 | 34.93 | 34.85 | 34.73 |
| Impulse Noise | 30.63 | 30.62 | 30.54 | 30.49 | 30.42 | 30.39 | 30.32 | 30.34 | 30.34 | 30.35 |
| Defocus Blur | 53.83 | 53.87 | 53.80 | 53.77 | 53.67 | 53.63 | 53.54 | 53.53 | 53.53 | 53.53 |
| Glass Blur | 35.36 | 35.31 | 35.27 | 35.18 | 35.03 | 34.91 | 34.81 | 34.87 | 34.94 | 34.94 |
| Motion Blur | 52.86 | 52.80 | 52.75 | 52.67 | 52.65 | 52.62 | 52.62 | 52.54 | 52.53 | 52.56 |
| Zoom Blur | 56.79 | 56.74 | 56.78 | 56.73 | 56.66 | 56.65 | 56.52 | 56.45 | 56.41 | 56.31 |
| Snow | 54.16 | 54.22 | 54.25 | 54.22 | 54.23 | 54.22 | 54.30 | 54.24 | 54.23 | 54.20 |
| Frost | 54.91 | 54.87 | 54.89 | 54.87 | 54.87 | 54.85 | 54.91 | 54.87 | 54.88 | 54.84 |
| Fog | 53.75 | 53.74 | 53.85 | 53.93 | 53.98 | 53.99 | 53.98 | 53.95 | 53.93 | 53.85 |
| Brightness | 64.34 | 64.42 | 64.46 | 64.48 | 64.49 | 64.50 | 64.49 | 64.46 | 64.52 | 64.52 |
| Contrast | 54.98 | 54.86 | 54.79 | 54.77 | 54.71 | 54.67 | 54.64 | 54.72 | 54.68 | 54.70 |
| Elastic Transform | 43.52 | 43.48 | 43.50 | 43.41 | 43.37 | 43.27 | 43.19 | 43.18 | 43.11 | 42.99 |
| Pixelate | 45.30 | 45.27 | 45.28 | 45.30 | 45.29 | 45.29 | 45.21 | 45.19 | 45.07 | 44.98 |
| JPEG Compression | 40.93 | 41.04 | 41.04 | 41.06 | 40.96 | 40.84 | 40.78 | 40.61 | 40.46 | 40.49 |

### A.6.4 RESULTS ON FINE-GRAINED DATASET

The effectiveness of $\mathcal{L}_{cos}$ relies on the discriminative text prototypes, which can be a limitation in extremely fine-grained datasets (e.g., Pets). To empirically evaluate the contribution of $\mathcal{L}_{cos}$, we validate the $\mathcal{L}_{cos}$ without $\mathcal{L}_{IM}$ on fine-grained datasets (Please see the Table below). As reported in the Table 43, introducing $\mathcal{L}_{cos}$ consistently improves upon CLIP-OT across all datasets. This suggests that $\mathcal{L}_{cos}$ does not shift the model toward incorrect priors but acts as a reliable regularizer

Table 22: Test accuracy (%) under different $\lambda_1$ values ($\lambda_2 = 0.3$) on CIFAR-100-C dataset.

| Corruption | $\lambda_1$ | | | | | | | | | |
|---|---|---|---|---|---|---|---|---|---|---|
| | 0.10 | 0.20 | 0.30 | 0.40 | 0.50 | 0.60 | 0.70 | 0.80 | 0.90 | 1.00 |
| Gaussian Noise | 33.55 | 33.50 | 33.44 | 33.46 | 33.47 | 33.40 | 33.36 | 33.32 | 33.21 | 33.22 |
| Shot Noise | 35.23 | 35.17 | 35.21 | 35.14 | 35.08 | 35.02 | 34.98 | 34.86 | 34.81 | 34.68 |
| Impulse Noise | 30.61 | 30.48 | 30.51 | 30.41 | 30.34 | 30.30 | 30.31 | 30.39 | 30.42 | 30.34 |
| Defocus Blur | 53.77 | 53.77 | 53.74 | 53.64 | 53.56 | 53.56 | 53.53 | 53.53 | 53.57 | 53.52 |
| Glass Blur | 35.30 | 35.19 | 35.08 | 35.00 | 34.93 | 34.88 | 34.92 | 34.90 | 34.94 | 34.79 |
| Motion Blur | 52.81 | 52.73 | 52.69 | 52.65 | 52.59 | 52.58 | 52.51 | 52.52 | 52.49 | 52.49 |
| Zoom Blur | 56.83 | 56.79 | 56.69 | 56.69 | 56.58 | 56.50 | 56.42 | 56.39 | 56.30 | 56.24 |
| Snow | 54.24 | 54.28 | 54.26 | 54.20 | 54.22 | 54.29 | 54.26 | 54.24 | 54.21 | 54.21 |
| Frost | 54.91 | 54.92 | 54.95 | 54.97 | 54.91 | 54.93 | 54.97 | 54.84 | 54.82 | 54.77 |
| Fog | 53.87 | 53.90 | 53.95 | 53.98 | 53.90 | 53.95 | 53.95 | 53.92 | 53.90 | 53.92 |
| Brightness | 64.47 | 64.48 | 64.46 | 64.46 | 64.54 | 64.52 | 64.55 | 64.50 | 64.52 | 64.48 |
| Contrast | 54.87 | 54.84 | 54.78 | 54.75 | 54.72 | 54.75 | 54.72 | 54.70 | 54.72 | 54.74 |
| Elastic Transform | 43.47 | 43.41 | 43.38 | 43.30 | 43.23 | 43.20 | 43.12 | 43.04 | 42.93 | 42.83 |
| Pixelate | 45.31 | 45.31 | 45.29 | 45.29 | 45.31 | 45.19 | 45.18 | 45.16 | 45.03 | 44.96 |
| JPEG Compression | 41.07 | 41.09 | 41.03 | 40.98 | 40.88 | 40.80 | 40.72 | 40.56 | 40.51 | 40.41 |

Table 23: Test accuracy (%) under different $\lambda_1$ values ($\lambda_2 = 0.4$) on CIFAR-100-C dataset.

| Corruption | $\lambda_1$ | | | | | | | | | |
|---|---|---|---|---|---|---|---|---|---|---|
| | 0.10 | 0.20 | 0.30 | 0.40 | 0.50 | 0.60 | 0.70 | 0.80 | 0.90 | 1.00 |
| Gaussian Noise | 33.52 | 33.46 | 33.44 | 33.50 | 33.39 | 33.29 | 33.29 | 33.24 | 33.25 | 33.22 |
| Shot Noise | 35.23 | 35.20 | 35.18 | 35.04 | 34.95 | 34.94 | 34.92 | 34.81 | 34.71 | 34.60 |
| Impulse Noise | 30.44 | 30.47 | 30.40 | 30.34 | 30.26 | 30.34 | 30.34 | 30.36 | 30.37 | 30.33 |
| Defocus Blur | 53.79 | 53.70 | 53.65 | 53.61 | 53.61 | 53.55 | 53.56 | 53.58 | 53.54 | 53.53 |
| Glass Blur | 35.16 | 35.07 | 35.00 | 34.94 | 34.93 | 34.92 | 34.86 | 34.90 | 34.81 | 34.70 |
| Motion Blur | 52.70 | 52.64 | 52.65 | 52.59 | 52.55 | 52.48 | 52.45 | 52.45 | 52.50 | 52.53 |
| Zoom Blur | 56.74 | 56.65 | 56.57 | 56.52 | 56.48 | 56.42 | 56.37 | 56.30 | 56.28 | 56.21 |
| Snow | 54.32 | 54.25 | 54.23 | 54.20 | 54.25 | 54.29 | 54.24 | 54.24 | 54.22 | 54.16 |
| Frost | 54.96 | 54.99 | 54.98 | 54.96 | 55.02 | 54.95 | 54.87 | 54.82 | 54.82 | 54.81 |
| Fog | 53.96 | 53.92 | 53.98 | 53.96 | 53.97 | 53.94 | 53.97 | 53.96 | 53.93 | 53.84 |
| Brightness | 64.48 | 64.48 | 64.52 | 64.53 | 64.55 | 64.54 | 64.48 | 64.46 | 64.44 | 64.48 |
| Contrast | 54.76 | 54.74 | 54.77 | 54.74 | 54.71 | 54.72 | 54.73 | 54.74 | 54.79 | 54.77 |
| Elastic Transform | 43.31 | 43.36 | 43.30 | 43.21 | 43.17 | 43.11 | 43.02 | 42.86 | 42.76 | 42.67 |
| Pixelate | 45.28 | 45.23 | 45.30 | 45.22 | 45.21 | 45.23 | 45.19 | 45.06 | 44.99 | 44.97 |
| JPEG Compression | 41.07 | 41.01 | 40.99 | 40.92 | 40.81 | 40.75 | 40.61 | 40.53 | 40.51 | 40.48 |

Table 24: Test accuracy (%) under different $\lambda_1$ values ($\lambda_2 = 0.5$) on CIFAR-100-C dataset.

| Corruption | $\lambda_1$ | | | | | | | | | |
|---|---|---|---|---|---|---|---|---|---|---|
| | 0.10 | 0.20 | 0.30 | 0.40 | 0.50 | 0.60 | 0.70 | 0.80 | 0.90 | 1.00 |
| Gaussian Noise | 33.50 | 33.47 | 33.45 | 33.34 | 33.24 | 33.32 | 33.23 | 33.23 | 33.18 | 33.14 |
| Shot Noise | 35.19 | 35.15 | 35.05 | 34.95 | 34.94 | 34.91 | 34.89 | 34.72 | 34.60 | 34.45 |
| Impulse Noise | 30.47 | 30.43 | 30.38 | 30.27 | 30.32 | 30.30 | 30.38 | 30.38 | 30.34 | 30.38 |
| Defocus Blur | 53.69 | 53.65 | 53.69 | 53.68 | 53.62 | 53.63 | 53.61 | 53.58 | 53.57 | 53.49 |
| Glass Blur | 35.02 | 34.96 | 34.91 | 34.90 | 34.89 | 34.90 | 34.94 | 34.80 | 34.66 | 34.55 |
| Motion Blur | 52.67 | 52.63 | 52.58 | 52.51 | 52.48 | 52.42 | 52.44 | 52.50 | 52.49 | 52.48 |
| Zoom Blur | 56.57 | 56.58 | 56.54 | 56.43 | 56.36 | 56.35 | 56.29 | 56.30 | 56.23 | 56.20 |
| Snow | 54.23 | 54.20 | 54.22 | 54.24 | 54.27 | 54.27 | 54.27 | 54.21 | 54.20 | 54.20 |
| Frost | 55.00 | 55.01 | 55.01 | 54.95 | 54.95 | 54.89 | 54.86 | 54.85 | 54.82 | 54.83 |
| Fog | 53.95 | 53.93 | 53.93 | 53.95 | 53.99 | 53.99 | 53.93 | 53.92 | 53.85 | 53.84 |
| Brightness | 64.52 | 64.53 | 64.52 | 64.51 | 64.49 | 64.45 | 64.48 | 64.49 | 64.46 | 64.48 |
| Contrast | 54.77 | 54.70 | 54.77 | 54.78 | 54.78 | 54.81 | 54.80 | 54.81 | 54.80 | 54.75 |
| Elastic Transform | 43.23 | 43.18 | 43.15 | 43.13 | 43.05 | 43.00 | 42.86 | 42.75 | 42.60 | 42.52 |
| Pixelate | 45.22 | 45.23 | 45.21 | 45.21 | 45.20 | 45.13 | 45.04 | 44.99 | 45.01 | 44.95 |
| JPEG Compression | 41.01 | 41.00 | 40.94 | 40.84 | 40.80 | 40.66 | 40.56 | 40.51 | 40.48 | 40.39 |

within the OT framework. We will explore more discriminative text representations in future work to solve the limitation of $\mathcal{L}_{cos}$.

### A.6.5   ECE VALUE FOR OTHER TTA TECHNIQUES WITH THE PROPOSED METHOD

We add more results about ECE value for other TTA techniques (TENT, TPT) with the proposed dual approach (Table 44 and Table 45). As reported, adding the proposed method to TENT can reduce the ECE value for certain noises such as Gaussian (0.08 vs. 0.12) and Shot (0.08 vs. 0.12).

Table 25: Test accuracy (%) under different $\lambda_1$ values ($\lambda_2 = 0.6$) on CIFAR-100-C dataset.

| Corruption | $\lambda_1$ | | | | | | | | | |
|---|---|---|---|---|---|---|---|---|---|---|
| | 0.10 | 0.20 | 0.30 | 0.40 | 0.50 | 0.60 | 0.70 | 0.80 | 0.90 | 1.00 |
| Gaussian Noise | 33.44 | 33.37 | 33.31 | 33.27 | 33.28 | 33.24 | 33.17 | 33.14 | 33.12 | 33.09 |
| Shot Noise | 35.14 | 35.07 | 34.88 | 34.93 | 34.89 | 34.86 | 34.72 | 34.57 | 34.49 | 34.39 |
| Impulse Noise | 30.44 | 30.36 | 30.30 | 30.35 | 30.30 | 30.36 | 30.38 | 30.34 | 30.35 | 30.34 |
| Defocus Blur | 53.79 | 53.75 | 53.77 | 53.65 | 53.67 | 53.61 | 53.56 | 53.56 | 53.52 | 53.45 |
| Glass Blur | 35.02 | 34.94 | 34.91 | 34.93 | 34.96 | 34.88 | 34.79 | 34.67 | 34.54 | 34.48 |
| Motion Blur | 52.60 | 52.56 | 52.59 | 52.52 | 52.50 | 52.48 | 52.46 | 52.47 | 52.49 | 52.43 |
| Zoom Blur | 56.61 | 56.53 | 56.45 | 56.40 | 56.33 | 56.32 | 56.27 | 56.25 | 56.21 | 56.12 |
| Snow | 54.18 | 54.16 | 54.17 | 54.17 | 54.24 | 54.21 | 54.18 | 54.16 | 54.17 | 54.14 |
| Frost | 55.01 | 55.01 | 54.98 | 54.97 | 54.89 | 54.86 | 54.88 | 54.88 | 54.85 | 54.80 |
| Fog | 53.95 | 53.92 | 53.95 | 53.97 | 53.98 | 53.97 | 53.89 | 53.87 | 53.84 | 53.79 |
| Brightness | 64.54 | 64.51 | 64.51 | 64.48 | 64.42 | 64.48 | 64.49 | 64.48 | 64.48 | 64.48 |
| Contrast | 54.74 | 54.77 | 54.82 | 54.81 | 54.82 | 54.86 | 54.84 | 54.82 | 54.79 | 54.74 |
| Elastic Transform | 43.18 | 43.10 | 43.08 | 43.03 | 42.93 | 42.82 | 42.71 | 42.62 | 42.55 | 42.47 |
| Pixelate | 45.22 | 45.16 | 45.17 | 45.16 | 45.07 | 45.01 | 45.02 | 44.98 | 44.95 | 44.90 |
| JPEG Compression | 40.97 | 40.99 | 40.83 | 40.82 | 40.76 | 40.57 | 40.51 | 40.49 | 40.48 | 40.41 |

Table 26: Test accuracy (%) under different $\lambda_1$ values ($\lambda_2 = 0.7$) on CIFAR-100-C dataset.

| Corruption | $\lambda_1$ | | | | | | | | | |
|---|---|---|---|---|---|---|---|---|---|---|
| | 0.10 | 0.20 | 0.30 | 0.40 | 0.50 | 0.60 | 0.70 | 0.80 | 0.90 | 1.00 |
| Gaussian Noise | 33.32 | 33.28 | 33.30 | 33.27 | 33.18 | 33.16 | 33.14 | 33.17 | 33.09 | 33.04 |
| Shot Noise | 35.00 | 34.90 | 34.95 | 34.90 | 34.80 | 34.73 | 34.64 | 34.53 | 34.40 | 34.35 |
| Impulse Noise | 30.39 | 30.33 | 30.31 | 30.32 | 30.39 | 30.41 | 30.38 | 30.35 | 30.37 | 30.33 |
| Defocus Blur | 53.81 | 53.76 | 53.71 | 53.67 | 53.61 | 53.62 | 53.56 | 53.54 | 53.42 | 53.38 |
| Glass Blur | 34.96 | 34.93 | 34.90 | 34.92 | 34.92 | 34.72 | 34.65 | 34.58 | 34.43 | 34.37 |
| Motion Blur | 52.56 | 52.55 | 52.52 | 52.50 | 52.49 | 52.47 | 52.50 | 52.47 | 52.45 | 52.48 |
| Zoom Blur | 56.48 | 56.45 | 56.43 | 56.39 | 56.32 | 56.30 | 56.26 | 56.21 | 56.17 | 56.10 |
| Snow | 54.12 | 54.15 | 54.19 | 54.21 | 54.16 | 54.15 | 54.16 | 54.15 | 54.13 | 54.12 |
| Frost | 54.99 | 54.96 | 54.96 | 54.87 | 54.87 | 54.88 | 54.87 | 54.87 | 54.84 | 54.81 |
| Fog | 53.93 | 53.92 | 53.95 | 53.93 | 53.93 | 53.88 | 53.90 | 53.80 | 53.77 | 53.73 |
| Brightness | 64.54 | 64.50 | 64.45 | 64.48 | 64.48 | 64.52 | 64.48 | 64.48 | 64.51 | 64.49 |
| Contrast | 54.81 | 54.82 | 54.83 | 54.86 | 54.84 | 54.84 | 54.78 | 54.79 | 54.74 | 54.72 |
| Elastic Transform | 43.09 | 43.05 | 42.99 | 42.94 | 42.86 | 42.71 | 42.60 | 42.53 | 42.51 | 42.44 |
| Pixelate | 45.15 | 45.10 | 45.11 | 45.07 | 45.03 | 45.02 | 45.02 | 44.96 | 44.89 | 44.83 |
| JPEG Compression | 40.99 | 40.88 | 40.82 | 40.79 | 40.65 | 40.57 | 40.51 | 40.50 | 40.46 | 40.42 |

Table 27: Test accuracy (%) under different $\lambda_1$ values ($\lambda_2 = 0.8$) on CIFAR-100-C dataset.

| Corruption | $\lambda_1$ | | | | | | | | | |
|---|---|---|---|---|---|---|---|---|---|---|
| | 0.10 | 0.20 | 0.30 | 0.40 | 0.50 | 0.60 | 0.70 | 0.80 | 0.90 | 1.00 |
| Gaussian Noise | 33.31 | 33.32 | 33.29 | 33.23 | 33.16 | 33.11 | 33.13 | 33.06 | 33.05 | 33.01 |
| Shot Noise | 34.95 | 34.91 | 34.88 | 34.85 | 34.72 | 34.70 | 34.54 | 34.43 | 34.34 | 34.23 |
| Impulse Noise | 30.35 | 30.31 | 30.28 | 30.35 | 30.39 | 30.40 | 30.36 | 30.38 | 30.35 | 30.28 |
| Defocus Blur | 53.75 | 53.76 | 53.66 | 53.69 | 53.62 | 53.59 | 53.55 | 53.44 | 53.41 | 53.36 |
| Glass Blur | 34.91 | 34.96 | 34.93 | 34.83 | 34.71 | 34.68 | 34.50 | 34.43 | 34.36 | 34.34 |
| Motion Blur | 52.55 | 52.57 | 52.55 | 52.56 | 52.52 | 52.49 | 52.47 | 52.49 | 52.47 | 52.46 |
| Zoom Blur | 56.47 | 56.45 | 56.39 | 56.34 | 56.28 | 56.30 | 56.27 | 56.24 | 56.20 | 56.11 |
| Snow | 54.11 | 54.16 | 54.16 | 54.14 | 54.18 | 54.17 | 54.12 | 54.14 | 54.16 | 54.13 |
| Frost | 54.97 | 54.91 | 54.87 | 54.82 | 54.84 | 54.86 | 54.83 | 54.86 | 54.85 | 54.81 |
| Fog | 53.97 | 53.92 | 53.92 | 53.91 | 53.87 | 53.85 | 53.82 | 53.79 | 53.76 | 53.74 |
| Brightness | 64.52 | 64.46 | 64.48 | 64.51 | 64.54 | 64.48 | 64.49 | 64.45 | 64.47 | 64.51 |
| Contrast | 54.87 | 54.86 | 54.87 | 54.86 | 54.84 | 54.83 | 54.81 | 54.75 | 54.74 | 54.75 |
| Elastic Transform | 43.00 | 43.00 | 42.94 | 42.80 | 42.67 | 42.64 | 42.55 | 42.50 | 42.44 | 42.45 |
| Pixelate | 45.14 | 45.09 | 45.04 | 44.99 | 45.02 | 44.98 | 44.95 | 44.91 | 44.88 | 44.84 |
| JPEG Compression | 40.88 | 40.80 | 40.79 | 40.75 | 40.64 | 40.58 | 40.51 | 40.47 | 40.44 | 40.44 |

Similar situation can be found using TPT (e.g., 0.06 vs. 0.07 with Frost). These results highlight the effectivenss of the proposed method in reducing ECE.

Table 28: Test accuracy (%) under different $\lambda_1$ values ($\lambda_2 = 0.9$) on CIFAR-100-C dataset.

| Corruption | $\lambda_1$ | | | | | | | | | |
|---|---|---|---|---|---|---|---|---|---|---|
| | 0.10 | 0.20 | 0.30 | 0.40 | 0.50 | 0.60 | 0.70 | 0.80 | 0.90 | 1.00 |
| Gaussian Noise | 33.29 | 33.29 | 33.21 | 33.17 | 33.12 | 33.14 | 33.10 | 33.05 | 32.99 | 32.97 |
| Shot Noise | 34.90 | 34.85 | 34.86 | 34.74 | 34.72 | 34.56 | 34.42 | 34.32 | 34.25 | 34.21 |
| Impulse Noise | 30.31 | 30.33 | 30.32 | 30.38 | 30.38 | 30.39 | 30.38 | 30.35 | 30.28 | 30.30 |
| Defocus Blur | 53.74 | 53.70 | 53.69 | 53.64 | 53.60 | 53.56 | 53.50 | 53.41 | 53.37 | 53.39 |
| Glass Blur | 34.95 | 34.88 | 34.83 | 34.71 | 34.64 | 34.56 | 34.42 | 34.37 | 34.34 | 34.31 |
| Motion Blur | 52.59 | 52.60 | 52.59 | 52.54 | 52.53 | 52.51 | 52.47 | 52.46 | 52.48 | 52.45 |
| Zoom Blur | 56.41 | 56.42 | 56.32 | 56.29 | 56.27 | 56.27 | 56.23 | 56.16 | 56.13 | 56.11 |
| Snow | 54.15 | 54.11 | 54.14 | 54.18 | 54.16 | 54.16 | 54.16 | 54.15 | 54.13 | 54.15 |
| Frost | 54.91 | 54.84 | 54.84 | 54.80 | 54.82 | 54.85 | 54.86 | 54.87 | 54.84 | 54.80 |
| Fog | 53.94 | 53.91 | 53.87 | 53.87 | 53.86 | 53.81 | 53.79 | 53.78 | 53.74 | 53.72 |
| Brightness | 64.50 | 64.52 | 64.52 | 64.54 | 64.49 | 64.52 | 64.48 | 64.50 | 64.50 | 64.52 |
| Contrast | 54.88 | 54.87 | 54.85 | 54.84 | 54.81 | 54.79 | 54.78 | 54.75 | 54.73 | 54.73 |
| Elastic Transform | 43.03 | 42.94 | 42.80 | 42.66 | 42.62 | 42.60 | 42.51 | 42.46 | 42.43 | 42.35 |
| Pixelate | 45.11 | 45.01 | 45.02 | 45.03 | 44.96 | 44.95 | 44.93 | 44.94 | 44.84 | 44.90 |
| JPEG Compression | 40.81 | 40.77 | 40.73 | 40.65 | 40.57 | 40.57 | 40.58 | 40.51 | 40.50 | 40.44 |

Table 29: Test accuracy (%) under different $\lambda_1$ values ($\lambda_2 = 1.0$) on CIFAR-100-C dataset.

| Corruption | $\lambda_1$ | | | | | | | | | |
|---|---|---|---|---|---|---|---|---|---|---|
| | 0.10 | 0.20 | 0.30 | 0.40 | 0.50 | 0.60 | 0.70 | 0.80 | 0.90 | 1.00 |
| Gaussian Noise | 33.27 | 33.25 | 33.20 | 33.14 | 33.15 | 33.14 | 33.06 | 33.03 | 32.98 | 32.93 |
| Shot Noise | 34.86 | 34.89 | 34.74 | 34.66 | 34.56 | 34.44 | 34.32 | 34.28 | 34.20 | 34.23 |
| Impulse Noise | 30.33 | 30.32 | 30.39 | 30.40 | 30.40 | 30.41 | 30.38 | 30.38 | 30.28 | 30.31 |
| Defocus Blur | 53.74 | 53.67 | 53.62 | 53.60 | 53.59 | 53.49 | 53.45 | 53.39 | 53.39 | 53.42 |
| Glass Blur | 34.92 | 34.81 | 34.70 | 34.64 | 34.55 | 34.42 | 34.34 | 34.31 | 34.29 | 34.22 |
| Motion Blur | 52.62 | 52.59 | 52.57 | 52.54 | 52.53 | 52.51 | 52.48 | 52.47 | 52.48 | 52.48 |
| Zoom Blur | 56.36 | 56.32 | 56.32 | 56.28 | 56.25 | 56.22 | 56.21 | 56.16 | 56.17 | 56.13 |
| Snow | 54.13 | 54.15 | 54.16 | 54.17 | 54.14 | 54.14 | 54.17 | 54.14 | 54.16 | 54.18 |
| Frost | 54.87 | 54.85 | 54.83 | 54.84 | 54.85 | 54.85 | 54.87 | 54.81 | 54.82 | 54.73 |
| Fog | 53.90 | 53.89 | 53.86 | 53.87 | 53.85 | 53.80 | 53.79 | 53.76 | 53.75 | 53.75 |
| Brightness | 64.50 | 64.54 | 64.54 | 64.53 | 64.52 | 64.52 | 64.52 | 64.50 | 64.52 | 64.53 |
| Contrast | 54.89 | 54.85 | 54.85 | 54.82 | 54.78 | 54.78 | 54.77 | 54.72 | 54.71 | 54.75 |
| Elastic Transform | 42.90 | 42.73 | 42.65 | 42.63 | 42.59 | 42.54 | 42.47 | 42.44 | 42.40 | 42.32 |
| Pixelate | 45.03 | 45.00 | 44.95 | 44.97 | 44.97 | 44.91 | 44.95 | 44.91 | 44.91 | 44.88 |
| JPEG Compression | 40.80 | 40.77 | 40.71 | 40.59 | 40.54 | 40.57 | 40.54 | 40.47 | 40.47 | 40.40 |

Table 30: Accuracy (%) of the different approaches on Tiny-ImageNet and Tiny-ImageNet-C benchmarks, using ViT-B/32 as backbone.

| Dataset | CLIP ICLR'21 | TENT ICLR'21 | TPT NeurIPS'22 | CLIPTT WACV'25 | WATT-S NeurIPS'24 | TDA CVPR'24 | CLIP-OT Arxiv'25 | CLIP-TTA Ours |
|---|---|---|---|---|---|---|---|---|
| Tiny-Imagenet | 58.29 | 57.72± 0.14 | 58.90± 0.15 | 59.85± 0.01 | 61.35± 0.17 | 62.36±0.13 | 63.69± 0.15 | **64.38±0.14** |
| Gaussian Noise | 7.08 | 8.01± 0.05 | 9.29± 0.03 | 14.44± 0.25 | 13.02± 0.07 | 24.01±0.26 | 21.36± 0.31 | **24.76±0.31** |
| Shot Noise | 9.41 | 10.04± 0.07 | 11.70± 0.07 | 17.44± 0.12 | 15.94± 0.07 | 28.21±0.11 | 24.51± 0.15 | **32.53±0.14** |
| Impulse Noise | 3.44 | 4.18± 0.02 | 4.85± 0.01 | 10.37± 0.16 | 6.90± 0.01 | 18.65±0.03 | 18.21± 0.14 | **24.68±0.09** |
| Defocus Blur | 21.71 | 24.53± 0.05 | 27.56± 0.10 | 31.46± 0.26 | 29.91± 0.13 | 36.77±0.11 | 36.25± 0.17 | **44.42±0.15** |
| Glass Blur | 9.12 | 10.09± 0.06 | 11.03± 0.10 | 15.84± 0.09 | 14.01± 0.10 | 22.58±0.13 | 23.08± 0.18 | **29.24±0.14** |
| Motion Blur | 34.52 | 36.94± 0.07 | 38.97± 0.05 | 41.34± 0.20 | 41.26± 0.04 | 37.38±0.22 | 47.78± 0.34 | **49.45±0.32** |
| Zoom Blur | 27.44 | 29.48± 0.06 | 34.29± 0.07 | 35.06± 0.15 | 33.96± 0.07 | 36.59±0.07 | 40.96± 0.06 | **44.78±0.08** |
| Snow | 32.51 | 32.20± 0.03 | 34.45± 0.10 | 36.79± 0.08 | 37.76± 0.09 | 33.23±0.11 | 41.28± 0.09 | **43.85±0.13** |
| Frost | 36.33 | 35.72± 0.02 | 37.13± 0.05 | 38.37± 0.13 | 39.65± 0.03 | 35.94±0.15 | 46.03± 0.21 | **47.66±0.16** |
| Fog | 25.94 | 27.46± 0.04 | 28.89± 0.08 | 33.51± 0.19 | 32.13± 0.08 | 38.17±0.12 | 39.93± 0.25 | **49.31±0.23** |
| Brightness | 40.15 | 39.79± 0.08 | 43.31± 0.10 | 46.52±0.06 | 46.93± 0.13 | 42.98±0.07 | 53.81± 0.09 | **57.9±0.09** |
| Contrast | 1.81 | 2.24± 0.04 | 3.15± 0.04 | 6.07±0.12 | 3.53± 0.02 | 14.70±0.09 | 12.94± 0.21 | **18.21±0.15** |
| Elastic Transform | 30.40 | 31.92± 0.05 | 33.88± 0.14 | 33.74±0.11 | 35.01± 0.13 | 42.38±0.11 | 41.53± 0.17 | **49.5±0.13** |
| Pixelate | 22.78 | 24.79± 0.03 | 27.70± 0.12 | 34.84±0.09 | 31.55± 0.08 | 34.97±0.11 | 42.90± 0.13 | **47.54±0.08** |
| JPEG Compression | 29.59 | 30.93± 0.13 | 33.60± 0.15 | 37.29±0.14 | 36.46± 0.11 | 40.64±0.13 | 42.88± 0.05 | **50.93±0.07** |
| **Mean** | 22.14 | 23.22 | 25.32 | 28.87 | 27.87 | 32.48 | 35.56 | **40.98** |

Table 31: Test accuracy (%) under different $\lambda_1$ values ($\lambda_2 = 0.1$) on TinyImageNet-C dataset.

| Corruption | $\lambda_1$ | | | | | | | | | |
| --- | --- | --- | --- | --- | --- | --- | --- | --- | --- | --- |
| | 0.10 | 0.20 | 0.30 | 0.40 | 0.50 | 0.60 | 0.70 | 0.80 | 0.90 | 1.00 |
| Gaussian Noise | 23.87 | 24.10 | 24.26 | 24.44 | 24.59 | 24.59 | 24.67 | 24.61 | 24.46 | 24.76 |
| Shot Noise | 29.16 | 29.57 | 29.94 | 30.19 | 30.42 | 30.86 | 31.03 | 31.41 | 31.90 | 32.53 |
| Impulse Noise | 20.55 | 21.16 | 21.71 | 22.10 | 22.35 | 22.55 | 22.50 | 22.89 | 23.26 | 24.68 |
| Defocus Blur | 33.09 | 37.84 | 39.96 | 40.35 | 40.99 | 41.39 | 41.83 | 42.47 | 42.95 | 44.42 |
| Glass Blur | 26.81 | 26.87 | 26.99 | 27.38 | 27.80 | 28.14 | 28.46 | 28.32 | 28.61 | 29.24 |
| Motion Blur | 48.99 | 49.00 | 48.95 | 49.08 | 49.25 | 49.25 | 49.30 | 49.37 | 49.33 | 49.45 |
| Zoom Blur | 38.74 | 39.53 | 40.33 | 41.52 | 42.49 | 42.78 | 43.46 | 44.12 | 44.84 | 44.78 |
| Snow | 43.18 | 43.20 | 43.29 | 43.33 | 43.36 | 43.46 | 43.51 | 43.57 | 43.67 | 43.85 |
| Frost | 47.16 | 47.16 | 47.24 | 47.37 | 47.41 | 47.51 | 47.48 | 47.45 | 47.53 | 47.66 |
| Fog | 47.49 | 47.78 | 47.91 | 48.03 | 48.06 | 48.13 | 48.25 | 48.51 | 48.75 | 49.31 |
| Brightness | 56.75 | 57.00 | 57.09 | 57.21 | 57.36 | 57.44 | 57.59 | 57.68 | 57.71 | 57.9 |
| Contrast | 14.83 | 15.42 | 15.69 | 16.00 | 16.19 | 16.29 | 17.34 | 17.61 | 17.81 | 18.21 |
| Elastic Transform | 38.56 | 42.15 | 44.06 | 45.02 | 45.85 | 46.77 | 47.44 | 47.97 | 48.39 | 49.5 |
| Pixelate | 45.50 | 45.85 | 46.15 | 46.45 | 46.59 | 46.81 | 47.00 | 47.16 | 47.21 | 47.54 |
| JPEG Compression | 48.85 | 49.25 | 49.49 | 49.85 | 50.04 | 50.18 | 50.33 | 50.37 | 50.46 | 50.93 |

Table 32: Test accuracy (%) under different $\lambda_1$ values ($\lambda_2 = 0.2$) on TinyImageNet-C dataset.

| Corruption | $\lambda_1$ | | | | | | | | | |
| --- | --- | --- | --- | --- | --- | --- | --- | --- | --- | --- |
| | 0.10 | 0.20 | 0.30 | 0.40 | 0.50 | 0.60 | 0.70 | 0.80 | 0.90 | 1.00 |
| Gaussian Noise | 23.99 | 24.11 | 24.41 | 24.48 | 24.51 | 24.54 | 24.51 | 24.34 | 24.40 | 24.49 |
| Shot Noise | 29.49 | 29.86 | 30.06 | 30.29 | 30.65 | 30.83 | 31.21 | 31.70 | 31.80 | 31.91 |
| Impulse Noise | 21.04 | 21.52 | 21.96 | 22.21 | 22.36 | 22.33 | 22.59 | 23.02 | 23.32 | 23.50 |
| Defocus Blur | 38.26 | 39.80 | 40.12 | 40.82 | 41.15 | 41.64 | 41.99 | 42.48 | 42.92 | 43.65 |
| Glass Blur | 26.86 | 26.96 | 27.33 | 27.72 | 28.03 | 27.89 | 27.99 | 28.24 | 28.46 | 28.92 |
| Motion Blur | 48.93 | 49.00 | 49.14 | 49.22 | 49.24 | 49.30 | 49.35 | 49.36 | 49.44 | 49.51 |
| Zoom Blur | 39.43 | 40.25 | 41.35 | 42.22 | 42.67 | 43.17 | 43.87 | 44.35 | 44.75 | 44.32 |
| Snow | 43.23 | 43.25 | 43.34 | 43.38 | 43.46 | 43.49 | 43.65 | 43.67 | 43.70 | 43.74 |
| Frost | 47.18 | 47.27 | 47.34 | 47.40 | 47.37 | 47.39 | 47.42 | 47.55 | 47.56 | 47.53 |
| Fog | 47.58 | 47.75 | 47.96 | 47.96 | 48.04 | 48.13 | 48.30 | 48.62 | 48.89 | 48.96 |
| Brightness | 57.03 | 57.03 | 57.22 | 57.32 | 57.43 | 57.59 | 57.63 | 57.66 | 57.75 | 57.75 |
| Contrast | 15.29 | 15.56 | 15.83 | 15.97 | 16.16 | 16.25 | 17.33 | 17.71 | 17.81 | 18.01 |
| Elastic Transform | 41.92 | 43.70 | 44.83 | 45.57 | 46.51 | 47.14 | 47.74 | 48.03 | 48.45 | 48.81 |
| Pixelate | 45.93 | 46.19 | 46.45 | 46.59 | 46.80 | 47.00 | 47.13 | 47.20 | 47.28 | 47.40 |
| JPEG Compression | 49.19 | 49.32 | 49.76 | 49.95 | 50.14 | 50.18 | 50.31 | 50.44 | 50.47 | 50.66 |

Table 33: Test accuracy (%) under different $\lambda_1$ values ($\lambda_2 = 0.3$) on TinyImageNet-C dataset.

| Corruption | $\lambda_1$ | | | | | | | | | |
| --- | --- | --- | --- | --- | --- | --- | --- | --- | --- | --- |
| | 0.10 | 0.20 | 0.30 | 0.40 | 0.50 | 0.60 | 0.70 | 0.80 | 0.90 | 1.00 |
| Gaussian Noise | 24.11 | 24.30 | 24.35 | 24.41 | 24.46 | 24.46 | 24.24 | 24.29 | 24.36 | 24.25 |
| Shot Noise | 29.74 | 29.99 | 30.20 | 30.51 | 30.59 | 30.94 | 31.32 | 31.58 | 31.71 | 31.74 |
| Impulse Noise | 21.35 | 21.77 | 22.06 | 22.23 | 22.25 | 22.43 | 22.73 | 23.12 | 23.46 | 23.65 |
| Defocus Blur | 39.62 | 39.99 | 40.45 | 40.90 | 41.14 | 41.65 | 42.10 | 42.49 | 42.99 | 43.65 |
| Glass Blur | 27.01 | 27.36 | 27.54 | 27.82 | 27.67 | 27.85 | 27.86 | 27.95 | 28.39 | 28.96 |
| Motion Blur | 49.08 | 49.19 | 49.30 | 49.26 | 49.31 | 49.33 | 49.37 | 49.44 | 49.55 | 49.56 |
| Zoom Blur | 40.15 | 41.03 | 42.07 | 42.52 | 42.79 | 43.66 | 44.11 | 44.71 | 44.15 | 44.30 |
| Snow | 43.27 | 43.38 | 43.43 | 43.45 | 43.50 | 43.63 | 43.60 | 43.66 | 43.68 | 43.72 |
| Frost | 47.23 | 47.30 | 47.34 | 47.30 | 47.32 | 47.36 | 47.50 | 47.53 | 47.44 | 47.50 |
| Fog | 47.63 | 47.87 | 47.92 | 47.93 | 48.05 | 48.11 | 48.40 | 48.68 | 48.87 | 48.94 |
| Brightness | 57.05 | 57.20 | 57.31 | 57.46 | 57.51 | 57.59 | 57.63 | 57.69 | 57.75 | 57.77 |
| Contrast | 15.41 | 15.71 | 15.89 | 16.01 | 16.19 | 16.44 | 17.44 | 17.74 | 17.92 | 18.04 |
| Elastic Transform | 43.38 | 44.71 | 45.27 | 46.26 | 46.92 | 47.45 | 47.81 | 48.18 | 48.57 | 48.84 |
| Pixelate | 46.20 | 46.47 | 46.60 | 46.79 | 47.01 | 47.11 | 47.18 | 47.28 | 47.38 | 47.43 |
| JPEG Compression | 49.20 | 49.59 | 49.82 | 50.08 | 50.06 | 50.28 | 50.39 | 50.39 | 50.56 | 50.71 |

Table 34: Test accuracy (%) under different $\lambda_1$ values ($\lambda_2 = 0.4$) on TinyImageNet-C dataset.

| Corruption | $\lambda_1$ | | | | | | | | | |
|---|---|---|---|---|---|---|---|---|---|---|
| | 0.10 | 0.20 | 0.30 | 0.40 | 0.50 | 0.60 | 0.70 | 0.80 | 0.90 | 1.00 |
| Gaussian Noise | 24.22 | 24.21 | 24.31 | 24.36 | 24.37 | 24.17 | 24.18 | 24.25 | 24.18 | 24.16 |
| Shot Noise | 29.90 | 30.12 | 30.38 | 30.57 | 30.72 | 31.06 | 31.25 | 31.35 | 31.52 | 31.66 |
| Impulse Noise | 21.68 | 21.91 | 22.02 | 22.23 | 22.27 | 22.55 | 22.88 | 23.12 | 23.33 | 23.68 |
| Defocus Blur | 39.75 | 40.09 | 40.53 | 40.86 | 41.21 | 41.72 | 42.23 | 42.68 | 43.07 | 43.62 |
| Glass Blur | 27.28 | 27.46 | 27.62 | 27.52 | 27.73 | 27.73 | 27.73 | 27.81 | 28.18 | 28.83 |
| Motion Blur | 49.21 | 49.27 | 49.28 | 49.29 | 49.30 | 49.33 | 49.43 | 49.54 | 49.59 | 49.59 |
| Zoom Blur | 40.91 | 41.82 | 42.37 | 42.70 | 43.52 | 43.94 | 44.57 | 44.50 | 44.04 | 44.06 |
| Snow | 43.31 | 43.45 | 43.51 | 43.49 | 43.63 | 43.59 | 43.64 | 43.64 | 43.68 | 43.72 |
| Frost | 47.23 | 47.33 | 47.30 | 47.32 | 47.41 | 47.45 | 47.46 | 47.47 | 47.52 | 47.55 |
| Fog | 47.81 | 47.89 | 47.86 | 47.94 | 48.03 | 48.26 | 48.47 | 48.67 | 48.85 | 48.90 |
| Brightness | 57.21 | 57.33 | 57.46 | 57.49 | 57.58 | 57.58 | 57.64 | 57.68 | 57.74 | 57.84 |
| Contrast | 15.63 | 15.78 | 15.93 | 16.02 | 16.17 | 16.46 | 17.55 | 17.76 | 17.96 | 18.06 |
| Elastic Transform | 44.36 | 45.10 | 46.03 | 46.70 | 47.22 | 47.61 | 48.05 | 48.39 | 48.57 | 48.77 |
| Pixelate | 46.51 | 46.63 | 46.82 | 46.95 | 47.09 | 47.18 | 47.23 | 47.35 | 47.40 | 47.44 |
| JPEG Compression | 49.46 | 49.76 | 49.96 | 50.03 | 50.10 | 50.31 | 50.39 | 50.41 | 50.55 | 50.81 |

Table 35: Test accuracy (%) under different $\lambda_1$ values ($\lambda_2 = 0.5$) on TinyImageNet-C dataset.

| Corruption | $\lambda_1$ | | | | | | | | | |
|---|---|---|---|---|---|---|---|---|---|---|
| | 0.10 | 0.20 | 0.30 | 0.40 | 0.50 | 0.60 | 0.70 | 0.80 | 0.90 | 1.00 |
| Gaussian Noise | 24.11 | 24.17 | 24.29 | 24.26 | 24.11 | 24.09 | 24.15 | 24.13 | 24.12 | 24.02 |
| Shot Noise | 29.98 | 30.19 | 30.42 | 30.53 | 30.85 | 31.06 | 31.09 | 31.18 | 31.36 | 31.52 |
| Impulse Noise | 21.81 | 22.02 | 22.18 | 22.18 | 22.46 | 22.64 | 22.89 | 23.12 | 23.39 | 23.60 |
| Defocus Blur | 39.67 | 40.00 | 40.44 | 40.79 | 41.12 | 41.77 | 42.27 | 42.61 | 42.97 | 43.63 |
| Glass Blur | 27.44 | 27.44 | 27.39 | 27.63 | 27.68 | 27.71 | 27.72 | 27.86 | 28.14 | 28.37 |
| Motion Blur | 49.27 | 49.28 | 49.28 | 49.34 | 49.39 | 49.46 | 49.58 | 49.58 | 49.62 | 49.56 |
| Zoom Blur | 41.53 | 42.06 | 42.56 | 43.33 | 43.77 | 44.07 | 44.74 | 44.04 | 44.10 | 44.03 |
| Snow | 43.44 | 43.47 | 43.49 | 43.60 | 43.58 | 43.59 | 43.70 | 43.74 | 43.70 | 43.71 |
| Frost | 47.27 | 47.24 | 47.30 | 47.40 | 47.39 | 47.39 | 47.42 | 47.48 | 47.52 | 47.56 |
| Fog | 47.84 | 47.82 | 47.84 | 47.93 | 48.13 | 48.38 | 48.52 | 48.71 | 48.84 | 48.95 |
| Brightness | 57.37 | 57.52 | 57.49 | 57.58 | 57.60 | 57.66 | 57.72 | 57.77 | 57.84 | 57.90 |
| Contrast | 15.69 | 15.79 | 15.92 | 16.13 | 16.30 | 16.72 | 17.65 | 17.82 | 17.95 | 17.91 |
| Elastic Transform | 44.97 | 45.82 | 46.41 | 46.95 | 47.28 | 47.69 | 48.15 | 48.42 | 48.58 | 48.71 |
| Pixelate | 46.65 | 46.77 | 46.93 | 47.04 | 47.13 | 47.21 | 47.35 | 47.36 | 47.45 | 47.49 |
| JPEG Compression | 49.69 | 49.91 | 49.97 | 50.02 | 50.22 | 50.37 | 50.42 | 50.46 | 50.61 | 50.90 |

Table 36: Test accuracy (%) under different $\lambda_1$ values ($\lambda_2 = 0.6$) on TinyImageNet-C dataset.

| Corruption | $\lambda_1$ | | | | | | | | | |
|---|---|---|---|---|---|---|---|---|---|---|
| | 0.10 | 0.20 | 0.30 | 0.40 | 0.50 | 0.60 | 0.70 | 0.80 | 0.90 | 1.00 |
| Gaussian Noise | 24.12 | 24.25 | 24.20 | 24.08 | 23.98 | 24.05 | 24.09 | 23.99 | 23.88 | 23.81 |
| Shot Noise | 30.12 | 30.17 | 30.34 | 30.59 | 30.79 | 30.90 | 30.94 | 31.17 | 31.41 | 31.46 |
| Impulse Noise | 21.86 | 22.04 | 22.17 | 22.28 | 22.57 | 22.70 | 22.88 | 23.17 | 23.40 | 23.60 |
| Defocus Blur | 39.76 | 40.17 | 40.65 | 40.84 | 41.21 | 41.84 | 42.34 | 42.75 | 43.13 | 43.52 |
| Glass Blur | 27.37 | 27.32 | 27.52 | 27.54 | 27.56 | 27.64 | 27.75 | 27.98 | 28.09 | 28.23 |
| Motion Blur | 49.28 | 49.27 | 49.34 | 49.39 | 49.46 | 49.57 | 49.59 | 49.62 | 49.55 | 49.59 |
| Zoom Blur | 41.96 | 42.38 | 43.04 | 43.71 | 43.82 | 44.58 | 44.32 | 43.92 | 43.97 | 44.07 |
| Snow | 43.49 | 43.49 | 43.54 | 43.55 | 43.61 | 43.68 | 43.70 | 43.70 | 43.69 | 43.82 |
| Frost | 47.23 | 47.30 | 47.37 | 47.29 | 47.34 | 47.38 | 47.42 | 47.46 | 47.51 | 47.53 |
| Fog | 47.78 | 47.73 | 47.84 | 48.01 | 48.26 | 48.36 | 48.50 | 48.74 | 48.88 | 49.03 |
| Brightness | 57.47 | 57.48 | 57.59 | 57.63 | 57.66 | 57.70 | 57.76 | 57.83 | 57.89 | 57.98 |
| Contrast | 15.65 | 15.76 | 15.93 | 16.17 | 16.44 | 17.10 | 17.62 | 17.80 | 17.84 | 17.89 |
| Elastic Transform | 45.54 | 46.23 | 46.69 | 47.09 | 47.57 | 47.96 | 48.20 | 48.42 | 48.52 | 48.67 |
| Pixelate | 46.77 | 46.92 | 47.01 | 47.14 | 47.18 | 47.33 | 47.31 | 47.46 | 47.46 | 47.58 |
| JPEG Compression | 49.72 | 49.88 | 49.96 | 50.17 | 50.34 | 50.31 | 50.41 | 50.57 | 50.73 | 50.92 |

Table 37: Test accuracy (%) under different $\lambda_1$ values ($\lambda_2 = 0.7$) on TinyImageNet-C dataset.

| Corruption | $\lambda_1$ | | | | | | | | | |
|---|---|---|---|---|---|---|---|---|---|---|
| | 0.10 | 0.20 | 0.30 | 0.40 | 0.50 | 0.60 | 0.70 | 0.80 | 0.90 | 1.00 |
| Gaussian Noise | 24.18 | 24.13 | 23.95 | 23.92 | 23.98 | 24.06 | 23.94 | 23.87 | 23.79 | 23.65 |
| Shot Noise | 30.05 | 30.10 | 30.26 | 30.58 | 30.79 | 30.74 | 30.97 | 31.16 | 31.27 | 31.39 |
| Impulse Noise | 21.88 | 22.16 | 22.25 | 22.45 | 22.63 | 22.66 | 22.88 | 23.10 | 23.37 | 23.44 |
| Defocus Blur | 39.95 | 40.29 | 40.46 | 40.94 | 41.26 | 41.92 | 42.45 | 42.83 | 43.14 | 43.72 |
| Glass Blur | 27.28 | 27.50 | 27.50 | 27.50 | 27.63 | 27.75 | 27.89 | 27.98 | 28.09 | 28.13 |
| Motion Blur | 49.37 | 49.36 | 49.37 | 49.51 | 49.56 | 49.60 | 49.59 | 49.53 | 49.55 | 49.58 |
| Zoom Blur | 42.31 | 42.81 | 43.44 | 43.61 | 44.12 | 44.57 | 43.96 | 43.93 | 43.94 | 44.02 |
| Snow | 43.44 | 43.46 | 43.58 | 43.55 | 43.68 | 43.72 | 43.68 | 43.70 | 43.77 | 43.82 |
| Frost | 47.31 | 47.37 | 47.29 | 47.30 | 47.34 | 47.36 | 47.42 | 47.48 | 47.49 | 47.48 |
| Fog | 47.72 | 47.76 | 47.95 | 48.09 | 48.31 | 48.42 | 48.57 | 48.73 | 48.90 | 49.00 |
| Brightness | 57.52 | 57.59 | 57.60 | 57.64 | 57.71 | 57.77 | 57.81 | 57.91 | 57.98 | 57.99 |
| Contrast | 15.75 | 15.80 | 16.04 | 16.19 | 16.57 | 17.28 | 17.58 | 17.77 | 17.86 | 17.93 |
| Elastic Transform | 46.07 | 46.45 | 46.87 | 47.41 | 47.69 | 47.98 | 48.22 | 48.39 | 48.50 | 48.72 |
| Pixelate | 46.86 | 46.99 | 47.13 | 47.17 | 47.31 | 47.31 | 47.42 | 47.46 | 47.55 | 47.66 |
| JPEG Compression | 49.81 | 49.92 | 50.04 | 50.24 | 50.39 | 50.25 | 50.32 | 50.62 | 50.87 | 50.98 |

Table 38: Test accuracy (%) under different $\lambda_1$ values ($\lambda_2 = 0.8$) on TinyImageNet-C dataset.

| Corruption | $\lambda_1$ | | | | | | | | | |
|---|---|---|---|---|---|---|---|---|---|---|
| | 0.10 | 0.20 | 0.30 | 0.40 | 0.50 | 0.60 | 0.70 | 0.80 | 0.90 | 1.00 |
| Gaussian Noise | 24.08 | 23.90 | 23.88 | 23.92 | 23.94 | 23.91 | 23.79 | 23.70 | 23.54 | 23.53 |
| Shot Noise | 29.96 | 30.05 | 30.33 | 30.55 | 30.63 | 30.69 | 30.93 | 31.12 | 31.18 | 31.17 |
| Impulse Noise | 21.99 | 22.22 | 22.37 | 22.53 | 22.61 | 22.67 | 22.85 | 23.06 | 23.23 | 23.38 |
| Defocus Blur | 40.04 | 40.20 | 40.66 | 41.04 | 41.47 | 42.14 | 42.51 | 42.94 | 43.29 | 43.83 |
| Glass Blur | 27.44 | 27.41 | 27.46 | 27.52 | 27.68 | 27.79 | 27.87 | 27.95 | 27.99 | 28.06 |
| Motion Blur | 49.37 | 49.41 | 49.47 | 49.52 | 49.60 | 49.60 | 49.53 | 49.57 | 49.55 | 49.58 |
| Zoom Blur | 42.59 | 43.17 | 43.59 | 43.87 | 44.52 | 44.04 | 43.78 | 43.86 | 43.83 | 44.02 |
| Snow | 43.47 | 43.53 | 43.57 | 43.62 | 43.69 | 43.71 | 43.70 | 43.73 | 43.75 | 43.78 |
| Frost | 47.32 | 47.26 | 47.27 | 47.33 | 47.35 | 47.39 | 47.40 | 47.45 | 47.43 | 47.46 |
| Fog | 47.73 | 47.88 | 48.03 | 48.15 | 48.32 | 48.47 | 48.62 | 48.77 | 48.90 | 49.00 |
| Brightness | 57.53 | 57.58 | 57.61 | 57.70 | 57.74 | 57.80 | 57.91 | 57.96 | 57.98 | 58.10 |
| Contrast | 15.72 | 15.94 | 16.14 | 16.36 | 16.60 | 17.38 | 17.61 | 17.73 | 17.85 | 17.83 |
| Elastic Transform | 46.26 | 46.65 | 47.04 | 47.52 | 47.83 | 48.06 | 48.21 | 48.39 | 48.52 | 48.83 |
| Pixelate | 47.00 | 47.12 | 47.13 | 47.26 | 47.34 | 47.38 | 47.44 | 47.53 | 47.62 | 47.74 |
| JPEG Compression | 49.91 | 50.03 | 50.09 | 50.25 | 50.26 | 50.23 | 50.46 | 50.76 | 50.92 | 50.99 |

Table 39: Test accuracy (%) under different $\lambda_1$ values ($\lambda_2 = 0.9$) on TinyImageNet-C dataset.

| Corruption | $\lambda_1$ | | | | | | | | | |
|---|---|---|---|---|---|---|---|---|---|---|
| | 0.10 | 0.20 | 0.30 | 0.40 | 0.50 | 0.60 | 0.70 | 0.80 | 0.90 | 1.00 |
| Gaussian Noise | 23.85 | 23.83 | 23.85 | 23.86 | 23.80 | 23.72 | 23.64 | 23.47 | 23.46 | 23.40 |
| Shot Noise | 29.91 | 30.11 | 30.31 | 30.52 | 30.52 | 30.64 | 30.82 | 30.94 | 30.99 | 30.96 |
| Impulse Noise | 22.12 | 22.28 | 22.43 | 22.55 | 22.63 | 22.73 | 22.89 | 23.06 | 23.25 | 23.32 |
| Defocus Blur | 40.09 | 40.45 | 40.79 | 41.17 | 41.64 | 42.14 | 42.74 | 43.05 | 43.34 | 43.87 |
| Glass Blur | 27.50 | 27.47 | 27.44 | 27.60 | 27.74 | 27.86 | 27.90 | 28.00 | 28.06 | 27.88 |
| Motion Blur | 49.41 | 49.48 | 49.52 | 49.58 | 49.56 | 49.54 | 49.55 | 49.51 | 49.53 | 49.54 |
| Zoom Blur | 43.03 | 43.34 | 43.70 | 44.29 | 44.36 | 43.83 | 43.85 | 43.82 | 43.86 | 44.01 |
| Snow | 43.52 | 43.56 | 43.60 | 43.67 | 43.69 | 43.67 | 43.71 | 43.79 | 43.76 | 43.71 |
| Frost | 47.21 | 47.28 | 47.27 | 47.32 | 47.38 | 47.39 | 47.46 | 47.42 | 47.42 | 47.41 |
| Fog | 47.84 | 48.02 | 48.15 | 48.23 | 48.35 | 48.49 | 48.70 | 48.79 | 48.91 | 48.99 |
| Brightness | 57.59 | 57.62 | 57.71 | 57.71 | 57.81 | 57.89 | 57.94 | 57.95 | 58.09 | 58.14 |
| Contrast | 15.82 | 16.02 | 16.25 | 16.49 | 16.70 | 17.44 | 17.62 | 17.65 | 17.69 | 17.73 |
| Elastic Transform | 46.50 | 46.76 | 47.30 | 47.70 | 47.87 | 48.06 | 48.25 | 48.40 | 48.56 | 48.90 |
| Pixelate | 47.06 | 47.18 | 47.23 | 47.31 | 47.34 | 47.43 | 47.48 | 47.58 | 47.73 | 47.71 |
| JPEG Compression | 49.96 | 50.00 | 50.09 | 50.31 | 50.27 | 50.38 | 50.64 | 50.85 | 50.97 | 50.99 |

Table 40: Test accuracy (%) under different $\lambda_1$ values ($\lambda_2 = 1.0$) on TinyImageNet-C dataset.

| Corruption | $\lambda_1$ | | | | | | | | | |
|---|---|---|---|---|---|---|---|---|---|---|
| | 0.10 | 0.20 | 0.30 | 0.40 | 0.50 | 0.60 | 0.70 | 0.80 | 0.90 | 1.00 |
| Gaussian Noise | 23.78 | 23.74 | 23.75 | 23.72 | 23.67 | 23.60 | 23.48 | 23.40 | 23.31 | 23.24 |
| Shot Noise | 29.87 | 30.17 | 30.31 | 30.43 | 30.43 | 30.59 | 30.76 | 30.79 | 30.88 | 30.84 |
| Impulse Noise | 22.20 | 22.20 | 22.53 | 22.54 | 22.67 | 22.76 | 22.83 | 23.00 | 23.12 | 23.25 |
| Defocus Blur | 40.30 | 40.58 | 40.94 | 41.32 | 41.94 | 42.34 | 42.74 | 43.07 | 43.41 | 43.90 |
| Glass Blur | 27.50 | 27.46 | 27.62 | 27.74 | 27.74 | 27.85 | 27.89 | 27.88 | 27.87 | 27.83 |
| Motion Blur | 49.49 | 49.51 | 49.57 | 49.56 | 49.54 | 49.53 | 49.47 | 49.52 | 49.47 | 49.48 |
| Zoom Blur | 43.24 | 43.63 | 43.96 | 44.41 | 43.94 | 43.81 | 43.84 | 43.78 | 43.84 | 43.92 |
| Snow | 43.60 | 43.62 | 43.61 | 43.68 | 43.68 | 43.73 | 43.73 | 43.76 | 43.74 | 43.68 |
| Frost | 47.23 | 47.24 | 47.29 | 47.37 | 47.39 | 47.40 | 47.45 | 47.38 | 47.38 | 47.42 |
| Fog | 47.98 | 48.11 | 48.19 | 48.24 | 48.41 | 48.54 | 48.68 | 48.79 | 48.86 | 49.01 |
| Brightness | 57.66 | 57.69 | 57.70 | 57.82 | 57.87 | 57.95 | 57.97 | 58.09 | 58.13 | 58.19 |
| Contrast | 15.91 | 16.16 | 16.27 | 16.54 | 16.84 | 17.45 | 17.57 | 17.65 | 17.65 | 17.69 |
| Elastic Transform | 46.59 | 47.11 | 47.50 | 47.73 | 47.86 | 48.10 | 48.30 | 48.39 | 48.60 | 48.99 |
| Pixelate | 47.14 | 47.19 | 47.34 | 47.32 | 47.40 | 47.44 | 47.59 | 47.65 | 47.69 | 47.69 |
| JPEG Compression | 49.93 | 50.07 | 50.23 | 50.25 | 50.30 | 50.46 | 50.82 | 50.90 | 50.94 | 51.02 |

Table 41: Performance on LAION-C dataset for CLIP-DR and CLIP-OT (ViT-B-32). **Bold** indicates the best result.

| Methods | LAION-C | | | | | | |
|---|---|---|---|---|---|---|---|
| | mosaic | geometric | glitched | luminance | stickers | vertical lines | Avg. |
| CLIP | 15.33 | 21.48 | 19.12 | 7.93 | 6.18 | 8.45 | 13.08 |
| CLIP-OT | 15.62 | 31.69 | 21.09 | 13.75 | **7.57** | 15.62 | 17.56 |
| CLIP-DR | **16.24** | **38.43** | **21.39** | **14.65** | 7.52 | **15.89** | **19.02** |

Table 42: Performance on CIFAR-10-C. **Bold** indicates the best result.

| Methods | Noises | | | | | | | | | | | | | | | |
|---|---|---|---|---|---|---|---|---|---|---|---|---|---|---|---|---|
| CIFAR-10-C | Gaussian | Shot | Impulse | Defocus | Glass | Motion | Zoom | Snow | Frost | Fog | Brightness | Contrast | Elastic | Pixelate | JPEG | Avg. |
| w/ $\phi$ optimized | 64.91 | 67.34 | 62.12 | 82.09 | 67.48 | 81.49 | 83.55 | 84.01 | 83.4 | 82.33 | 89.57 | 84.89 | 76.05 | 76.92 | 70.44 | 77.10 |
| w/o $\phi$ optimized | **65.25** | **67.38** | **62.24** | **82.4** | **67.69** | **81.64** | **83.6** | **84.1** | **83.45** | **82.38** | **89.75** | **84.98** | **76.16** | **77.11** | **70.5** | **77.24** |

Table 43: Accuracy (%) on fine-grained datasets. **Bold** indicates the best result.

| Methods | Datasets | | | |
|---|---|---|---|---|
| | EuroSAT | Aircraft | DTD | Pets |
| CLIP-OT | 34.87 | 26.50 | 43.59 | 77.30 |
| CLIP-DR (w/o $\mathcal{L}_{IM}$) | 35.47 | 26.56 | 43.63 | 77.55 |
| CLIP-DR | **35.94** | **26.63** | **43.70** | **77.85** |

Table 44: ECE value on CIFAR-10-C for othet TTA techniques with the proposed method.

| Methods | Noises | | | | | | | | | | | | | | |
|---|---|---|---|---|---|---|---|---|---|---|---|---|---|---|---|
| | Gaussian | Shot | Impulse | Defocus | Glass | Motion | Zoom | Snow | Frost | Fog | Brightness | Contrast | Elastic | Pixelate | JPEG |
| TENT | 0.12 | 0.12 | 0.05 | 0.02 | 0.04 | 0.02 | 0.01 | 0.03 | 0.03 | 0.03 | 0.03 | 0.04 | 0.02 | 0.03 | 0.04 |
| TENT (w/ ours) | 0.08 | 0.08 | 0.05 | 0.02 | 0.02 | 0.01 | 0.01 | 0.03 | 0.03 | 0.03 | 0.03 | 0.04 | 0.02 | 0.02 | 0.03 |
| TPT | 0.05 | 0.04 | 0.03 | 0.06 | 0.03 | 0.02 | 0.03 | 0.06 | 0.07 | 0.06 | 0.07 | 0.09 | 0.04 | 0.03 | 0.03 |
| TPT (w/ ours) | 0.06 | 0.04 | 0.03 | 0.05 | 0.03 | 0.02 | 0.02 | 0.05 | 0.06 | 0.05 | 0.06 | 0.08 | 0.03 | 0.02 | 0.02 |

Table 45: ECE value on CIFAR-100-C for othet TTA techniques with the proposed method.

| Methods | Noises | | | | | | | | | | | | | | |
|---|---|---|---|---|---|---|---|---|---|---|---|---|---|---|---|
| | Gaussian | Shot | Impulse | Defocus | Glass | Motion | Zoom | Snow | Frost | Fog | Brightness | Contrast | Elastic | Pixelate | JPEG |
| TENT | 0.24 | 0.19 | 0.1 | 0.06 | 0.31 | 0.03 | 0.05 | 0.04 | 0.03 | 0.03 | 0.06 | 0.04 | 0.05 | 0.02 | 0.02 |
| TENT (w/ ours) | 0.10 | 0.08 | 0.05 | 0.06 | 0.26 | 0.04 | 0.05 | 0.04 | 0.04 | 0.03 | 0.06 | 0.04 | 0.03 | 0.02 | 0.02 |
| TPT | 0.12 | 0.09 | 0.05 | 0.07 | 0.23 | 0.05 | 0.06 | 0.06 | 0.05 | 0.04 | 0.07 | 0.05 | 0.03 | 0.02 | 0.02 |
| TPT (w/ ours) | 0.07 | 0.06 | 0.03 | 0.08 | 0.19 | 0.05 | 0.07 | 0.06 | 0.05 | 0.05 | 0.07 | 0.05 | 0.02 | 0.02 | 0.02 |

