# OpenReview forum: "CLIP-TTA: Robust Test-Time Adaptation via Dual Regularization Beyond Optimal Transport"
_ICLR.cc/2026/Conference — Submitted to ICLR 2026_

### Official Review · Reviewer_dKuy · 2025-10-29

**Soundness:** 3
**Presentation:** 3
**Contribution:** 3
**Rating:** 6
**Confidence:** 3

**Summary:**

This paper introduces a novel method, CLIP-TTA, which adapts CLIP models for downstream tasks without requiring labeled data. To leverage unlabeled data during testing, CLIP-TTA employs optimal transport for pseudo-labeling and incorporates two regularization losses to prevent pseudo-label collapse. Experimental results demonstrate that CLIP-TTA enhances the performance of CLIP under distribution shifts.

**Strengths:**

1. The problem that how to utilize unlabeled data is central to several areas of machine learning, such as unsupervised and semi-supervised learning. A common approach is self-training, which alternates between assigning pseudo-labels and training the model with confident data. In our view, this paper improves the self-training framework by integrating optimal transport into pseudo-labeling, which is an interesting and inspiring idea.

2. Building upon the improved self-training framework, the paper introduces two regularization losses which use the confidence (or entropy) of predicted sample to prevent collapse. These regularization methods are straightforward and conceptually sound.

3. Figure 3 (Left) shows that the proposed method is not only effective but also efficient.

**Weaknesses:**

1. In my opinion, this paper is somewhat incremental and similar with CLIP-OT [1]. While it adds two regularization losses to improve the pseudo-labeling process, the novelty feels reduced compared to CLIP-OT. I would appreciate a more detailed comparison to highlight the differences between this work and CLIP-OT.

2. In Figure 5, hyper-parameters $\lambda_1$ and $\lambda_2$  have minimal impact on the average accuracy of CIFAR-10-C and CIFAR-100-C. I suggest the authors provide further explanation on the effectiveness of the proposed regularization losses.

3. CLIP-TTA updates only the visual encoder $\theta$ during the adaptation process, assuming that distribution shifts affect only the images. This assumption limits the scope of application for this method.

4. If I understand correctly, both regularization losses are computed on the output logits. The additional lines in Figure 2 seem unnecessary and make the framework more complex and difficult to understand.

[1] Words Matter: Leveraging Individual Text Embeddings for Code Generation in CLIP Test-Time Adaptation, ArXiv 24

**Questions:**

See my questions in weakness.

---

> ### Author Response · Authors · 2025-11-17
> **Rebuttal (part I)**
>
> **We thank the reviewer for the constructive comments. We provided the revisions and additional experimental results for Weakness (1,2,3,4) as follows.**
>
> **W1**. We agree that the contribution is related to these two losses. However, the key difference is that the proposed method identifies and solves a critical challenge in CLIP-OT (i.e., low reliability in OT-based pseudo-labeling, especially on large-scale datasets and cross-domain setting), transforming it from a potentially unstable method into a robust TTA approach. Our experiments show that CLIP-OT degrades the performance of the CLIP model below zero-shot levels (e.g., on ImageNet-C: 37.89% \to 35.45%, **Please see Reviewer 9utf reply part I**). In CLIP-DR, we propose two mechanisms (i.e., \mathcal{L}_{cos} and \mathcal{L}_{IM}) that reverse this collapse within the OT optimization framework.
>
> **W2**. We agree that the hyperparameters \lambda_1 and \lambda_2 have a minimal impact on Avg. accuracy for CIFAR-10-C and CIFAR-100-C. The primary role of \mathcal{L}_{cos} and \mathcal{L}_{IM} is to prevent model collapse and reduce over-confidence, which is critical for reliable adaptation. On more complex datasets like TinyImageNet-C and ImageNet-C (Please see **Reviewer 9utf reply part I and part III**), CLIP-OT suffers from performance degradation (35.45% vs. 37.89% for zero-shot), while CLIP-DR avoids this collapse and achieves a higher Avg. accuracy of 39.03%. Similar situation can be found using different backbones such as EVA02-CLIP (**Please see Reviewer 9utf reply part II**). This demonstrates that the proposed losses are essential for stability under challenging distribution shifts.
>
> **W3**. The motivation of updating the visual encoder alone is based on the common and practical assumption in TTA that the primary source of distribution shift derives from the image domain (e.g., corruptions, style changes) as suggested in CLIP-OT [1]. In practical, the text embeddings for each class are fixed during initialization, while the model updates the parameters in the image encoders to reduce domain shifts. Updating the text encoder with unlabeled noisy test batches introduces a risk of corrupting the semantic meaning of the text prototypes, which are crucial as reliable anchors for classification. As suggested, we provide the experimental results on CIFAR-10-C (with text encoders optimized) as follows. It can be seen that optimizing the text encoder leads to negative effects. We will explore other techniques (e.g., learnable prefix parameters) to adapt the text encoder in future work.
>
> ## Performance on CIFAR-10-C (ViT-B-32)
> | Methods | Gaussian | Shot | Impulse | Defocus | Glass | Motion | Zoom | Snow | Frost | Fog | Brightness | Contrast | Elastic | Pixelate | JPEG | Avg. |
> | :--- | :---: | :---: | :---: | :---: | :---: | :---: | :---: | :---: | :---: | :---: | :---: | :---: | :---: | :---: | :---: | :---: |
> | w/ $\phi$ optimized | 64.91 | 67.34 | 62.12 | 82.09 | 67.48 | 81.49 | 83.55 | 84.01 | 83.40 | 82.33 | 89.57 | 84.89 | 76.05 | 76.92 | 70.44 | 77.10 |
> | w/o $\phi$ optimized | **65.25** | **67.38** | **62.24** | **82.40** | **67.69** | **81.64** | **83.60** | **84.10** | **83.45** | **82.38** | **89.75** | **84.98** | **76.16** | **77.11** | **70.50** | **77.24** |
>
> **W4**. We agree that both regularization losses require the output logits from the CLIP model. As suggested, we revised the Figure 2. Please view it in the revised manuscript.
>
> [1]. Words Matter: Leveraging Individual Text Embeddings for Code Generation in CLIP Test-Time Adaptation, ArXiv 24

---

> ### Author Response · Authors · 2025-11-26
>
> Dear reviewer dKuy,
>
> Thank you very much for your constructive comments on our work. We've made every effort to address the concerns raised. As the discussion period is nearing its conclusion, could we kindly inquire if you have any remaining questions or concerns? Thanks for your efforts in reviewing our work, and we sincerely look forward to your reply.
>
> Best regards,
>
> Authors

---

### Official Review · Reviewer_9utf · 2025-10-30

**Soundness:** 2
**Presentation:** 3
**Contribution:** 2
**Rating:** 4
**Confidence:** 3

**Summary:**

This paper proposes a new test-time adaptation method named CLIP-TTA for VLMs that addresses the unreliable pseudo-labels of prior work like CLIP-OT. The authors introduce two losses: a cosine similarity loss to align image logits with text prototypes and an information maximization regularizer to encourage confident and diverse predictions. Experiments show that CLIP-TTA improves robustness against corruptions and domain shifts over current methods.

**Strengths:**

1.	The method is presented clear and easy to understand. The component of CLIP-TTA is clearly organized by each section with correct and appropriate reference.

2.	The paper demonstrates consistent performance gains over the primary baseline CLIP-OT across a wide array of benchmarks.

3.	The authors provide detailed model analysis including ablation study, sensitivity of each different parameters and experimental settings.

**Weaknesses:**

1.	The paper's primary motivation is to solve the "over-confidence" problem (high ECE) of the CLIP-OT baseline. However, the proposed core components: a cosine similarity loss and an information maximization loss, lack a direct theoretical link to this stated goal. The direct objective of $\mathcal{L}\_{cos}$ is to align features, while $\mathcal{L}\_{IM}$ aims to promote confident and diverse predictions to prevent model collapse. The paper fails to clearly articulate the theoretical chain of reasoning for why "alignment" and "preventing collapse" directly solve the problem of over-confidence, making the connection feel indirect and insufficiently supported.

2.	The experimental validation omits several standard and challenging benchmarks. To better assess robustness, evaluation on ImageNet-C[1] is necessary. Furthermore, to test generalization on different data types, the paper would benefit from including fine-grained classification datasets from the CLIP zero-shot suite, such as the DTD[2] or EuroSAT[3].

3.	All experiments are conducted solely on the CLIP (ViT-B/32) backbone. To demonstrate the generalizability of the proposed dual-regularization approach, it should be tested on other vision-language model architectures, such as BLIP, to prove that the method is not just tailored to CLIP.

4.	The paper's core methodological contribution is arguably incremental. The problem formulation (Eq. 1) is standard, and the optimal transport mechanism (Eqs. 2-8) is adopted directly from the CLIP-OT baseline. The primary novelty lies in adding two existing loss functions,  constitutes a limited conceptual advance.

[1] Hendrycks, Dan, and Thomas Dietterich. "Benchmarking neural network robustness to common corruptions and perturbations." arXiv preprint arXiv:1903.12261 (2019).

[2] Cimpoi, Mircea, et al. "Describing textures in the wild." Proceedings of the IEEE conference on computer vision and pattern recognition. 2014.

[3] Helber, Patrick, et al. "Eurosat: A novel dataset and deep learning benchmark for land use and land cover classification." IEEE Journal of Selected Topics in Applied Earth Observations and Remote Sensing 12.7 (2019): 2217-2226.

**Questions:**

We would like to draw your attention to the recent, highly relevant paper by Lafon et al. (2025), "Cliptta: Robust contrastive vision-language test-time adaptation" (arXiv:2507.14312). (Already Cited in Section 2 in original paper)

1.	The title "Cliptta" used by Lafon et al. is practically identical to your proposed "CLIP-TTA". Given this, are you concerned that this will create significant ambiguity and confusion for future researchers when citing and attempting to differentiate these two distinct methods?

2.	Lafon et al. argue that gradient-based TTA can "degrade learned knowledge," and for this reason, they propose a gradient-free solution. How does your dual regularization specifically prevent this degradation?

---

> ### Author Response · Authors · 2025-11-16
> **Rebuttal (part I)**
>
> **We thank the reviewer for the constructive comments. We provided the revisions and additional experimental results for Weakness (1,2,4) and Questions (1, 2)**
>
> **W1**. As suggested, we revised the methodology to connect these components as follows. Firstly, over-confidence (i.e., high ECE) arises from noisy pseudo-labels and collapsed predictions. \mathcal{L}_{cos} reduces pseudo-label noise by aligning features with correct prototypes, minimizing confidently wrong predictions. Furthermore, \mathcal{L}_{IM} directly penalizes over-confidence by enforcing diverse outputs via entropy maximization while maintaining per-sample confidence, which is a regularization for calibration. Together, they jointly optimize the model to avoid feature misalignment and distribution collapse, providing promising calibration ability.
>
> **W2**. We provide the experimental results on ImageNet-C and several cross-domain datasets. Experiments on the ImageNet-C show that CLIP-OT suffers from negative adaptation, degrading the performance from 37.89% (CLIP zero-shot) to 35.45%. This demonstrates that the pseudo-labels generated by OT can be highly unreliable. CLIP-DR reduces performance degradation and provides a higher Avg. accuracy of 39.03%, outperforming CLIP-OT by 3.58%. This proves that our dual regularization framework is essential for stabilizing the adaptation process and preventing error accumulation. Similar for cross-domain datasets, while the gains of CLIP-DR on some datasets are modest, the key finding is that CLIP-DR never decreases the original performance, unlike CLIP-OT which can cause degradation (e.g., on Imagenet-R, Pets). This reliability is crucial for real-world TTA applications. Please see Table below for detailed results.
>
> ## Accuracy (%) on ImageNet-C for CLIP-DR and baselines (ViT-B-32)
>
> | Methods | Gaussian | Shot | Impulse | Defocus | Glass | Motion | Zoom | Snow | Frost | Fog | Brightness | Contrast | Elastic | Pixelate | JPEG | Avg. |
> | :--- | :---: | :---: | :---: | :---: | :---: | :---: | :---: | :---: | :---: | :---: | :---: | :---: | :---: | :---: | :---: | :---: |
> | CLIP | 35.40 | 35.75 | 35.96 | 35.59 | 19.45 | 36.63 | 26.88 | 31.35 | 30.04 | **43.99** | **53.38** | **46.95** | **46.59** | **44.42** | **46.01** | 37.89 |
> | CLIP-OT | 31.56 | 32.08 | 33.00 | 32.50 | 19.65 | 31.73 | 27.65 | 30.53 | 28.94 | 40.29 | 52.51 | 43.22 | 44.15 | 40.73 | 43.29 | 35.45 |
> | CLIP-DR | **37.51** | **37.75** | **37.71** | **36.73** | **25.10** | **38.22** | **30.98** | **33.23** | **30.42** | 43.50 | 52.97 | 46.36 | 45.69 | 44.02 | 45.35 | **39.03** |
>
> ##  Performance on other datasets for CLIP-DR and CLIP-OT (ViT-B-32)
> | Methods | Imagenet-R | EuroSAT | Aircraft | Caltech-101 | ImageNetV2 | DTD | ImageNet-S | Pets |
> | :--- | :---: | :---: | :---: | :---: | :---: | :---: | :---: | :---: |
> | CLIP | 68.36 | 33.30 | 25.87 | 80.74 | 55.02 | 43.38 | 74.33 | 77.51 |
> | CLIP-OT | 68.26 | 34.87 | 26.50 | **80.80** | 55.19 | 43.59 | 73.91 | 77.30 |
> | CLIP-DR | **68.40** | **35.94** | **26.63** | 80.77 | **55.25** | **43.70** | **74.70** | **77.85** |
>
> **W4**. We agree that the two existing loss funcions are known in the related topic, and the contribution is related to these two losses. However, it diagnoses fundamental instabilities in OT-based pseudo-labeling and proposes mechanisms that resolve them within the OT optimization framework. Specifically, it uses \mathcal{L}_{cos} and \mathcal{L}_{IM} mechanisms to transform CLIP-OT from an unreliable method into a robust one, as CLIP-DR consistently prevents performance collapse (e.g., on ImageNet-C) and achieves superior robustness across diverse backbones.
>
> **Q1**. We agree that it will introduce confusion for future works. We revised the titile to CLIP-DR: Robust Test-Time Adaptation via Dual Regularization Beyond Optimal Transport.
>
> **Q2**. Thanks for this critical comment. In CLIP-DR, the dual regularization prevents the degradation of pre-trained knowledge by constraining gradient updates toward conservative directions. For example, \mathcal{L}_{cos} anchors image features to the frozen text prototypes, preventing feature drift by aligning updates with the original semantic space. Simultaneously, the \mathcal{L}_{IM} maintains prediction diversity, avoiding collapse into few classes. Unlike gradient-free approaches, the proposed dual approach ensures that each adaptation step preserves foundational knowledge while fitting the test data. This is validated by our consistent improvements over the zero-shot and TTA baselines across various benchmarks.
>
> **We will provide our response to Weakness W3 in a separate comment due to length constraints. The corresponding revisions in the manuscript are highlighted in blue.**

---

> ### Author Response · Authors · 2025-11-16
> **Rebuttal (part II)**
>
> **We thank the reviewer for the constructive comments. We provided the revisions and additional experimental results for Weakness (3) as follows.**
>
> **W3**. We provide the test accuracy using different backbones such as SigLIP and EVA02-CLIP. The results show that CLIP-DR provides higher Avg. accuracy compared to CLIP-OT with all backbones (e.g., 89.38% vs. 86.92% with EVA02-CLIP on CIFAR-10-C). These results confirm that the proposed dual-regularization principle is a general and effective strategy for robust test-time adaptation, not just being a mere tweak for CLIP model. Please see Table below for detailed results.}
>
> ## Accuracy (%) on CIFAR-C Datasets with Different Backbones
>
> | Methods | Gaussian | Shot | Impulse | Defocus | Glass | Motion | Zoom | Snow | Frost | Fog | Brightness | Contrast | Elastic | Pixelate | JPEG | Avg. |
> | :--- | :---: | :---: | :---: | :---: | :---: | :---: | :---: | :---: | :---: | :---: | :---: | :---: | :---: | :---: | :---: | :---: |
> | **--- CIFAR-10-C (SigLIP) ---** | | | | | | | | | | | | | | | | |
> | SigLIP | 13.27 | 14.80 | 20.52 | 54.34 | 23.36 | 46.00 | 52.90 | 25.49 | 32.06 | 68.74 | 28.35 | 78.53 | 38.36 | 36.84 | 26.69 | 37.35 |
> | CLIP-OT | 39.98 | 50.20 | 60.65 | 76.92 | 56.69 | 74.88 | 78.65 | 53.04 | 64.15 | 85.06 | 64.27 | **89.42** | 66.83 | 68.10 | 53.73 | 65.50 |
> | CLIP-DR | **40.34** | **51.35** | **61.34** | **77.69** | **57.39** | **75.20** | **79.71** | **54.10** | **66.33** | **85.74** | **66.12** | 89.04 | **67.91** | **70.42** | **54.92** | **66.51** |
> | **--- CIFAR-10-C (EVA02-CLIP) ---** | | | | | | | | | | | | | | | | |
> | EVA02-CLIP | 71.87 | 74.20 | 73.11 | 90.14 | 64.85 | 86.87 | 91.44 | 92.40 | 92.55 | 88.82 | 96.50 | 89.42 | 75.94 | 61.92 | 79.05 | 81.93 |
> | CLIP-OT | 81.18 | 76.36 | 80.70 | 90.73 | 83.54 | 89.53 | 91.07 | **92.53** | 87.54 | 82.46 | **96.87** | 94.28 | 83.73 | **89.96** | 83.41 | 86.92 |
> | CLIP-DR | **83.05** | **81.61** | **86.55** | **93.14** | **83.66** | **90.53** | **94.03** | 91.67 | **93.69** | **92.21** | 94.89 | **94.52** | **87.85** | 87.72 | **85.59** | **89.38** |
> | **--- CIFAR-100-C (SigLIP) ---** | | | | | | | | | | | | | | | | |
> | SigLIP | 2.78 | 2.97 | 4.51 | 22.68 | 6.85 | 17.49 | 24.04 | 5.26 | 8.57 | 37.98 | 6.82 | 46.08 | 10.78 | 10.01 | 5.88 | 14.18 |
> | CLIP-OT | 11.96 | 14.63 | 20.39 | 40.41 | 19.66 | 37.45 | 42.03 | 20.01 | 26.08 | 54.21 | 21.65 | 58.16 | 25.66 | 26.87 | 18.40 | 29.17 |
> | CLIP-DR | **12.89** | **15.71** | **21.12** | **41.32** | **20.05** | **38.19** | **43.04** | **21.03** | **26.72** | **55.10** | **22.82** | **59.23** | **26.27** | **27.58** | **18.73** | **29.99** |
> | **--- CIFAR-100-C (EVA02-CLIP) ---** | | | | | | | | | | | | | | | | |
> | EVA02-CLIP | 45.66 | 47.74 | 45.37 | **68.84** | 36.99 | 64.78 | **72.28** | **71.86** | **71.55** | 65.29 | **80.21** | 64.23 | 45.28 | 40.14 | 53.75 | 58.26 |
> | CLIP-OT | 55.44 | 52.94 | 56.89 | 66.54 | 54.35 | 63.15 | 67.70 | 66.00 | 68.22 | **65.64** | 72.79 | 64.68 | 57.15 | 62.43 | 59.37 | 62.22 |
> | CLIP-DR | **56.74** | **56.44** | **58.86** | 67.78 | **57.10** | **65.86** | 69.94 | 68.75 | 71.00 | **65.64** | 76.54 | **67.51** | **61.26** | **65.65** | **61.35** | **64.69** |
>
> **The corresponding revisions in the manuscript are highlighted in blue.**

---

> ### Author Response · Authors · 2025-11-17
> **Rebuttal (part III) ----- Follow-up: Additional results on LAION-C (Cross-Domain)**
>
> **We thank the reviewer for the constructive comments. This rebuttal is a follow-up for weakness 2.**
>
> As suggested, we further provide additional experimental results on a recent and challenging benchmark: LAION-C [1].  LAION-C has 6 domains, each domain holds different type of corruptions (e.g., mosaic). In each corruption, it has 16 classes. Table below reports the test accuracy on LAION-C. The results conclusively prove that the proposed dual regularization can enhance the performance within the OT optimization framework (e.g., a reliable performance gain (+1.46% in Avg.)).
>
> ## Performance on LAION-C dataset for CLIP-DR and CLIP-OT (ViT-B-32)
> | Methods | mosaic | geometric | glitched | luminance | stickers | vertical lines | Avg. |
> | :--- | :---: | :---: | :---: | :---: | :---: | :---: | :---: |
> | CLIP | 15.33 | 21.48 | 19.12 | 7.93 | 6.18 | 8.45 | 13.08 |
> | CLIP-OT | 15.62 | 31.69 | 21.09 | 13.75 | **7.57** | 15.62 | 17.56 |
> | CLIP-DR | **16.24** | **38.43** | **21.39** | **14.65** | **7.57** | **15.89** | **19.02** |
>
> [1] LAION-C: An Out-of-Distribution Benchmark for Web-Scale Vision Models. Fanfei Li et.al. ICML 2025.

---

> ### Comment · Reviewer_9utf · 2025-11-25
>
> Thank authors for the responses and additional experimental results. I have no more questions.

---

> > ### Author Response · Authors · 2025-11-26
> >
> > Dear reviewer 9utf,
> >
> > We really appreciate having the opportunity to address the reviewer’s concerns. Thank you once again for acknowledging our work and providing valuable comments.
> >
> > Best regards,
> >
> > Authors

---

### Official Review · Reviewer_P4FT · 2025-10-30

**Soundness:** 3
**Presentation:** 3
**Contribution:** 2
**Rating:** 4
**Confidence:** 4

**Summary:**

This paper proposes CLIP-TTA, a test-time adaptation (TTA) method designed to address the issue of unreliable pseudo-labels generated by the previous CLIP-OT approach.  CLIP-TTA introduces two key components: (1) a cosine similarity loss to align image features with textual prototypes, ensuring stable adaptation;  and (2) an information maximization regularizer to encourage confident and diverse predictions, preventing model collapse.  Extensive experiments across 7 benchmarks demonstrate competitive performance.

**Strengths:**

- The paper is well-written and easy to understand.

**Weaknesses:**

- The contribution of the paper is limited, as the proposed method framework is largely similar to OT-CLIP, with the addition of only two extra losses. Furthermore, there is no theoretical evidence provided to support how these losses contribute to the reduction of ECE.​
- The effectiveness of L_cos relies on the assumption of highly distinguishable text prototypes. However, in many fine-grained tasks, text templates are unable to differentiate between subclasses, which may lead to pushing the model toward incorrect priors.
- Sensitivity to Hyperparameters. The method shows extreme sensitivity to hyperparameters, as shown in Figure 5. Different tasks exhibit strong dependence on hyperparameter settings, which undermines the robustness claimed by the paper.

**Questions:**

- The manuscript should include validation of CLIP-TTA on cross-dataset and cross-domain benchmarks, as this would make the method's claims more convincing.
- The method should be tested on a broader range of TTA techniques (e.g., TPT) to demonstrate its effectiveness in reducing ECE, rather than being evaluated solely on OT-CLIP.

---

> ### Author Response · Authors · 2025-11-18
> **Rebuttal (part I)**
>
> **We thank the reviewer for the constructive comments. We provided the revisions and additional experimental results for Weakness (1,2,3)  and Questions (1)**
>
> **W1**. We agree that our contribution builds upon CLIP-OT, but its core value lies in identifying and solving its over-confidence, which causes high ECE. We validate the effectiveness of the proposed method on various benchmarks (**Please see Reviewer dKuy reply part I, Reviewer 9utf reply part I and the revised manuscript**). The findings suggest that using cosine alignment with information maximization regularization effectively guides the OT to learn robust features, providing a better performance.
>
> To explain why these losses help to reduce ECE, basically, high ECE arises from noisy pseudo-labels and distribution collapse. For \mathcal{L}_{cos}, it reduces pseudo-label noise by explicitly aligning image features with the correct virtual textual prototypes in the feature space. This ensures that high confidence is directed toward the correct class, directly mitigating the challenge of noisy pseudo-labels. For \mathcal{L}_{IM}, it regularizes the output distribution. The diversity term prevents collapse, while the confidence term sharpens predictions. This combination is a known information-theoretic principle for calibration, as established in prior work [1]. Their synergy provides a direct, theoretically-grounded solution to reduce ECE, which we have empirically validated.
>
> **W2**. We agree that the effectiveness of \mathcal{L}_{cos} relies on the discriminative text prototypes, which can be a limitation in extremely fine-grained datasets (e.g., Pets). To empirically evaluate the contribution of \mathcal{L}_{cos}, we validate the \mathcal{L}_{cos} without \mathcal{L}_{IM} on fine-grained datasets (Please see the Table below). As reported in the Table, introducing \mathcal{L}_{cos} consistently improves upon CLIP-OT across all datasets. This suggests that \mathcal{L}_{cos} does not shift the model toward incorrect priors but acts as a reliable regularizer within the OT framework. We will explore more discriminative text representations in future work to solve the limitation of \mathcal{L}_{cos}.
>
> ## Accuracy (%) on Fine-Grained Datasets （ViT-B-32)
> | Methods | EuroSAT | Aircraft | DTD | Pets |
> | :--- | :---: | :---: | :---: | :---: |
> | CLIP-OT | 34.87 | 26.50 | 43.59 | 77.30 |
> | CLIP-DR (w/o \mathcal{L}_{IM}) | 35.47 | 26.56 | 43.63 | 77.55 |
> | CLIP-DR | **35.94** | **26.63** | **43.70** | **77.85** |
>
> **W3**. We acknowledge that the optimal hyperparameters ($\lambda_1$ and $\lambda_2$) can vary across datasets, as shown in Figure 5. While the optimal values differ (e.g., CIFAR-10-C: (0.1, 0.5), CIFAR-100-C: (0.1, 0.1), TinyImageNet-C: (1.0, 0.1)), the performance degradation when using suboptimal hyperparameters is minimal (e.g., $<0.5\%$ on CIFAR series). In the paper, we claimed that ``a simple tuning strategy can be drawn from these findings: prioritize tuning $\lambda_1$ on large-scale datasets, while for smaller datasets, careful tuning of $\lambda_2$ is essential.'' Furthermore, both $\lambda_1$ and $\lambda_2$ to 0.5 can lead to suboptimal performance. However, the Avg. accuracy is still higher than 77.20\% for CIFAR-10-C (0.16\% higher than CLIP-OT), 47.18\% for CIFAR-100-C (0.16\% higher than CLIP-OT), and 39.27\% for TinyImageNet-C (3.71\% higher than CLIP-OT). This demonstrates that with a fixed and suboptimal hyperparameter set, CLIP-DR consistently outperforms the baseline, highlighting its robustness. We will explore dynamic hyperparameter selection in future work.
>
> **Q1**. As suggested, we validate the CLIP-DR on 10 cross-dataset and cross-domain benchmarks (**Please see Reviewer 9utf reply part I, II and III**). The results show that the proposed dual approach can reduce negative adaptation in CLIP-OT on many datasets such as ImageNet-C, Imagenet-R and Pets. These results are highlighted in blue in the revised manuscript.
>
> [1]. Patel, Kanil, et al. "Multi-class uncertainty calibration via mutual information maximization-based binning." ICLR (2021).

---

> ### Author Response · Authors · 2025-11-18
> **Rebuttal (part II)**
>
> **We thank the reviewer for the constructive comments. We provided the revisions and additional experimental results for Questions (2)**
>
> **Q2**. The proposed method specifically addresses the overconfidence in CLIP-OT, which is the primary goal of this study. Although this dual approach is not tailored to other TTA methods, we agree that it should be tested on more TTA techniques. As suggested, we add more results about ECE value for other TTA techniques (TENT, TPT) with the proposed dual approach (Please see the Table below). As reported, on CIFAR-10-C, adding the proposed method to TENT can reduce the ECE value for certain noises such as Gaussian (0.08 vs. 0.12) and Shot (0.08 vs. 0.12). Similar situation can be found using TPT (e.g., 0.06 vs. 0.07 with Frost). These results highlight the effectiveness of the proposed method in reducing ECE.
>
> ## ECE Value on CIFAR-10-C for Other TTA Techniques with the Proposed Method (ViT-B-32)
> | Methods | Gaussian | Shot | Impulse | Defocus | Glass | Motion | Zoom | Snow | Frost | Fog | Brightness | Contrast | Elastic | Pixelate | JPEG |
> | :--- | :---: | :---: | :---: | :---: | :---: | :---: | :---: | :---: | :---: | :---: | :---: | :---: | :---: | :---: | :---: |
> | TENT | 0.12 | 0.12 | 0.05 | 0.02 | 0.04 | 0.02 | 0.01 | 0.03 | 0.03 | 0.03 | 0.03 | 0.04 | 0.02 | 0.03 | 0.04 |
> | TENT (w/ ours) | 0.08 | 0.08 | 0.05 | 0.02 | 0.02 | 0.01 | 0.01 | 0.03 | 0.03 | 0.03 | 0.03 | 0.04 | 0.02 | 0.02 | 0.03 |
> | TPT | 0.05 | 0.04 | 0.03 | 0.06 | 0.03 | 0.02 | 0.03 | 0.06 | 0.07 | 0.06 | 0.07 | 0.09 | 0.04 | 0.03 | 0.03 |
> | TPT (w/ ours) | 0.06 | 0.04 | 0.03 | 0.05 | 0.03 | 0.02 | 0.02 | 0.05 | 0.06 | 0.05 | 0.06 | 0.08 | 0.03 | 0.02 | 0.02 |
>
> ## ECE Value on CIFAR-100-C for Other TTA Techniques with the Proposed Method (ViT-B-32)
> | Methods | Gaussian | Shot | Impulse | Defocus | Glass | Motion | Zoom | Snow | Frost | Fog | Brightness | Contrast | Elastic | Pixelate | JPEG |
> | :--- | :---: | :---: | :---: | :---: | :---: | :---: | :---: | :---: | :---: | :---: | :---: | :---: | :---: | :---: | :---: |
> | TENT | 0.24 | 0.19 | 0.10 | 0.06 | 0.31 | 0.03 | 0.05 | 0.04 | 0.03 | 0.03 | 0.06 | 0.04 | 0.05 | 0.02 | 0.02 |
> | TENT (w/ ours) | 0.10 | 0.08 | 0.05 | 0.06 | 0.26 | 0.04 | 0.05 | 0.04 | 0.04 | 0.03 | 0.06 | 0.04 | 0.03 | 0.02 | 0.02 |
> | TPT | 0.12 | 0.09 | 0.05 | 0.07 | 0.23 | 0.05 | 0.06 | 0.06 | 0.05 | 0.04 | 0.07 | 0.05 | 0.03 | 0.02 | 0.02 |
> | TPT (w/ ours) | 0.07 | 0.06 | 0.03 | 0.08 | 0.19 | 0.05 | 0.07 | 0.06 | 0.05 | 0.05 | 0.07 | 0.05 | 0.02 | 0.02 | 0.02 |

---

> ### Author Response · Authors · 2025-11-26
>
> Dear reviewer P4FT,
>
> Thank you very much for your constructive comments on our work. We've made every effort to address the concerns raised. As the discussion period is nearing its conclusion, could we kindly inquire if you have any remaining questions or concerns? Thanks for your efforts in reviewing our work, and we sincerely look forward to your reply.
>
> Best regards,
>
> Authors

---

> ### Comment · Reviewer_P4FT · 2025-11-27
>
> Thank you for the rebuttal.  At this stage, I am inclined to maintain my current score, primarily for the following reasons:
>
> - Limited contribution. Although the paper highlights that the synergy between the two losses can substantially reduce ECE, I still do not see an explicit connection between each loss and ECE—e.g. , whether they are positively or negatively correlated with calibration.  Without such analysis, the claimed mechanism remains insufficiently supported.
>
> - Marginal gains on fine-grained datasets. The results show that, compared to CLIP, CLIP-DR yields almost no improvement on fine-grained benchmarks.  This suggests that the proposed design may not be an optimal choice in certain scenarios.
>
> Given the above points, I recommend that the authors incorporate a discussion of C-TPT [1], whose analytical framework may provide useful insights for a more refined version of this work.
>
> [1] C-TPT: Calibrated Test-Time Prompt Tuning for Vision-Language Models via Text Feature Dispersion. ICLR 2024

---

> > ### Author Response · Authors · 2025-11-28
> >
> > **We thank the reviewer for the constructive comments. We provided the revisions and additional experimental results for the questions (1 & 2).**
> >
> > **Limited contribution**. We provide the analysis of \mathcal{L}_{cos} and \mathcal{L}_{IM} for reducing ECE. Specifically, the ECE can be formulated as follows:
> >
> > $\text{ECE} = \sum_{m=1}^{M} \frac{|B_m|}{n} |\text{acc}(B_m) - \text{conf}(B_m)|$
> >
> > where $B_m$ is the m-th bin. $n$ is the number of samples. A higher ECE value is dominated by confidently wrong samples residing in high-confidence, low-accuracy bins.
> >
> > Concerning \mathcal{L}_{IM}, it is defined as:
> >
> > $\mathcal{L}_{\text{IM}} = \frac{1}{|\mathcal{B}_t|} \sum_{\mathbf{x} \in \mathcal{B}_t} \underbrace{H(p(\mathbf{x}))}_{\text{confidence}} - \underbrace{H(\bar{p})}_{\text{diversity}}$
> >
> > The first term, namely confidence regularization, it encourages the model to make more confident predictions for each sample. This helps to reduce under-confidence, ensuring that predictions in medium-confidence bins are meaningful and not due to model uncertainty.
> >
> > The second term, namely diversity regularization, it controls over-confidence of all predictions through maximizing $H(\bar{p})$. This directly penalizes the model if it becomes over-confident on average (e.g., always predicting a few classes with high probability), reducing the number of samples that are incorrectly assigned to high-confidence bins.
> >
> > By jointly optimizing these two opposing objectives, \mathcal{L}_{IM} strikes a balance that prevents both under-confident and over-confident predictions, leading to a better-calibrated model and a lower ECE.
> >
> > Considering \mathcal{L}_{cos}, it is represented as:
> >
> > $\mathcal{L}_{cos} \,=\, \sum\limits_{j=1}^{B} {\text{exp}(C(\mathbf{x}_j)-C_0)\cdot (1-s_{\mathbf{O},{{{\mathbf{\bar{f}}}_{\hat y}}}}) \cdot \mathbb{I}\{C(\mathbf{x}_j)\ge C_0\}}$
> >
> > where $s$ is defined as:
> >
> > $s_{{\mathbf{O}},{{{\mathbf{\bar{f}}}_{\hat y}}}} = \frac{\langle{\mathbf{O}}, {{\mathbf{\bar{f}}}_{\hat y}} \rangle}{\|{\mathbf{O}}\| \!\cdot\!\|{{\mathbf{\bar{f}}}_{\hat y}}\|}\,,\, {\text{where}}\, \,{\mathbf{\bar{f}}_{\hat y}} = {\tr{\mathbf{t}_{km}}}{\mathbf{t}_{km}^{\hat{y}}}\,, \, \mathbf{O}=\mathbf{z}_i^\top\mathbf{t}_k/\tau$
> >
> > Specifically, it will push the image features toward the pseudo-text prototypes. Thus, it will change the logits $s$ with the following two points: 1) increase the number of correct predicted samples, and 2) decrease the incorrect predictions. This reduces the number of confidently wrong samples. These samples are the primary contributors to high ECE, as they cause large $|\text{acc}(B_m) - \text{conf}(B_m)|$ in high-confidence bins. By minimizing \mathcal{L}_{cos}, we systematically reduce this miscalibration.
> >
> > In summary, \mathcal{L}_{cos} and \mathcal{L}_{IM} form a synergistic framework: \mathcal{L}_{cos} works at the feature level to reduce the source of confident errors (noisy pseudo-labels), while \mathcal{L}_{IM} operates at the output level to ensure a well-calibrated probability distribution. This theoretical framework is validated by our empirical results.
> >
> > **Marginal gains on fine-grained datasets.** We agree that the performance gains on fine-grained datasets are modest compared to the more substantial improvements observed on corruption benchmarks like ImageNet-C, LAION-C, TinyImageNet-C. This result is consistent with the inherent limitation of CLIP textual space. The discriminative ability of CLIP-DR is build on the original semantic representation of CLIP pre-trained text encoder. In fine-grained tasks (e.g., distinguishing bird or car), the text prototypes for different classes are often very similar in the embedding space. Our method, which operates within this space, can provide a better refinement of the features to the extent that the text prototypes are separable. It never degrades performance below the zero-shot baseline, unlike CLIP-OT which can be unstable. Overall, CLIP-DR yields the highest performance on 16 datasets, especially on large-scale and diverse dataset such as ImageNet-C and LAION-C. It provides a robust and reliable default choice that provides performance gains on common distribution shifts (corruptions, domain shifts) while remaining safe and slightly beneficial on more challenging fine-grained tasks.
> >
> > **The corresponding revisions in the manuscript are highlighted in blue.**.

---

> ### Author Response · Authors · 2025-11-28
>
> **We thank the reviewer for the constructive comments. We provided the revisions and additional experimental results for the question 3**.
>
> **Given the above points, I recommend that the authors incorporate a discussion of C-TPT [1], whose analytical framework may provide useful insights for a more refined version of this work.**.
>
> In [1], they propose C-TPT (Calibrated Test-Time Prompt Tuning), sharing a similar goal of improving model calibration during TTA for CLIP without requiring labeled data. Specifically, C-TPT identifies that the choice of prompts considerably impacts calibration in CLIP, and proposes that prompts leading to higher text feature dispersion result in better-calibrated predictions. Based on this insight, they propose the Average Text Feature Dispersion (ATFD) to optimize prompts for enhanced calibration. Their results show that ATFD has a negative correlation with ECE value (i.e., a higher ATFD value leads to a lower ECE value and vice versa). This analytical framework of linking text feature dispersion to calibration error provides a valuable perspective. While C-TPT offers a novel prompt-based strategy, our work demonstrates that calibration can be achieved through a feature and output space regularizations within a pseudo-labeling TTA framework. We will explore the integration of both prompt-tuning and image feature adaptation for CLIP-based TTA.
>
> [1]. C-TPT: Calibrated Test-Time Prompt Tuning for Vision-Language Models via Text Feature Dispersion. ICLR 2024
>
> **The corresponding revisions in the manuscript are highlighted in blue.**

---

### Meta-Review · Area_Chair_wU5g · 2025-12-22

**Summary:**

The reviews for this paper are mixed. This paper was reviewed by 3 experts in the field and received the following scores: 2 Marginal Reject (4), and 1 Marginal Accept (6).

This article proposes CLIP-TTA, a test-time adaptation method for vision-language models that aims to improve robustness in the face of distribution shifts without requiring labeled data. Building on previous approaches such as CLIP-OT, which rely on pseudo-labeling based on optimal transport, the authors identify unreliable pseudo-labels as a major limitation that can lead to unstable adaptation and performance degradation. To address this issue, CLIP-TTA introduces two complementary components: (1) a cosine similarity loss that aligns image features with textual prototypes to stabilize the adaptation direction, and (2) an information maximization regulator that encourages reliable and diverse predictions, thereby preventing collapse during adaptation. Extensive experiments on seven benchmarks, covering multiple types of corruption and domain shifts, demonstrate that CLIP-TTA consistently improves robustness and outperforms existing adaptation methods at test time with minimal computational overhead.


The reviewers generally agree that the paper is clearly written and that the problem of test-time adaptation for vision–language models is timely and important. The proposed CLIP-TTA method introduces two additional regularization losses on top of an optimal-transport-based pseudo-labeling framework and demonstrates consistent empirical improvements over the CLIP-OT baseline across several benchmarks.

**Reviewer Concerns:**

The reviewers raised several substantive concerns:

1. Limited novelty and marginal contribution. All reviewers noted that the basic framework is largely inherited from CLIP-OT, with the main novelty being the addition of two existing loss functions. This was considered a modest conceptual advance for ICLR, especially since the objective and optimization rely heavily on previous formulations.
2. Insufficient theoretical justification. Several reviewers pointed out that the paper does not provide a clear theoretical or evidence-based explanation of why cosine alignment and information maximization directly address overconfidence or ECE reduction. The link between the stated motivation and the proposed losses remains indirect and largely empirical.
3. Sensitivity and robustness issues. The method is highly sensitive to hyperparameter choices across tasks, raising questions about its robustness and practical deployability. This sensitivity undermines the article's claims of stable and reliable adaptation.
4. Limited experimental scope and generality. The evaluation is restricted to a single CLIP architecture and a limited set of benchmarks datasets, omitting widely used robustness datasets (e.g., ImageNet-C) and other VLM architectures. Consequently, it is difficult to determine whether the approach generalizes beyond the specific context of CLIP.
5. Positioning and clarity relative to ongoing work. Reviewers also raised concerns about potential overlap and confusion with recent, closely related work (e.g., benchmarks by Lafon et al.), as well as unanswered questions regarding the possible degradation of pre-trained representations by gradient-based adaptation.

Based on these concerns, and after carefully weighing the reviews, I side with the reviewers recommending rejection.

**Reviewer Scores:**

The initial score for P4FT was 4, indicating good writing quality as only strength of the paper. The reviewer noted three weaknesses and one question.

The first weakness concerns the limited contribution without a theoretical foundation (W1). The authors acknowledge that their article is simply an incremental contribution to CLIP-OT, but attempt to justify the lack of a theoretical foundation with a vague explanation, without providing any real theoretical support. This response is therefore unsatisfactory.

The second weakness relates to the fact that L_cos relies on the assumption of a highly distinctive textual prototype (W2).

The authors appear to have provided an explanation for this, as after discussion, the reviewers seem satisfied with their response for W2. The final weakness concerns the sensitivity to hyperparameters (W3).

The problem lies in the fact that the loss function proposed by the authors depends on hyperparameters that are extremely difficult to optimize and very sensitive, which is critical in a time-translation analysis (TTA) context. The authors acknowledge that optimal hyperparameters can vary from one dataset to another. This led the reviewer to be dissatisfied with the final answer. The authors attempted to provide an unreadable final answer, but the equations are not compiled… Regarding the performance gain, the authors acknowledge that the gains on fine-grained datasets are modest, unlike on the specialized corrupted dataset. I don't think this argument would have improved the score.

Furthermore, the reviewer 9utf initially awarded a score of 4 and noted 4 weaknesses and 2 questions. The first weakness (W1) relates to the lack of a theory justifying why `Lcos` reduces the ECE, which detracts from the paper's appeal. The authors claim to have added this theory in the revised version, but I found no theory to support this claim. The second weakness (W2) relates to the lack of challenging benchmarks. I think the authors have addressed this point. Another weakness (W3) is related to the fact that most of the experiments were performed on CLIP. To address this, the authors used other architectures, but on a corrupted version of the dataset. This seems to be a way of masking the technique’s poor performance on classical tasks. Furthermore, the authors don’t show the ECE, which makes their answer unconvincing. Finally, the last criticism concerns the incremental nature of their innovation, which the authors acknowledge.

In conclusion, reviewer dKuy initially awarded the paper 6 stars, citing a first weakness (W1) explaining that the author's proposed idea is incremental. The authors confirm this point. The reviewer then asks why the hyperparameters $\lambda_1$ and $\lambda_2$ (W3) are useless for accuracy on certain corrupted datasets. The authors answer this question. The reviewer then asks why CLIP-TTA only updates the visual encoder. I think the authors have also clearly answered this question. Finally, the reviewer (W4) raises an issue concerning the figure, and the authors suggest adapting it.

After reading all these reviews, it is difficult to understand how the debate initially took place, as there was virtually no interaction between the authors and the reviewers. As a member of the AC, I apologize for this lack of interaction. Furthermore, I observe that there appear to be two groups of reviewers: one with more weaknesses and the other with more strengths. The idea  developed in the paper is interesting, but I acknowledge that it is very incremental and that the results are good for corrupted datasets. However, comparing the performance of the strategies with those of [1], I feel that on standard datasets, the results are a bit catastrophic. For all these reasons, I prefer to reject the article. I believe it is not yet fully developed and could be improved with further development.

[1] Farina, Matteo, et al. "Frustratingly easy test-time adaptation of vision-language models." Advances in Neural Information Processing Systems 37 (2024): 129062-129093.

---

### Decision · Program_Chairs · 2026-01-26

Reject